# Ice-nucleating particles in precipitation samples from West Texas

Hemanth S. K. Vepuri[1,*], Cheyanne A. Rodriguez[1], Dimitrios G. Georgakopoulos[4], Dustin Hume[2], James Webb[2], Greg D. Mayer[3], and Naruki Hiranuma[1,*]

[1]Department of Life, Earth, and Environmental Sciences, West Texas A&M University, Canyon, TX, USA
[2]Office of Information Technology, West Texas A&M University, Canyon, TX, USA
[3]Department of Environmental Toxicology, Texas Tech University, Lubbock, TX, USA
[4]Department of Crop Science, Agricultural University of Athens, Athens, Greece

*Corresponding authors: hsvepuri1@buffs.wtamu.edu and nhiranuma@wtamu.edu

**Abstract**

Ice-nucleating particles (INPs) influence the formation of ice crystals in clouds and many types of precipitation. This study reports unique properties of INPs collected from 42 precipitation samples in the Texas Panhandle region from June 2018 to July 2019. We used a cold-stage instrument called the West Texas Cryogenic Refrigerator Applied to Freezing Test system to estimate INP concentrations ($n_{INP}$) through immersion freezing in our precipitation samples with our detection capability of > 0.006 INP L$^{-1}$. A disdrometer was used for two purposes; (1) to characterize the ground level precipitation type and (2) to measure the precipitation intensity as well as size of precipitating particles at the ground level during each precipitation event. While no clear seasonal variations of $n_{INP}$ values were apparent, the analysis of yearlong ground level precipitation observation as well as INPs in the precipitation samples showed some INP variations, for example, the highest and lowest at -25 °C both in the summer for hail-involved severe thunderstorm samples (3.0 to 1,130 INP L$^{-1}$), followed by the second lowest at the same $T$ from one of our snow samples collected during the winter (3.2 INP L$^{-1}$). Furthermore, we conducted the bacteria speciation using a subset of our precipitation samples to examine the presence of known biological INPs. In parallel, we also performed metagenomics analysis of ambient dust samples collected at commercial feedlots in West Texas to check the similarity and to test if local feedlots can act as a source of bioaerosol particles and/or INPs found in the precipitation samples. Overall, our results showed that cumulative $n_{INP}$ in our precipitation samples below -20 °C could be high in the samples collected while observing > 10 mm hr$^{-1}$ precipitation with notably large hydrometeor sizes and an implication of feedlot bacteria inclusion.

## 1 Introduction

### 1.1. What are INPs?

Aerosol particles play a major role in altering the cloud properties, precipitation patterns, and ultimately the Earth's radiation budget (Lohmann and Feichter, 2005). In the past few decades, the aerosol particle direct effects (i.e., the impact of aerosol particles on net radiation through scattering and absorption of solar radiation) have been extensively studied (Satheesh and Krishna Moorthy, 2005). For example, the global radiative forcing by sea salt aerosols and dust is known to be in the range of −0.5 to −2 W m$^{-2}$ and −2 to +0.5 W m$^{-2}$, respectively. However, the aerosol particle indirect effects (i.e., radiative impact due to formation of clouds) have been enigmatic. Some atmospheric aerosol particles are known to act as ice-nucleating particles (INPs) and catalyze

the formation of ice crystals in the clouds, but their overall impact on the Earth's radiative budget remains quantitatively uncertain (Lohmann et al., 2007).

While INPs are sparse in the atmosphere, they have substantial impacts on the cloud microphysics and the precipitation formation (DeMott et al., 2010). The sources of atmospheric INPs are diverse as they emerge naturally and also through human activities, adding complexities to our comprehensive understanding in their impacts (e.g., Kanji et al., 2017; Zhao et al., 2019). In general, INPs provide a surface on which the water vapor and/or cloud droplet deposits and freezes (Van den Heever et al., 2006). This type of ice formation in the presence of INP is known as heterogenous freezing (Vali et al., 2015). In the absence of INPs, the formation of atmospheric ice particles follows the process of homogeneous nucleation, in which it requires the cloud droplet to be supercooled to the temperature ($T$) of -32 °C and below (depending on the pure water droplet size) to form ice crystals (Koop et al., 2000; Koop and Murray, 2016). Though our knowledge regarding INPs remains insufficient, there have been advances in understanding the different modes of heterogeneous ice nucleation (IN) in the atmosphere in the last few decades. For example, deposition nucleation is induced by the direct deposition of water vapor on to an INP's surface and ice embryo formation on the surface under ice supersaturation conditions (Kanji and Abbatt, 2006; Möhler et al., 2008). Recently, some studies have argued that the deposition nucleation could be interpreted as pore condensation and freezing (Marcolli, 2014). The presence of water in pores of mineral materials and the resulting inverse Kelvin effect cause an instantaneous water saturation condition in the confined space, allowing the water to freeze even at water sub-saturated ambient conditions (David et al., 2019; Marcolli, 2014). Amongst various IN paths, perhaps the most important mode is immersion freezing (De Boer et al., 2010). This process starts with the formation of cloud droplet followed by freezing due to an INP immersed in the supercooled droplet. In addition, the past studies have identified other modes of heterogeneous nucleation, such as condensation freezing (Belosi and Santachiara, 2019), contact freezing (Hoffmann et al., 2013), and inside-out evaporation freezing (Durant and Shaw, 2005). These modes are relatively less relevant in the mixed-phase clouds (MPCs) as discussed in the next section.

*1.2. Importance of Immersion Freezing*

INPs greatly influence cloud properties, especially in MPCs, which are typically observed in the altitude range of 2 km to 9 km above ground level (Hartmann et al., 1992). Out of all heterogeneous ice-nucleation modes, the immersion freezing is the most dominant mode of ice formation in MPCs (Ansmann et al., 2008; De Boer et al., 2010; Hande and Hoose, 2017; Vergara-Temprado et al., 2018). In Hande and Hoose (2017), different cloud types such as orographic, stratiform, and deep-convective systems were simulated and analyzed for different freezing modes under various polluted conditions. The authors demonstrate that immersion freezing is the predominant IN mode under various simulated circumstances, accounting for 85 to 99%, while other IN paths play a less significant role. Cui et al. (2006) also showed that immersion freezing is the primary mode of ice formation with little significance of the deposition mode in the early stages of the cloud development. Moreover, whereas contact freezing may be a highly efficient ice formation path, a previous simulation study showed that it is a negligible mode in the given MPC conditions (Phillips et al., 2007). Field et al. (2012) and De Boer et al. (2011) showed that the formation of cloud droplets is a precondition for ice formation in MPCs, thus highlighting the importance of immersion nucleation. Furthermore, using multiple ground-based instruments, including Lidar, AERONET Sun Photometer, and Vaisala Radiosonde, Ansmann et al. (2008) found that a high INP concentration ($n_{INP}$) (i.e., ~ 1 – 20 cm$^{-3}$) in the Saharan dust. This high dust-including $n_{INP}$ episode coincided with the presence of liquid droplets at cloud tops at $T$s of -22 to -25 °C. Similarly, Ansmann et al. (2009) shows the observation of tropical altocumulus clouds having liquid cloud tops. Due to the importance and dominance of immersion freezing,

the current study focuses on measuring the immersion freezing efficiency of the precipitation samples collected in the Texas Panhandle region.

### 1.3. INPs in Precipitation

It is known that INPs in MPCs have a notable impact on the properties of precipitation. Previously, Yang et al. (2019) studied the effect of INPs on cloud dynamics and precipitation through model simulations of an observed severe storm in Northern China. The authors show that an increase in INPs can enhance the storm, whereas an

90 excessive increase of INPs may impede the updrafts in the storm. The reason for this complex effect of INPs may be explained by the variation in the latent heat release in the convective system at different stages of its development. This latent heat is further influenced by INP episode, thus affecting the dynamics of the precipitation system. Furthermore, the increase in INP number might reduce the mean hail diameter (hail particles with smaller diameters melt more easily), which leads to decreased hail precipitation and an increased

rain formation in contrast to the previous studies (Fan et al., 2017; Van den Heever et al., 2006). Similar results have been found by Chen et al. (2019). The authors show that an increased $n_{INP}$ in the simulated hailstorm can reduce the graupel size and the concentration of hail stones. Likewise, the aircraft observations along with the model simulations of convective storms in West Texas and U.S. High Plains have shown that the addition of INPs at the base of warm clouds would result in an increase of the precipitation amount by strong updrafts in the

system (Rosenfeld et al., 2008), ultimately affecting the local hydrological cycle (Mülmenstädt et al., 2015). It has also been observed that INPs can be removed from the atmosphere through precipitation resulting in a net decrease in $n_{INP}$, affecting the precipitation development (Stopelli et al., 2015).

    Several previous studies have characterized the $n_{INP}$ in the precipitation samples from various locations (Creamean et al., 2019; Petters and Wright, 2015; Levin et al., 2019). Petters and Wright (2015) reported a wide

range of $n_{INP}$ values in their local precipitation samples collected approximately 3 km west of Raleigh, NC, USA for July 2012 and October 2013. Their study shows a variation of 10 orders of magnitude in the concentrations of INPs with a high variability in the $T$ range of -5 to -12 °C, suggesting inclusion of biological INPs, which are generally known to be active at relatively high freezing $T$s (Després et al., 2012). The lower limit for the INP spectrum as a function of $T$ derived from the cloud water and precipitation samples in Petters and Wright (2015) may highlight

the extreme rarity of INPs at $T$s warmer than -10 °C. Particularly, the authors showed that the highest ever observed $n_{INP}$ at -8 °C were three orders of magnitude lower than observed ice crystal concentrations in tropical cumuli at the same temperature. More precipitation studies may provide a constraint on minimum enhancement factors for secondary ice formation processes. In Levin et al. (2019) the $n_{INP}$ values during an atmospheric river event on the west coast of United States were studied. The authors found an increased concentration of marine

INPs in contrast to their previous studies, showing high mineral/soil dust during an atmospheric river precipitation.

### 1.4. Study Objectives

In this study, we characterized properties of INPs in precipitation samples collected in the Texas Panhandle region.

All of our samples were analyzed at our laboratory using a cold stage instrument. The estimated $n_{INP}$ in the precipitation samples were studied with ground level precipitation properties, such as the precipitation type, intensity of precipitation (mm hr$^{-1}$), and hydrometeor particle size (mm). A subset of the collected precipitation samples was analyzed for their bio-speciation to characterize potential biological INP sources in the West Texas region and also to examine the presence of known high T biological INPs. Although the estimation of $n_{INP}$ in

precipitation samples collected at the ground level does not represent INPs at the cloud height, we report the

INPs resolved by the ground level weather observation that help understanding of ambient INPs in the West Texas region, where unique and substantial INPs, i.e., several hundred and thousand INPs $L^{-1}$ at -20 °C and -25 °C, respectively, are consistently emitted from animal feeding operations (Hiranuma et al., 2020).

## 2 Methods

*2.1 Precipitation Sampling*
The precipitation samples were collected from different seasons throughout the year during June 2018 – July 2019. Sterilized polypropylene tubes of 50 ml volume (VWR® Centrifuge Tube) were used as sampling gauges. The gauges were placed at ~50 ft above the ground on the rooftop of the Natural Science Building at West Texas A&M University, Canyon, TX. This particular location was chosen to avoid any obstruction of our sampling
activities. The sampling tubes were well exposed to the ambient air without any canopies throughout the sampling process. The sampling gauges were replaced every 24 hours to minimize the effect of dry deposition prior to the precipitation sample collection. A blank dry deposition sample (Sample# 34) was specifically collected for 24 hours from January 2-3, 2019 in order to examine and quantify the effect of dry deposition on $n_{INP}$. The freezing spectrum of this dry deposition sample (suspended in HPLC grade pure water) was later compared with
the IN spectra of precipitation samples (see **Sect. 3.3**). We note that a volume of pure water (5 ml) for an atmospheric INP estimate based on a dry deposition sample was determined by averaging collected precipitation volumes of all samples prior to this dry deposition sample. For the duration of a given precipitation episode, some amount of sample was accumulated in the tube. The sampling tubes were then capped and stored at $T$ of 4 °C in the refrigerator, following the method described in Petters and Wright (2015), until the droplet-freezing assay
experiments were commenced. The effect of storage conditions on the IN activity was not considered in this study. We note that Beall et al. (2020) recently found a decrease in precipitation $n_{INP}$ by 42% when stored at 4 °C (i.e., Table 5) and suggested correction factors for the $T$ range of -7 to -17 °C. After the freezing experiment, a subset of our samples was kept under deep-freeze conditions (-80 °C) for further biological analysis (see **Sect. 2.6**). In total, 42 precipitation samples were collected from different weather systems observed at the surface
level. Based on these samples and observations, we estimated the $n_{INP}$ values from (1) snow, (2) hails/thunderstorm, (3) long-lasted rain, and (4) weak rain. More information about the samples used in this study, precipitation types and the amount of the precipitation collected for each sample are provided in the **Supplemental Information (SI) Sect. S1**.

*2.2. Disdrometer Measurements of Precipitation Properties*
For our precipitation measurements, we used the OTT Parsivel[2] (Particle Size Velocity 2) sensor. This device is a modern laser-optical disdrometer ($\lambda$ = 780 nm) which measures the size and fall velocity of precipitating particles. The OTT Parsivel[2] was deployed in side-by-side position with the precipitation gauge collector for the duration of our study period. A detailed technical description of OTT Parsivel[2] is given in a
160 previous study (Tokay et al., 2014), so only a brief description is provided here. A combination of the laser transmitter and receiver component was integrated as a single cluster in a weatherproof housing and detects precipitation particles passing through a horizontal strip of light. A nominal cross section area of a laser beam detection was 54 $cm^2$, and the system recorded the number of hydrometeors in a 32 x 32 matrix (i.e., fall velocity x diameter) in the ≥ 30 seconds time resolution. The measurable size range of hydrometeor particles was 0.062 -
165 24.5 mm in diameter ($D_p$) with bin size intervals ($\Delta D_p$) varying from 0.125 to 3.0 mm. Our disdrometer was coupled

with an OTT netDL Hydrosystem logger (40 channels). The OTT Parsivel[2] also measured the intensity of precipitation (mm hr$^{-1}$) and the number of precipitation particles passing through the horizontal strip of light in the event of precipitation. The OTT Parsivel[2] automatically categorized the precipitation type according to the National Weather Service (NWS) weather code based on the measured precipitation properties. Due to the intermittent nature of the precipitation, the OTT Parsivel[2] assigned multiple NWS precipitation codes during a single precipitation event (**Table S1** column 'NWS Code'). We compared our manual observations with the NWS precipitation code assigned by the disdrometer, and we categorized all observed precipitation into four different types. These four major precipitation types defined in this study included snow, hail/thunderstorm, long-lasted rain, and weak rain, and we collected 6, 18, 13, and 5 samples from each type, respectively, which sum up to a total of 42 samples More detailed methodology of precipitation categorization is discussed in **SI Sect. S1.1**.

### *2.3 IoT Air Quality Sensor Measurements*

A cluster of Arduino-based Internet of Things (IoT) air quality sensors was developed to measure ambient air conditions at our precipitation sampling location. This IoT cluster was deployed alongside the disdrometer and sampling gauge to complement this study. A DFRobot particulate matter (PM) laser dust sensor measured PM with size ranges of < 1 μm (PM$_{1.0}$), < 2.5 μm (PM$_{2.5}$), and < 10 μm (PM$_{10}$) with an estimated uncertainty of ±27% relative to an optical particle counter (Markowiz and Chiliński, 2020). Other ambient conditions, including $T$, barometric pressure, and humidity, were measured with a precision Bosch BME280 environmental sensor. We calibrated our sensors against a commercially available sensor (GlobalSat Inc., LS-113). Our sensors utilized Long Range and Wide Area Network (LoRaWAN) technology for data transmission. A LoRaWAN transceiver is connected to our sensors for wireless data transmission. This small IoT device operated with 915 MHz signal frequency, transmitting encrypted and signed packets of captured air quality data through a hosted LoRa network server to a Kibana visualization server. This data interface enabled in situ monitoring and processing of the data. The PM concentrations were later time-averaged for assessing contribution of wet scavenging of aerosol particles to $n_{INP}$ in the precipitation samples.

### *2.4 Immersion Freezing Experiment*

All immersion freezing experiments in this study were conducted using an offline instrument called West Texas - Cryogenic Refrigerator Applied to Freezing Test (WT-CRAFT) system (Hiranuma et al., 2019; Cory et al., 2019). The WT-CRAFT system is a cold-stage technique, in which the droplets are placed on an aluminum plate and cooled until they are frozen. A commercially available digital camera was used to record the droplet freezing events, and we visually evaluated the freezing $T$s based on the shift in droplet brightness while freezing. If there was an uncertainty in determining the $T$ at which a droplet was completely frozen, we used the ImageJ software for further image analysis of those droplets (see Table S4 in Hiranuma et al., 2019). This system was used to obtain $T$-resolved $n_{INP}$ in -25 °C < $T$ < 0 °C. The lower $T$ limit was -25 °C to ensure measuring INPs with negligible artefacts (Hiranuma et al., 2019). Our system is susceptible to low INP detection, and the minimum INP detection limit of the WT-CRAFT system for this study was 0.006 L$^{-1}$ air. To minimize any contamination during the IN measurement, the WT-CRAFT system was placed in a ventilated fume hood. For each experiment an aluminum plate surface was freshly coated with a thin layer of thermally conductive and IN-inert Vaseline to physically isolate individual droplets from the aluminum surface (otherwise, aluminum can act as a heterogeneous IN surface). A total of 70 suspension droplets of 3μL volume each were prepared for each run. The aluminum plate with the droplets on it was then placed inside a portable cryogenic refrigerator (Cryo-Porter). Freezing $T$s were measured by the sensor

taped on the aluminum surface with a resolution of 0.1 °C, and the external keypad controller was used to control cooling rate (°C min⁻¹). In this study, the freezing experiments were carried out at a cooling rate of 1 °C min⁻¹. The validity of using this cooling rate and another test regarding time trial aspect are demonstrated in **SI Sect. S2** (**Figs. S1 and S2**). The droplets were cooled until all 70 droplets were frozen before warming up the system to 5 °C to be prepared for a subsequent experiment.

If all the droplets were frozen at $T$ > -25 °C, a HPLC-grade ultrapure water was used to prepare different serial dilutions for the precipitation samples. The diluted suspensions were made to compute the $n_{INP}$ down to -25 °C. Some of our precipitation samples were diluted until the frozen fraction (the ratio of number of droplets frozen to the total number of droplets) curve was conformed to the background curve (i.e., frozen fraction curve for the HPLC ultrapure water). At the end of each WT-CRAFT experiment, the frozen fraction and ambient $n_{INP}$ were estimated as a function of $T$ with an interval of 0.5 °C. The IN measurements from the undiluted and diluted runs were merged by taking the lower $n_{INP}$ values, which typically possess the lowest confidence intervals, for the overlapped $T$ region.

The total systematic $T$ and $n_{INP}$ uncertainties in WT-CRAFT are ±0.5 °C and ±23.5% (Hiranuma et al., 2019). For this study, the experimental uncertainty in our estimated $n_{INP}$ was evaluated and reported using the 95% confidence interval method described in Schiebel (2017). Background contamination tests for WT-CRAFT were carried out weekly to make sure negligible background freezing at -25 °C. In this study, we consider the frozen fraction ≤ 0.05, accounting for less than 3% of pure water activation, as negligible background (Hiranuma et al., 2019). For these background tests, only HPLC grade ultrapure water was used for preparing the droplets.

*2.5 IN Parameterization*

Here we describe the parameterization used to estimate ambient $n_{INP}$. Initially, we computed the $C_{INP}(T)$ value, which is the nucleus concentration in precipitation suspension (L⁻¹ water) at a given $T$ as described in Vali (1971). This $C_{INP}(T)$ value was calculated as a function of unfrozen fraction, $f_{unfrozen}(T)$ (i.e., the ratio of number of droplets unfrozen to the total number of droplets) as:

$$C_{INP}(T) = -\frac{\ln\left(f_{unfrozen}(T)\right)}{V_d}$$

(1)

in which, $V_d$ is the volume of the droplet (3 $\mu$L). Next, we used the cloud water content (CWC) parameter in order to convert $C_{INP}(T)$ to $n_{INP}(T)$, INP in the unit volume of atmospheric air at standard $T$ and pressure (STP) conditions, which is 273.15 K and 1013 mbar. We assumed CWC to be a constant of 0.4 g m⁻³, following Petters and Wright (2015). This assumption would be reasonable for the following three reasons: (1) Petters and Wright (2015) and references therein showed typical values of CWC for different cloud types could narrowly range from 0.2 g m⁻³ to a factor of few more, (2) the authors also showed that the variation of $n_{INP}$ with CWC values for different cloud types in the atmosphere would typically be limited within a factor of two, and our $n_{INP}$ uncertainties could be larger than that, and (3) based on a parametrization for rainwater evaporation, Zhang et al. (2006) suggests that evaporation does not contribute to $n_{INP}$ bias for both strong convective systems and persistent rain events with cloud base heights of ≈3 km. Thus, the variation of CWC on the $n_{INP}$ was considered to be negligible. Nonetheless, it is necessary in the future to further investigate in cloud specific CWCs incorporating with loss of water through partial evaporation of raindrops during free fall based on vertical vapor deficit profiles to conclusively assess if this assumption is fair or not. Precipitation evaporation rate might

introduce bias in $n_{INP}$ for precipitation systems with high cloud base, and the correction can be applied accordingly (Petters and Wright, 2015). Direct comparison between INP measurements in cloud water samples and those in precipitation samples might also be key to answer this question (e.g., Pereira et al., 2020).

The sample air volume ($V_{air}$) at the cloud level was calculated by converting the volume of the precipitation sample collected ($V_l$) using the Eqn. (2) from Petters and Wright (2015):

$$V_{air} = \frac{V_l \times 1000 \times \rho_w}{CWC}$$

(2)

where $\rho_w$ is a unit density of water (1 g ml$^{-1}$). $V_{air}$ is in liters (L), whereas $V_l$ is given in ml. The multiplication
factor '1000' is used to convert the volume from cubic meter (m$^3$) of air to liter of air. The cumulative $n_{INP}$ per unit volume of sample air, described in the previous study DeMott et al. (2017), was then estimated as:

$$n_{INP}(T) = C_{INP}(T) \times DF \times \frac{V_l}{V_{air}}$$

(3)

where DF is a serial dilution factor (e.g., DF = 1 or 10 or 100 and so on).

### 2.6. Microbiome of Feedlot Dust and Precipitation Samples

The overall goal of our metagenomics analysis was to identify known ice-nucleation-active bacterial and fungal species in feedlot dust and precipitation samples collected in the West Texas region. This biological speciation is also useful to examine if local feedlots can act as a source of bioaerosol particles and/or INPs found in the precipitation samples. In this study, we have examined a heterogeneous set of samples including four feedlot samples locally collected on March 28, 2019 and July 22, 23, and 24, 2018 (see Table 1 of Hiranuma et al., 2020),
precipitation samples (Sample# 1, 2, 7, and 50), and a 24-hour dry deposition sample (Sample# 34). We note that the precipitation Sample# 50 (another hail/thunderstorm sample), which was collected on March 23, 2019 when a tornado warming was issued, was preserved only for metagenomics due to its low volume (≈ 1ml). It is also noteworthy that we attempted to analyze samples of all precipitation types, but acquired quantitative results only for those hail/thunderstorm samples (the reason is unknown). Next, we describe our microbiome analysis
procedure in four different steps, including (1) DNA Extraction, (2) 16S rRNA Amplicon Diversity Sequencing, (3) Bioinformatics, and (4) Data Analysis. For DNA extraction, genomic DNA was first extracted from all samples using PowerSoil DNA Isolation Kits (MoBio Laboratories, Inc., Carlsbad, CA, USA). Extraction proceeded following the manufacturer's protocol, with the following minor changes: solutions C1 and C6 were heated to 65 °C and solution C6 was allowed to remain on the filter membrane for at least one minute before centrifugation. Additionally, the
C6 step was repeated. Library preparation for bacterial 16S DNA amplicon sequencing utilized primers for the V1-V3 hypervariable region of the 16S gene. These primers were constructed for the 16S amplicon using a combination of the 28F and Illumina i5 sequencing primer and the Illumina i7 sequencing primer with the 519R primer. Amplifications were performed in 25 μl reactions with Qiagen HotStar Taq master mix (Qiagen Inc, Valencia, CA, USA). Reactions were performed with 1 μl of each 5μM primer and the template DNA. Amplification
was performed on an ABI Veriti thermocycler (Applied Biosytems, Carlsbad, CA, USA) under the following thermal profile: 95 °C for 5 min, then 25 cycles of 94 °C for 30 sec, 54 °C for 40 sec, 72 °C for 1 min, followed by one cycle of 72 °C for 10 min and 4 °C hold. An ethidium bromide-stained gel was used to qualitatively determine the

amount of the amplification product to add to the second amplification stage. Primers for the second PCR were designed based on the Illumina Nextera PCR primers. The second stage amplification proceeded using the same

cycling protocol as the first round, except it was amplified for only 10 cycles. SPRIselect beads (BeckmanCoulter, Indianapolis, IN, USA) were used at a 0.7 ratio to size-select the DNA amplicons from an equimolar pooled sample. Pooled samples were then quantified using a Quibit 2.0 fluorometer (Life Technologies) and loaded on an Illumina MiSeq (Illumina, Inc. San Diego, CA, USA) 2x300 flow cell at 10pM.

      For bioinformatics, raw data were initially processed using a standard microbial diversity analysis pipeline

(QIIME2-2020). Raw data was first checked for sequencing quality and chimeric sequences, before being parsed through a microbial diversity pipeline. During the cleanup stage; denoising of the raw data was performed using various techniques to remove short sequences, singleton sequences, and reads with poor quality scores. Next, chimera detection software was used to filter out any potentially chimeric sequences. Finally, remaining high-quality sequences were corrected base by base to check for sequencer miscalls. The diversity analysis pipeline

clustered all sequences based on 97% similarity to yield operational taxonomic units (OTUs), before running a seed sequence from each OTU through a taxonomic database curated in-house by RTLGenomics. Finally, the taxonomy was assigned to each sequence using a classifier that was pretrained on GreenGenes database with 99% OTUs. The relative abundance of bacterial taxa within each sediment sample was determined by dividing each OTU by the total number of reads.

**3 Results and Discussion**

*3.1 Ambient and Precipitation Properties*

The time series summary of ambient and precipitation properties measured by our disdrometer as well as IoT cluster is shown in **Fig. 1**. Each data point in **Fig. 1a** shows the average temperature measured over the sampling period of a given precipitation event. A notable seasonal variation of ambient $T$ at our sampling location was

observed.  The highest average temperature measured during a precipitation event was 34.9 ± 12.2 °C, which was in the summer of 2018 (i.e., ID# 7; a long-lasted rain sample), while the lowest $T$ was -6.5 ± 6.7 °C, measured during the winter of 2018 (i.e., ID# 23; a snow sample). The annual mean $T$ for Canyon, TX region measured at our sampling site was 17.7 °C. The diurnal cycles of ambient properties are not shown in **Fig. 1a**. Nevertheless, we typically observed suppression of $T$ before precipitation events in our study. It  is known that the $T$ gradient

plays a major role in the development and growth of the precipitation systems (Vaid and Liang 2015).  Next, each relative humidity data point shown in **Fig. 1b** corresponds to the average during each precipitation event.  With an overall average of 54.0%, the highest and lowest relative humidity values measured were 70.7 ± 2.3 % (ID# 26; a weak rain sample) and 30.8 ± 0.7 % (ID# 7; a long-lasted rain sample). The observed low ground level relative humidities during some precipitation events (**Tables S1 - S2**) may be a concern as loss of water through partial

evaporation of hydrometeors during free fall. But, it is noteworthy that the water evaporation might have negligible effect on $n_{INP}$ estimated from precipitation samples as discussed in **Sect. 2.5**. Third, **Fig. 1c** displays the time series of the cumulative number of detected precipitation particles in individual precipitation events and the overall mean number of detected particles (dashed line). In our study period, a disdrometer detected a substantial number of precipitation particles with a cumulative number ranging from 1.0 x $10^4$ to 6.6 x $10^5$

particles passing through its laser beam cross section per event. More details of each precipitation event and its properties are shown in the **Tables S1 - S3**. As seen in **Table S3**, high numbers of precipitation particles  were observed in conjunction with snow/hail-involving precipitation events during our study period, which may

increase the wet scavenging efficiency of ambient aerosol particles during precipitation (see **Sect. 3.2 and SI Sect. S4**). Out of all the 42 samples, the highest number of precipitation particles was detected on the 5[th] of Nov, 2018 (ID# 19; a snow sample), while the lowest was observed on the 2[nd] of Sep, 2018 (ID# 13; weak rain). Finally, **Fig. 1d** shows the average, maximum, and minimum precipitation intensity (mm hr$^{-1}$) measured during each precipitation event. Due to the intermittent nature of the precipitation, the intensity widely ranged from 1.1 to 129.3 mm hr$^{-1}$ per event. The highest maximum intensity of 129.3 mm hr$^{-1}$ was measured during a hail/thunderstorm event (ID# 40), while the lowest was 1.1 mm hr$^{-1}$ during a snow event (ID# 23). These intensity data were used for our wet deposition analysis (**SI Sect. S4**).

The variation of precipitation properties was further investigated by analyzing the size distribution of precipitation particles measured by the OTT Parsivel$^2$ disdrometer. **Figure 2** shows the precipitation particle size distribution for each category of ground level observed precipitation type. The size of precipitation particles was represented at the median diameter of the corresponding disdrometer's size bin. As shown in the **Fig. 2a and 2b**, both snow and hail/thunderstorm samples had particles of diameter greater than 10 mm with the maximum particle diameter of 17 mm. Although there are three episodes of long-lasted rain with a particle diameter greater than 14 mm (**Fig. 2c**), a clear trend of overall decrease in the hydrometeor size was seen for this category as well as the weak rain samples (**Fig. 2d**). In fact, all weak rain samples contained particles only smaller than 6.5 mm. Moreover, the mode precipitation particle diameter for the snow, hail/thunderstorm, and long-lasted rain samples was 0.44 mm, whereas it was 0.31 mm for the weak rain samples (see **Table S3**). This variation in mode diameter along with the results shown in **Fig. 2** generally exhibited the shift in hydrometeor particle size distribution towards a larger diameter with an increased intensity of precipitation at the ground level.

### 3.2 IoT Air Quality Sensor Results and Implication of Wet Deposition

The overall mean PM concentrations (± standard error) measured by an IoT air quality sensor for our study period were 3.9 ± 9.2 x 10$^{-2}$ $\mu$g m$^{-3}$ (PM$_{1.0}$), 4.0 ± 4.5 x 10$^{-2}$ $\mu$g m$^{-3}$ (PM$_{2.5}$), and 10.0 ± 2.2 x 10$^{-1}$ $\mu$g m$^{-3}$ (PM$_{10}$). Although there was an inconsistent variation of PM concentrations with precipitation type, we observed a substantial increase in all PM values for the period July – Aug 2018 and May 2019. In contrast, a decrease in all PM concentrations was observed during Sep 2018 – Mar 2019. This increase in PM values during summer and decrease during winter suggested a seasonal variation at the sampling site. The seasonal variation in PMs may be indicative of different aerosol particle sources or the local meteorological conditions. In the Southern Great Plains, the local sources include harvesting crop fields and agricultural burning (Garcia et al., 2012; DeMott et al., 2015). Based on the long-term measurements of aerosol particle composition at Southern Great Plains (SGP), Parworth et al. (2015) found a seasonally varying interstate transport of biogenic aerosols to the SGP site. The authors also observed a springtime increase in biomass burning organic aerosols at SGP, which were mainly associated with local fires. The long-distance dispersion of *Juniperus ashei* pollen into the SGP area by the southern winds was previously observed by Van de Water et al. (2003). Elevated layers of haze have been observed over the same site due to the inter-oceanic and intercontinental transport of smoke from intense Siberian fires (Arnott et al., 2006; Damoah et al., 2004). It was also evident from previous observation and simulation modeling studies that Saharan dust can reach southeastern parts of USA through the transatlantic long-range transport (Weinzierl et al., 2017). Thus, PMs observed in the West Texas region may be a mixture of aerosol particles from different sources and spatial scales of transport.

**Table 1** shows the hourly time-averaged PM data measured prior to vs. after precipitation. During intense precipitation, aerosol particle concentrations below cloud tend to decrease due to the wet scavenging effect (Hanlon et al., 2017). In fact, the reduction in our hourly averaged PM$_1$, PM$_{2.5}$, and PM$_{10}$ after precipitation is

apparent in **Table 1**, presumably because of scavenging in part at least. Note that any counter mechanisms, such as primary biological aerosol particles and surface material rupture after rainfall (e.g., Huffman et al., 2013), were not considered in our data interpretation. The first order calculations are performed to understand implications of scavenging processes towards the reduction in the PM after rain event (**SI Sect. S4**). These calculations contain ±61.5% uncertainty and can be further extended with some assumptions to estimate INP. However, to better constrain these estimates, direct vertical INP (He et al., 2020) and scavenging measurements (Hanlon et al., 2017) are needed. A total of 28 precipitation events was analyzed, and our estimated $n_{INP}(T)$ of scavenged aerosol particles appeared to be constantly an order magnitude lower as compared to total $n_{INP}(T)$ measured in our precipitation samples (**Fig. S3**). This trend is true across all ranges of examined $T$s (> -25 °C). Nevertheless, our estimates imply some (but negligible) contributions of scavenged aerosol particles on $n_{INP}(T)$ in our precipitation samples.

### 3.3 INP Results

The time series of cumulative $n_{INP}$ from precipitation samples at different $T$s (i.e., -5, -10, -15, -20, and -25 °C) are shown in **Fig. 3**. The $T$-resolved averaged cumulative $n_{INP}$ ± standard error is also presented in **Fig. 3**. Note that **Fig. 3b** shows $n_{INP}$ for two precipitation samples (ID# 26 and 27) observed on the same day of 12 March 2019. Overall, three orders of magnitude variations of averaged cumulative $n_{INP}$ values were observed between -10 °C (0.17 ± 0.04 $L^{-1}$) and -25 °C (74.74 ± 28.28 $L^{-1}$) for our precipitation samples. Occasionally, we observed $n_{INP}$ detected at ≥ -5 °C, but such a high $T$ INPs was randomly found in only 7 out of 42 samples within our detection capability.

Attempts to examine the distribution of $n_{INP}$ based on the precipitation type, meteorological season, and maximum precipitation intensity (mm $hr^{-1}$) were made (see **SI Sect. S5**). Due to the limited total number of samples we collected, we cannot conclusively state anything regarding seasonal variations of $n_{INP}$ in our precipitation samples. Nonetheless, our INP results showed that the lowest $n_{INP}$ at -25 °C (3.0 $L^{-1}$) was found in a hail/thunderstorm sample (ID#37; no inclusion of large hydrometeors as seen in **Fig. 2b**) collected during the summer 2019. Likewise, the highest $n_{INP}$ at -25 °C (1,130 $L^{-1}$) was found in a hail-involved severe thunderstorm sample (ID# 1) collected in summer 2018. This observation is interesting because the measured $PM_{10}$ of ~6.2 µg $m^{-3}$ prior to precipitation of ID# 1 (**Table 1**) is not the highest $PM_{10}$ recorded in 2018-2019, suggesting wet scavenging does not control the total INPs in precipitation samples. The fact that the second lowest $n_{INP}$ (-25 °C), which is 3.2 $L^{-1}$, is from the snow sample (ID# 23) also supports a negligible contribution of scavenging in our INP data. Moreover, our results showed that cumulative $n_{INP}$ below -20 °C in our precipitation samples could be high in the samples collected while observing > 10 mm $hr^{-1}$ hail/thunderstorm and snow precipitation with notably large hydrometeor sizes.

**Figure 4** shows a compilation of $n_{INP}(T)$ spectra of each precipitation type in comparison to previously reported precipitation $n_{INP}(T)$. In general, most of $n_{INP}$ spectra fall in the upper range of the previous precipitation $n_{INP}$ data presented in Petters and Wright (2015) and Vali (1968). INP humps shaping the reference spectra (i.e., one below -20 °C and another at > -20 °C) are also found in our spectra. The observed hump is especially obvious for $n_{INP}$ at $T$ above -20 °C, and some of our spectra exceed the upper bound of the reference spectra in any precipitation types. For $T$s below -20 °C, our $n_{INP}(T)$ data match fairly well within the range of the reference $n_{INP}(T)$ for all four precipitation types. Thus, the precipitation type observed at the ground level would not have any relationships with INP propensity at least for our 42 samples collected for this study. However, it is interesting that most of our $n_{INP}$ data points above -15 °C fall within the range of estimated $n_{INP}$ at cloud height with < 50% storm efficiency, reported in Vali (1968). In fact, regardless of precipitation type, we see reasonable overlaps of

our $n_{INP}(T)$ with Vali (1968). The author stated that the large differences in IN content among precipitation samples were mainly caused by differences in the nucleus content of the air entering the storm. This implies that the cloud level dynamics like cloud entrainment impact the cloud level INP concentrations. Hence, we compared our precipitation INP data with the lower and upper limits of the IN concentrations in the air entering the storm given by Vali (1968) (Table 2, Chapter# 9). These cloud level INP concentrations given by Vali (1968) were for two different storm efficiencies, which is the ratio of mass of precipitation to the mass of water input. The storm efficiency of 10% represents the time when high concentrations of precipitation inside the storm begins to develop. Likewise, 50% is at the peak intensity of the storm. These different combinations of storm efficiencies and water content accounted for a tenfold variation in the ice nucleus content. As more air is entered into the storm with 50% efficiency, more IN concentrations are observed at cloud level. Though our data are comparable to Vali (1968), there is still indeed the need for cloud level INP measurements to define the relationship between the ground level INP concentrations and precipitation intensity.

In addition, **Fig. 4** also shows the $n_{INP}$ result of our 24-hour dry deposition blank sample. For the measured $T$ range, $n_{INP}$ values from the dry deposition blank sample were at least an order of magnitude lower than that from our precipitation samples. This finding corroborated our assumption of negligible contribution of dry deposition in our WT-CRAFT estimated $n_{INP}$ from precipitation samples.

**Figure 5** shows another compilation plot of our precipitation $n_{INP}(T)$ spectra compared to ambient $n_{INP}(T)$ data of local agricultural dusts from Fig. 3 of Hiranuma (2020). As seen, most of our precipitation INP spectra are accumulated near the lower end of the feedlot IN spectra, implying some inclusion of these local dusts as INPs in our samples. Although we are not certain if these local dusts play a role in precipitation, and assessing the potential of locally emitted aerosol particles to precipitation formation is beyond the scope of the current study, it is important to study the contribution of local agricultural dust in wet scavenging and INP formation at cloud height separately in the future. It is noteworthy that adjacent feedlots (> 45,000 head capacity) are located within 33 miles of our sampling site, and the role of feedlot dusts in atmospheric INPs is described in more detail in Hiranuma et al. (2020). Further discussion regarding the feedlot contribution in INPs in our precipitation samples is provided in **Sect. 3.4**.

### 3.4. Microbiome of Feedlot and Precipitation Samples

Furthermore, we conducted the bacteria speciation of a subset of our precipitation samples and ambient dust samples collected at commercial feedlots in West Texas to identify potential biological sources of INPs in our precipitation samples.

We successfully generated data on the bacterial microbiome of our precipitation and feedlot dust samples. Unfortunately, our attempt to extract fungal microbes was not successful due to the limitation in sample amount. Thus, we focus on bacterial discussions hereafter. In most cases, bacterial phyla were classified to the level of genus. The majority of bacteria in all samples belonged to phyla *Proteobacteria* and *Bacteroidetes* (**Fig. 6** and **Table S9**). In hailstorm samples, the main taxa of *Proteobacteria* were *Massilia* (a genus found in clinical samples and mammals, but also the soil, rhizosphere, and even aerosols), genera belonging to the order *Sphingomonadales* (bacteria with wide metabolic abilities), *Caulobacterales* (bacteria living in diverse terrestrial and aquatic habitats; some are minor human pathogens), and *Rhizobiales* (nitrogen-fixing bacteria forming symbioses with the roots of legumes). Among the *Bacteroidetes* phylum, the genus *Marinoscillum* was relatively the most abundant. This genus is a recently described marine bacterium, and it is interesting that it was found in hailstorm samples at percentages from 17.3% to 3.2% of the microbiome. Our results perhaps indicate some connection with storms or winds originating from the North Atlantic Ocean (back-trajectory analyses done, but

not shown). Other *Bacteroidetes* taxa with notable presence in hailstorm microbiome included *Saprospirales* and *Chitinophagales* orders with bacteria living on animals and in the gut of animals as expected.

The microbiomes commonly found in our precipitation samples included the genus *Massilia* in significant numbers (11.3% of the microbiome), bacteria of the Proteobacterial orders *Rhizobiales*, *Sphingomonadales*, and *Burkholderiales*; a significant percentage (8.5%) of the marine genus *Marinoscillum* and bacteria in order *Saprospirales* of phylum *Bacteroidetes*. Our results suggest that no known IN active species were detected in precipitation microbiomes. The order *Pseudomonadales*, which includes most known IN active species, was found at the limit of detection.

*Massilia* and other unidentified genera of the family *Oxalobacteraceae* were also relatively dominant in all four feedlot samples with percentages from 6.5% to 65.4% of the microbiome. *Marinoscillum,* a marine bacterium surprisingly found in all precipitation samples, was also found in all feedlot samples from 3% to 8.5% of the microbiome. These similarities of the predominant bacteria in the microbiome of four feedlot dust samples and of four precipitation samples taken at an area distant from the feedlots, perhaps indicate some connection of the feedlot dust and precipitation microbiomes, either with the formation of precipitation or with their presence in aerosols during precipitation events. Although we cannot rule out the possibility that scavenging of aerosolized bacteria explains the presence of these bacteria both in feedlot and precipitation samples taken even at a distance from feedlots, our dry deposition background result shows different biological composition (**Fig. 6**). It is also noteworthy to mention that neither of the genera (*Massilia* and *Marinoscillum*) were detected in the background deposition blank sample and it is not known whether they have any IN activity. Therefore, the scavenging may not be the main reason for the presence of *Massilia* and *Marinoscillum* found in our precipitation samples. Other bacterial taxa with a significant presence in feedlot samples included members of orders *Caulobacterales* and *Burkholderiales*.

*3.5. Caveats and Future Studies*

A surface level air mass on a plain is not necessarily the same as the air mass where precipitation forms at the cloud level. Studying the vertical gradient in INP concentrations in this region would hint at the link between these two vertical zones (e.g., He et al., 2020). The future investigation should also include investigations in physicochemical transformation of hydrometers and INPs, which might occur between the cloud height and the ground (e.g., Pereira et al., 2020), impact of aerosol dynamics and processing, effect of solutes to alter the freezing point (Whale et al., 2018), secondary ice formation, and cloud macrophysics addressed in Wright and Petters (2015 - Sects. 4.1 to 4.3).

The precipitation intensity strongly depends on several other dynamical factors and thermodynamic conditions, including the land use, moisture levels, land surface temperatures, and convective available potential energy. For instance, recent observational study showed that the irrigation practices in the Great Plains region had enhanced summer precipitation intensity (Alter et al., 2015) resulting an increase in the total precipitation received. Hence, it is not straightforward to link the precipitation intensity to the estimated INP concentrations and more future studies involving cloud level and surface level INP measurements might help in elucidating this problem. To assess the impact of INPs on precipitation properties (and vice versa), it is necessary to conduct the INP measurement of cloud water samples, aerosol particle characterizations below cloud, and more detailed analysis of precipitation-forming cloud properties as well as cloud height. More detailed scavenging analysis without many assumptions and limitations, such as assuming a constant scavenging rate over precipitation, limited particle size distributions, and assuming a well-mixed boundary layer, is also necessary to connect the surface observation to cloud level phenomenon. Diffusional scavenging of small particles may not contribute to

495 IN unless they are highly ice active macromolecules or other small biological species. Regardless, robust aerosol particle size distribution data across the ground to cloud base segment would definitely complement to accurately and precisely estimate scavenging efficiencies. Some previous studies support the assumption of a well-mixed boundary layer near the study area. Further effort may be needed to characterize the climatology of boundary layer height in the West Texas region at different times of a day, as demonstrated in Schmid and Niyogi

(2012) and Zhu et al. (2001). Incorporating more local specific vertical ambient profiles (lapse rate, Dong et al., 2008) for further analysis would also be helpful.

As for more future studies, INPs derived from precipitation samples collected over multiple years would give comprehensive insight into their impact on local precipitation systems. This work highlights this need for more precipitation-based INP studies from different geographical locations. The reduced uncertainties in $n_{INP}$

along with the high INP detection sensitivity could help in addressing the long-debated issue of INP rarity at $T$s ≥ -10 °C.

## 4. Conclusion

We have successfully estimated $n_{INP}$ (per liter of air) in the immersion freezing mode from different precipitation samples collected in Canyon, TX, USA during June 2018 – July 2019. IN spectra were derived for MPC $T$ range (0 to -25 °C) from four different precipitation types (snow, thunder/hailstorm, long-lasted rain, and weak rain) using a cold-stage instrument (WT-CRAFT). Our disdrometer measurements showed a clear variation in the precipitation properties among the four different categories of precipitation samples. Severe precipitation, such

as hail/thunderstorms, had the highest rainfall intensity (mm hr$^{-1}$) and the number of precipitation particles were highest in the snow samples. We also found an increased number of large hydrometeors (> 10 mm in diameter) in both the snow and hail/thunderstorm samples. In contrast, there were no precipitation particles > 6.5 mm in diameter observed in the weak rain samples. Our PM concentration measurements implied some possibilities of wet deposition (but neglected). The IN spectra from each precipitation category in this study were compared with

the IN spectra from previous precipitation-based INP studies (Petters and Wright, 2015; Vali, 1986). We have found that $n_{INP}$ values from our precipitation samples match or exceed previously derived $n_{INP}$ from precipitation. Notably, the high $T$ (≥ -15 °C) INPs in some of our precipitation samples are in the same order of magnitude as what is reported in Vali (1986). Although we found no clear seasonal variations in $n_{INP}$ values, in part due to the limited number of samples, the analysis of yearlong ground level precipitation observations as well as INPs for

the precipitation samples showed that the highest $n_{INP}$ at -25 °C of 1,130 L$^{-1}$ coincided with a hail-involved severe thunderstorm event observed during the summer in 2018 (ID# 1). Similarly, the lowest cumulative INP at the same temperature, 3.0 INP L$^{-1}$, was found in another hail/thunderstorm samples collected in June, 2019 (ID# 37). The second lowest $n_{INP}$ (-25 °C) was found in one of our snow samples collected during the winter (ID# 23 = 3.2 INP L$^{-1}$). Overall, our results showed that cumulative $n_{INP}$ in our precipitation samples below -20 °C could be high

in the samples collected while observing > 10 mm hr$^{-1}$ precipitation with the presence of notably large hydrometeor sizes. While our results cannot conclusively define the relationship between INPs and precipitation, our precipitation INP data is an important asset for understanding ambient INPs in the West Texas region, where a rural agricultural environment prevails.

We also identified the similarity in bacterial microbiomes between our precipitation and local feedlot

dust samples. While we cannot conclude if local feedlot dust contributes to precipitation formation, we find some indications of the inclusion of agricultural dust in our precipitation samples. Regardless, we did not find the previously known bacterial INPs, such as *Pseudomonas* and *Xanthomonas* (Morris et al., 2004) in either the

precipitation or feedlot samples. To further seek a connection between local dust and precipitation, it is worthwhile to characterize the local feedlot dust in cloud water samples, as it can be the source of INPs and may impact the local hydrological cycle. Collecting long-term pollen and other biogenic aerosol particles samples and associated observational data for multiple years may add important knowledge regarding the role of local bioaerosols on precipitation INPs.

**Author Contributions**

Research design: NH, JW; Measurements: HSKV, CAR, GDM, DH, JW, NH; Analysis: HSKV, DGG, NH; Writing: HSKV, NH, DGG. GDM conducted the metagenomics investigation without knowing the identity of samples.

**Competing Interests**

The authors declare that they have no conflict of interest.

**Data Availability**

Original data created for the study will be available in a persistent repository upon publication within www.wtamu.edu.

**Acknowledgements**

The authors acknowledge the financial support by Killgore Graduate Student Research Grant (WT20-017) provided by West Texas A&M University.  This material is based upon work supported by the U.S. Department of Energy, Office of Science, Office of Biological and Environmental Research under Award Number DE-SC-0018979. We also acknowledge Drs. Gourihar Kulkarni for useful discussions regarding implications of scavenging processes on our data.

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

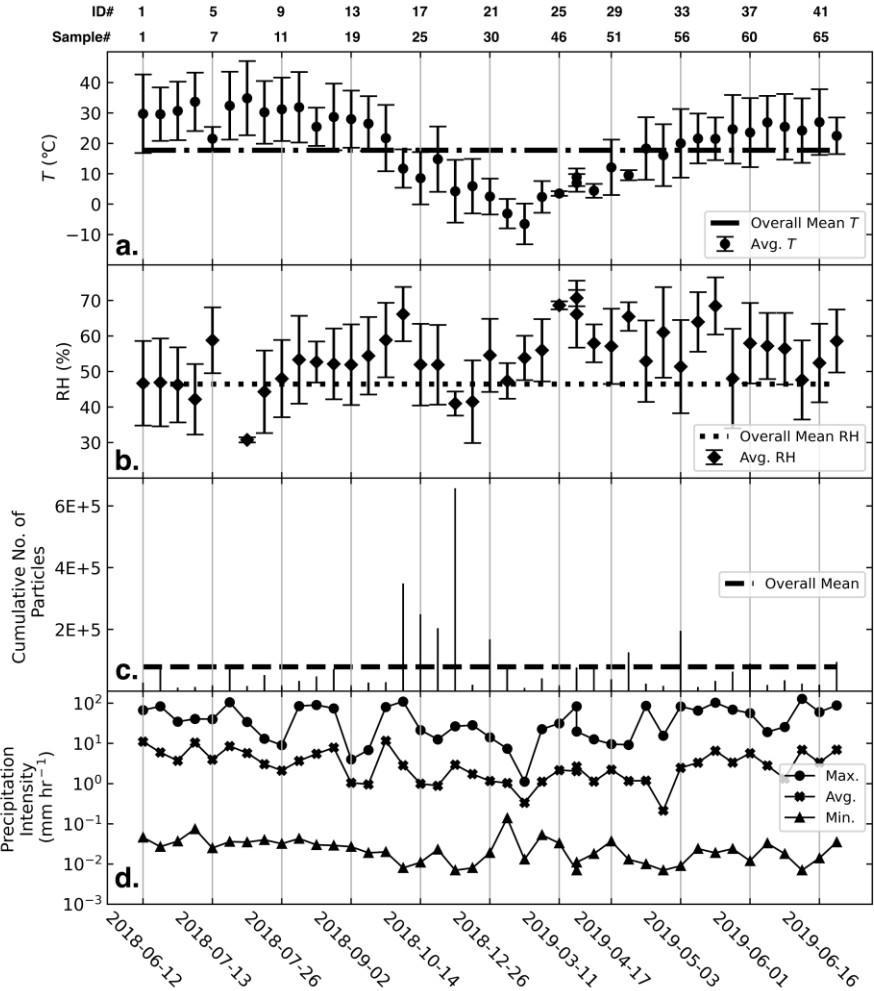

**Figure 1.**  Time series of disdrometer and IoT sensor measurements for  (a) average $T$ ± standard deviation, (b) average relative humidity ± standard deviation, (c) cumulative number of detected hydrometeors in each precipitation event, and (d) maximum, average, and minimum precipitation intensity. Each data point corresponds to the sampling start time for each precipitation event.

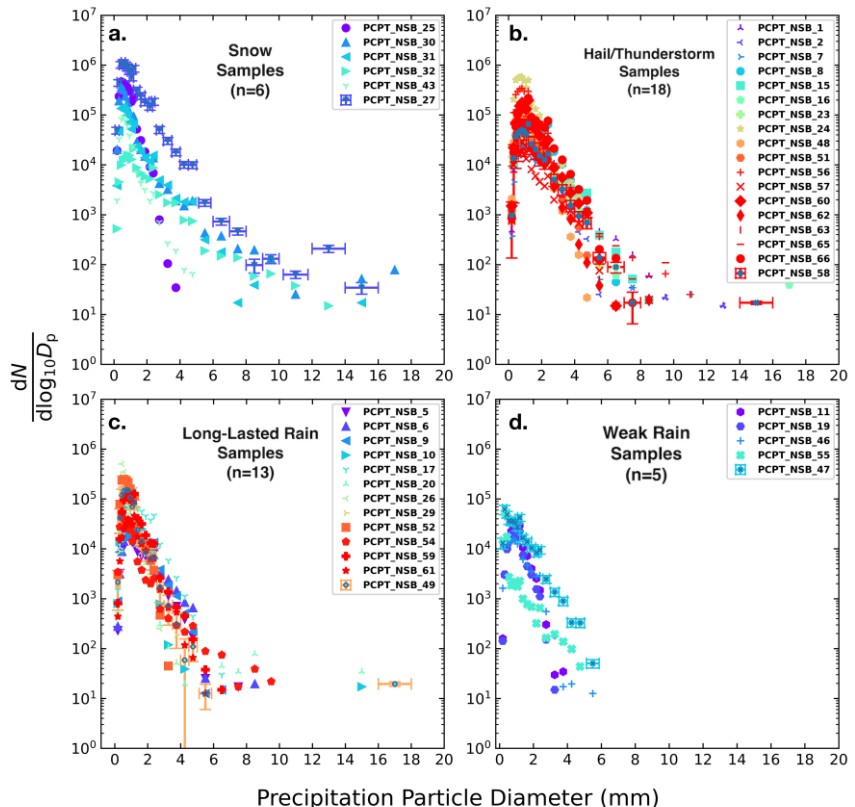

**Figure 2.** Size distribution of precipitation particles detected in (a) Snow, (b) Hail/Thunderstorm, (c) Long-lasted rain, and (d) Weak rain samples. A subset of distributions shows varying uncertainty in diameter (mm). The X-axis error bars are ±1.0 mm of size class for diameter < 2mm and ±0.5 mm of size class for diameter > 2mm. The Y-axis error bars represent standard errors at each diameter. The sub-total number of precipitation samples in each category is shown by the value of 'n'.

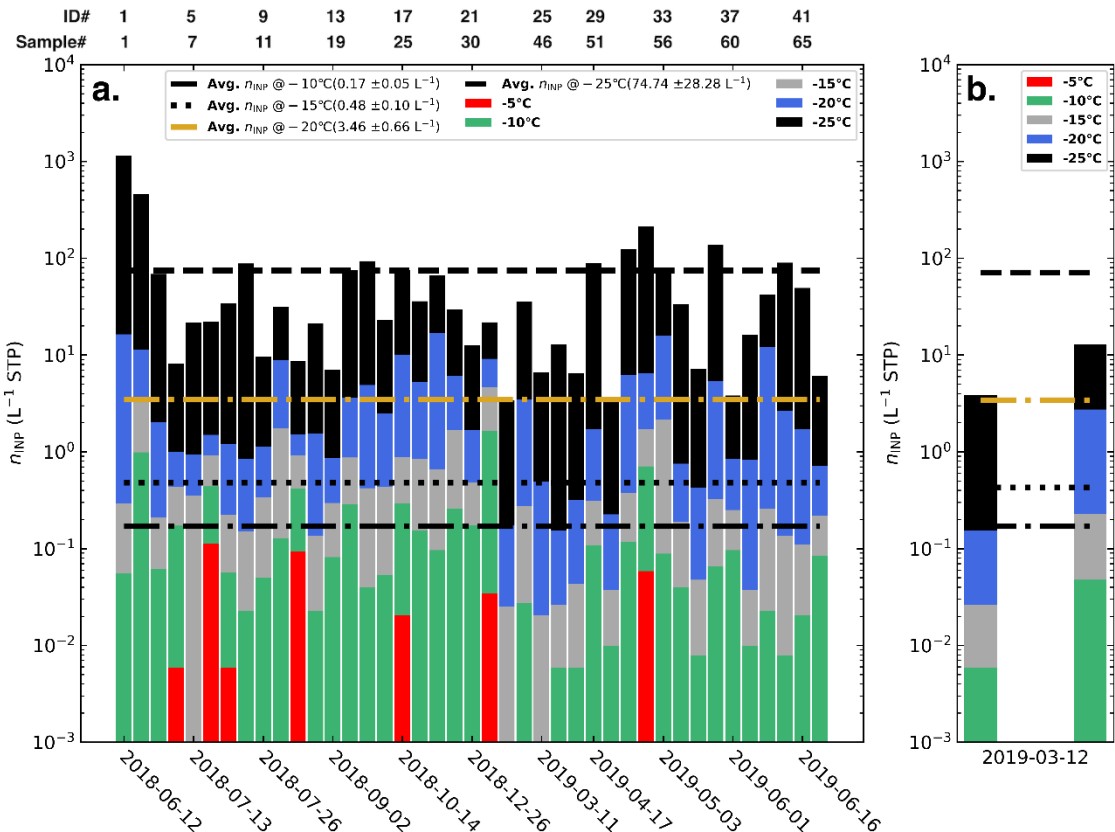

**Figure 3.** (a) Time series of cumulative $n_{INP}$ ($L^{-1}$ air) in each precipitation sample at different temperatures. (b) $n_{INP}$ for two precipitation samples (ID# 26 and 27) observed on the same day of 12 March 2019. The uncertainty in the average $n_{INP}$ at each temperature (± numbers in parentheses) is the standard error calculated for 42 samples.

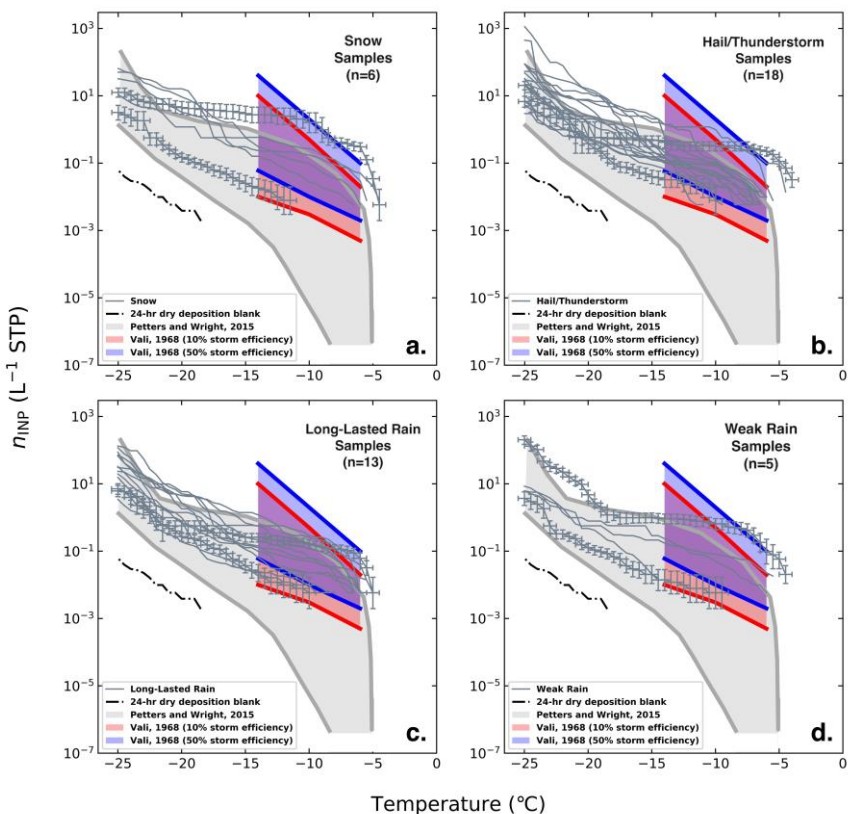

**Figure 4.** IN spectra of (a) Snow, (b), Hail/Thunderstorm, (c) Long-Lasted rain, and (d) Weak rain samples superposed on nucleation spectra from previous precipitation INP studies (shaded areas). A subset of spectra shows error bars. The X-axis error bars represent constant uncertainty of ±0.5 ˚C in temperature. The Y-axis error bars are 95% confidence interval for $n_{INP}$ shown only for two samples from each category. The number of precipitation samples in each category is shown by the value of 'n'.

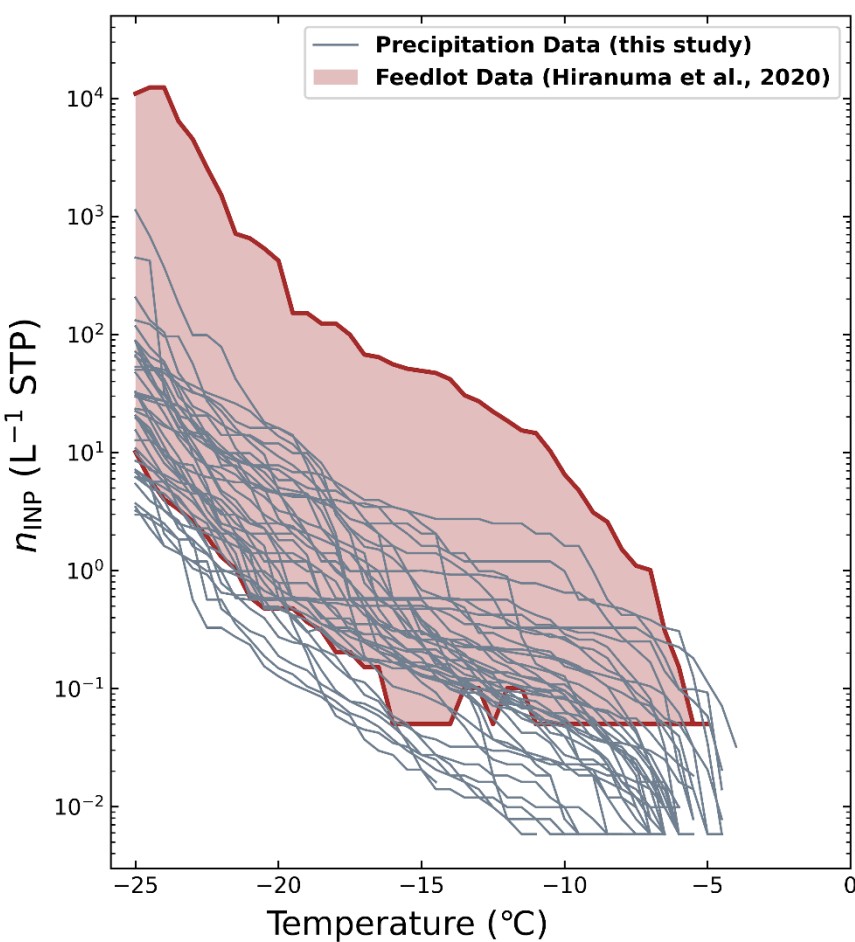

**Figure 5.** Compiled IN spectra of our precipitation samples superposed on nucleation spectra from local feedlot dust study (shaded area). The feedlot INP data are adapted from Fig. 3 of Hiranuma et al. (2020).

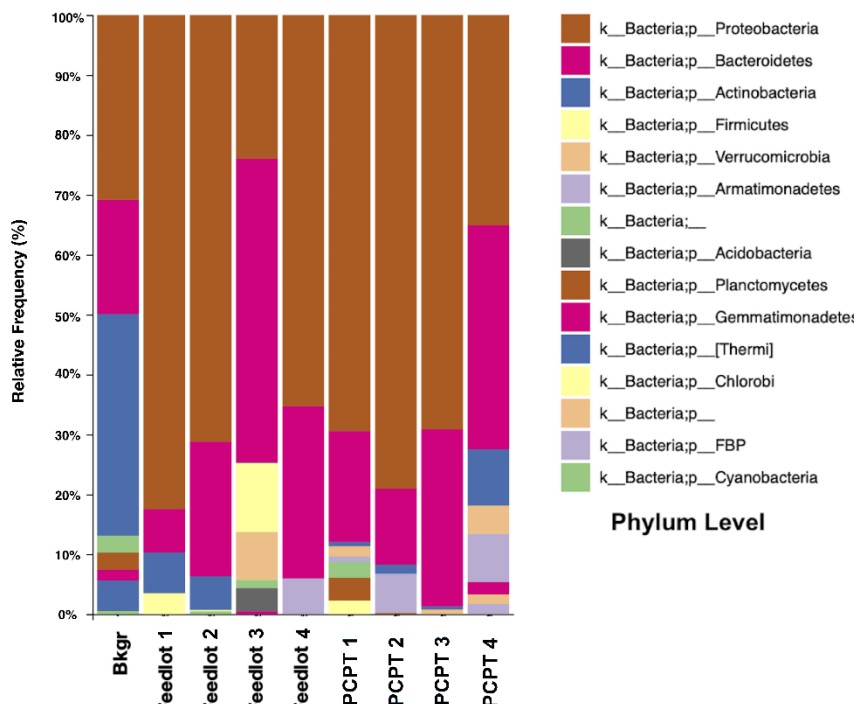

**Figure 6.** Metagenomics analysis of precipitation and feedlot dust samples showing Relative Frequency (%) or abundance of Bacterial taxonomy. 'Bkgr' represents the 24-hour dry deposition blank sample (Sample# 34). Our feedlot samples are collected locally on March 28, 2019 (1), July 22, 2018 (2), July 23, 2018 (3), and July 24, 2018 (4) – see Hiranuma et al. (2020). PCPT 1-4 corresponds to our Sample# 1, 2, 50, and 7, respectively.

**Table 1**. Adjacent hourly averaged PM values before and after each precipitation event. We excluded 14 data where PM data were not recorded due to technical issues etc. (ID# of 6-7, 17, 20, 22-24, 26, 28-33).

| ID# | Sample# | Precipitation type | $PM_1$ ($\mu g\ m^{-3}$) | | $PM_{2.5}$ ($\mu g\ m^{-3}$) | | $PM_{10}$ ($\mu g\ m^{-3}$) | |
|---|---|---|---|---|---|---|---|---|
| | | | Before | After | Before | After | Before | After |
| 1 | PCPT_NSB_1 | Hail/Thunderstorm | 1.969 | 0.111 | 4.090 | 1.693 | 6.188 | 1.990 |
| 2 | PCPT_NSB_2 | Hail/Thunderstorm | 0.010 | 0 | 1.811 | 0.001 | 2.111 | 0.001 |
| 3 | PCPT_NSB_5 | Long-Lasted Rain | 4.667 | 0.660 | 5.734 | 1.947 | 10.790 | 3.690 |
| 4 | PCPT_NSB_6 | Long-Lasted Rain | 3.755 | 3.755 | 5.956 | 5.721 | 8.867 | 8.580 |
| 5 | PCPT_NSB_7 | Hail/Thunderstorm | 0 | N/A | 0.557 | N/A | 0.723 | N/A |
| 8 | PCPT_NSB_10 | Long-Lasted Rain | 7.479 | 1.495 | 9.894 | 3.409 | 14.771 | 4.742 |
| 9 | PCPT_NSB_11 | Weak Rain | 5.760 | 3.812 | 8.165 | 6.190 | 12.770 | 9.436 |
| 10 | PCPT_NSB_15 | Hail/Thunderstorm | 14.289 | 4.020 | 16.078 | 5.124 | 30.794 | 9.277 |
| 11 | PCPT_NSB_16 | Hail/Thunderstorm | 4.913 | N/A | 5.423 | N/A | 10.534 | N/A |
| 12 | PCPT_NSB_17 | Long-Lasted Rain | 4.551 | N/A | 6.414 | N/A | 10.633 | N/A |
| 13 | PCPT_NSB_19 | Weak Rain | 0.049 | N/A | 1.283 | N/A | 6.301 | N/A |
| 14 | PCPT_NSB_20 | Long-Lasted Rain | 1.780 | N/A | 4.312 | N/A | 5.890 | N/A |
| 15 | PCPT_NSB_23 | Hail/Thunderstorm | 3.867 | 2.167 | 5.740 | 5.740 | 9.551 | 7.235 |
| 16 | PCPT_NSB_24 | Hail/Thunderstorm | 1.592 | 0 | 4.984 | 0.003 | 5.786 | 0.003 |
| 18 | PCPT_NSB_26 | Long-Lasted Rain | 0.657 | 0 | 2.830 | 0 | 3.192 | 0 |
| 19 | PCPT_NSB_27 | Snow Sample | 0 | N/A | 0.011 | N/A | 0.080 | N/A |
| 21 | PCPT_NSB_30 | Snow Sample | 0.760 | 0 | 2.627 | 0.275 | 3.180 | 0.275 |
| 25 | PCPT_NSB_46 | Weak Rain | 1.461 | 0 | 4.525 | 1.233 | 5.449 | 1.233 |
| 27 | PCPT_NSB_48 | Hail/Thunderstorm | 0 | 0 | 0.427 | 0.002 | 0.427 | 0.002 |
| 34 | PCPT_NSB_57 | Hail/Thunderstorm | 29.649 | 13.515 | 29.649 | 13.770 | 58.946 | 26.604 |
| 35 | PCPT_NSB_58 | Hail/Thunderstorm | 12.450 | 0.680 | 13.245 | 1.400 | 24.390 | 2.860 |
| 36 | PCPT_NSB_59 | Long-Lasted Rain | 10.515 | 6.912 | 11.516 | 7.918 | 21.192 | 12.892 |
| 37 | PCPT_NSB_60 | Hail/Thunderstorm | 9.740 | 3.423 | 10.661 | 4.396 | 18.750 | 7.269 |
| 38 | PCPT_NSB_61 | Long-Lasted Rain | 4.396 | 0.192 | 5.912 | 1.215 | 10.069 | 2.051 |
| 39 | PCPT_NSB_62 | Hail/Thunderstorm | 0.039 | N/A | 1.555 | N/A | 1.804 | N/A |
| 40 | PCPT_NSB_63 | Hail/Thunderstorm | 2.217 | 1.365 | 4.348 | 2.479 | 6.533 | 4.781 |
| 41 | PCPT_NSB_65 | Hail/Thunderstorm | 1.694 | 0 | 3.994 | 0.316 | 5.306 | 0.316 |
| 42 | PCPT_NSB_66 | Hail/Thunderstorm | 1.750 | 0.080 | 2.881 | 1.459 | 5.771 | 1.530 |

NOTE: N/A: either below detection limit of our PM sensor (< 0.001 $\mu g\ m^{-3}$) or sensor failure return values.

Ice-nucleating particles in precipitation samples from~~impact the severity of precipitations in~~ West Texas

Hemanth S. K. Vepuri[1,*], Cheyanne A. Rodriguez[1], Dimitrios~~Dimitri~~ G. Georgakopoulos[4], Dustin Hume[2], James Webb[2], Greg D. Mayer[3], and Naruki Hiranuma[1,*]

[1]Department of Life, Earth, and Environmental Sciences, West Texas A&M University, Canyon, TX, USA
[2]Office of Information Technology, West Texas A&M University, Canyon, TX, USA
[3]Department of Environmental Toxicology, Texas Tech University, Lubbock, TX, USA
[4]Department of Crop Science, Agricultural University of Athens, Athens, Greece

*Corresponding authors: hsvepuri1@buffs.wtamu.edu and nhiranuma@wtamu.edu

**Abstract**
Ice-nucleating particles (INPs) influence the formation of ice crystals in clouds and many types of precipitation. ~~However, our knowledge of the relationship between INPs and precipitation is still insufficient.~~ This study reports unique properties of INPs collected from ~~This study was conducted to fill this gap by assessing precipitation properties and INP concentrations ($n_{INP}$) from a total of~~ 42 precipitation samples~~events observed~~ in the Texas Panhandle region from June 2018 to July 2019. We used a cold-stage instrument called the West Texas Cryogenic Refrigerator Applied to Freezing Test system to estimate INP concentrations ($n_{INP}$) through immersion freezing in our precipitation samples with our detection capability of > 0.006 INP L$^{-1}$. A disdrometer was used for two purposes; (1) to characterize the ground level precipitation type and (2) to measure the precipitation intensity as well as size of precipitating particles at the ground level during each precipitation event. While no clear seasonal variations of $n_{INP}$ values were apparent, the analysis of yearlong ground level precipitation observation as well as INPs in the precipitation samples showed some INP variations, for example, the highest and lowest at -25 °C both in the summer for hail-involved severe thunderstorm samples (3.0 to 1,130 INP L$^{-1}$), followed by the second lowest at the same $T$ from one of our snow samples collected during the winter (3.2 INP L$^{-1}$). Furthermore, we conducted the bacteria speciation using a subset of our precipitation samples to examine the presence of known biological INPs. In parallel, we also performed metagenomics analysis of ambient dust samples collected at commercial feedlots in West Texas to check the similarity and to test if local feedlots can act as a source of bioaerosol particles and/or INPs found in the precipitation samples. Overall, our results showed that cumulative $n_{INP}$ in our precipitation samples below -20 °C could be high in the samples collected while observing > 10 mm hr$^{-1}$ precipitation with notably large hydrometeor sizes and an implication of feedlot bacteria inclusion. ~~A disdrometer was used to measure the precipitation intensity and size of precipitating particles during each precipitation event. The analysis of yearlong precipitation properties as well as INPs for the samples shed a light on the seasonal variation of the $n_{INP}$ values in West Texas. Furthermore, we characterized the bacteria speciation of the storm and ambient dust samples collected at a commercial feedlot in West Texas to identify potential biological sources of INPs in our precipitation samples. Overall, our results showed a positive correlation between $n_{INP}$ and intensity of precipitation with notably large hydrometeor sizes in storm precipitations. Amongst all observed precipitation types, the highest INPs were found in the snow samples, and hail/thunderstorm samples have the highest INPs at high temperature -5°C.~~

# 1 Introduction

## 1.1. What are INPs?

Aerosol particles play a major role in altering the cloud properties, precipitation patterns, and ultimately the Earth's radiation budget (Lohmann and Feichter, 2005). In the past few decades, the aerosol particle direct effects (i.e., the impact of aerosol particles on net radiation through scattering and absorption of solar radiation) have been extensively studied (Satheesh and Krishna Moorthy, 2005). For example, the global radiative forcing by sea salt aerosols and dust is known to be in the range of −0.5 to −2 W m$^{-2}$ and −2 to +0.5 W m$^{-2}$, respectively. However, the aerosol particle indirect effects (i.e., radiative impact due to formation of clouds) have been enigmatic. Some atmospheric aerosol particles are known to act as ice-nucleating particles (INPs) and catalyze the formation of ice crystals in the clouds, but their overall impact on the Earth's radiative budget remains quantitatively uncertain (Lohmann et al., 2007).

While INPs are sparse in the atmosphere, they have substantial impacts on the cloud microphysics and the precipitation formation (DeMott et al., 2010). The sources of atmospheric INPs are diverse as they emerge naturally and also through human activities, adding complexities to our comprehensive understanding in their impacts (e.g., Kanji et al., 2017; Zhao et al., 2019). In general, INPs provide a surface on which the water vapor and/or cloud droplet deposits and freezes (Van den Heever et al., 2006). This type of ice formation in the presence of INP is known as heterogenous freezing (Vali et al., 2015). In the absence of INPs, the formation of atmospheric ice particles follows the process of homogeneous nucleation, in which it requires the cloud droplet to be supercooled to the temperature (*T*) of -32 °C and below (depending on the pure water droplet size) to form ice crystals (Koop et al., 2000; Koop and Murray, 2016). Though our knowledge regarding INPs remains insufficient, there have been advances in understanding the different modes of heterogeneous ice nucleation (IN) in the atmosphere in the last few decades. For example, deposition nucleation is induced by the direct deposition of water vapor on to an INP's surface and ice embryo formation on the surface under ice supersaturation conditions (Kanji and Abbatt, 2006; Möhler et al., 2008). Recently, some studies have argued that the deposition nucleation could be interpreted as pore condensation and freezing (Marcolli, 2014). The presence of water in pores of mineral materials and the resulting inverse Kelvin effect cause an instantaneous water saturation condition in the confined space, allowing the water to freeze even at water sub-saturated ambient conditions (David et al., 2019; Marcolli, 2014). Amongst various IN paths, perhaps the most important mode is immersion freezing (De Boer et al., 2010). This process starts with the formation of cloud droplet followed by freezing due to an INP immersed in the supercooled droplet. In addition, the past studies have identified other modes of heterogeneous nucleation, such as condensation freezing (Belosi and Santachiara, 2019), contact freezing (Hoffmann et al., 2013), and inside-out evaporation freezing (Durant and Shaw, 2005). These modes are relatively less relevant in the mixed-phase clouds (MPCs) as discussed in the next section.

## 1.2. Importance of Immersion Freezing

INPs greatly influence cloud properties, especially in MPCs, which are typically observed in the altitude range of 2 km to 9 km above ground level (Hartmann et al., 1992). Out of all heterogeneous ice-nucleation modes, the immersion freezing is the most dominant mode of ice formation in MPCs (Ansmann et al., 2008; De Boer et al.,

2010; Hande and Hoose, 2017; Vergara-Temprado et al., 2018). In Hande and Hoose (2017), different cloud types such as orographic, stratiform, and deep-convective systems were simulated and analyzed for different freezing modes under various polluted conditions. The authors demonstrate that immersion freezing is the predominant IN mode under various simulated circumstances, accounting for 85 to 99%, while other IN paths play a less significant role. Cui et al. (2006) also showed that immersion freezing is the primary mode of ice formation with little significance of the deposition mode in the early stages of the cloud development. Moreover, whereas contact freezing may be a highly efficient ice formation path, a previous simulation study showed that it is a negligible mode in the given MPC conditions (Phillips et al., 2007). Field et al. (2012) and De Boer et al. (2011) showed that the formation of cloud droplets is a precondition for ice formation in MPCs, thus highlighting the importance of immersion nucleation. Furthermore, using multiple ground-based instruments, including Lidar, AERONET Sun Photometer, and Vaisala Radiosonde, Ansmann et al. (2008) found that a high INP concentration ($n_{INP}$) (i.e., ~ 1 – 20 cm$^{-3}$) in the Saharan dust. This high dust-including $n_{INP}$ episode coincided with the presence of liquid droplets at cloud tops at $T$s of -22 °C to -25 °C. Similarly, Ansmann et al. (2009) shows the observation of tropical altocumulus clouds having liquid cloud tops. Due to the importance and dominance of immersion freezing, the current study focuses on measuring the immersion freezing efficiency of the precipitation samples collected in the Texas Panhandle region.

### 1.3. INPs ~~and Atmospheric~~in Precipitation

It is known that INPs in MPCs have a notable impact on the properties of precipitation. Previously, Yang et al. (2019) studied the effect of INPs on cloud dynamics and precipitation through model simulations of an observed severe storm in Northern China. The authors show that an increase in INPs can enhance the storm, whereas an excessive increase of INPs may impede the updrafts in the storm. The reason for this complex effect of INPs may be explained by the variation in the latent heat release in the convective system at different stages of its development. This latent heat is further influenced by INP episode, thus affecting the dynamics of the precipitation system. Furthermore, the increase in INP number might reduce the mean hail diameter (hail particles with smaller diameters melt more easily), which leads to decreased hail precipitation and an increased rain formation in contrast to the previous studies (Fan et al., 2017; Van den Heever et al., 2006). Similar results have been found by Chen et al. (2019). The authors show that an increased $n_{INP}$ in the simulated hailstorm can reduce the graupel size and ~~reduce~~ the concentration of hail stones. Likewise, the aircraft observations along with the model simulations of convective storms in West Texas and U.S. High Plains have shown that the addition of INPs at the base of warm clouds would result in an increase of the precipitation amount by strong updrafts in the system (Rosenfeld et al., 2008), ultimately affecting the local hydrological cycle (Mülmenstädt et al., 2015). It has also been observed that INPs can be removed from the atmosphere through precipitation resulting in a net decrease in $n_{INP}$, affecting the precipitation development (Stopelli et al., 2015). ~~The estimation of $n_{INP}$ in this study from the precipitation samples gives a quantitative approximation of INPs in the locally observed weather systems, potentially allowing us to parameterize the INP-precipitation relationship.~~

Several previous studies have characterized the $n_{INP}$ in the precipitation samples from various locations (Creamean et al., 2019; Petters and Wright, 2015; Levin et al., 2019). Petters and Wright (2015) reported a wide range of $n_{INP}$ values in their local precipitation samples collected approximately 3 km west of Raleigh, NC, USA for

July 2012 and October 2013. Their study shows a variation of 10 orders of magnitude in the concentrations of INPs with a high variability in the *T* range of -5 ~~°C~~ to -12 °C, suggesting inclusion of biological INPs, which are generally known to be active at relatively high freezing *T*s (Després et al., 2012). The lower limit for the INP spectrum as a function of *T* derived from the cloud water and precipitation samples in Petters and Wright (2015) may highlight the extreme rarity of INPs at *T*s warmer than -10 °C. Particularly, the authors showed that the highest ever observed $n_{INP}$ at -8 °C were three orders of magnitude lower than observed ice crystal concentrations in tropical cumuli at the same temperature. More precipitation studies may provide a constraint on minimum enhancement factors for secondary ice formation processes. In Levin et al. (2019) the $n_{INP}$ values during an atmospheric river event on the west coast of United States were studied. The authors found an increased concentration of marine INPs in contrast to their previous studies, showing high mineral/soil dust during an atmospheric river precipitation. ~~However, the relation between INPs and the physical properties of precipitation particles as well as the variation in severity of the precipitation is still uncertain, representing a knowledge gap regarding precipitation INPs. This study narrows this gap by investigating the role of INPs in different precipitation systems.~~

### *1.4. Study Objectives*

In this study, we characterized properties of INPs in~~calculated the $n_{INP}$ in~~ precipitation samples collected in the Texas Panhandle region. All of our samples were analyzed at our laboratory using a cold stage instrument. The estimated $n_{INP}$ in the precipitation samples were studied~~compared~~ with ground level precipitation properties, such as the precipitation type, intensity of precipitation (mm hr$^{-1}$), and hydrometeor particle size (mm). ~~In addition, the seasonal variation of $n_{INP}$ in the Texas Panhandle region was studied and compared with the particulate matter (PM) concentrations measured by our Internet of Things (IoT) sensors.~~ A subset of the collected precipitation samples was analyzed for their bio-speciation to characterize potential biological INP sources in the West Texas region and also to examine the presence of ~~investigate if the biological composition matches with any previously~~ known high *T* biological INPs. Although the estimation of $n_{INP}$ in precipitation samples collected at the ground level does not represent INPs at the cloud height, we report the INPs resolved by the ground level weather observation that help understanding of ambient INPs in the West Texas region, where unique and substantial INPs, i.e., several hundred and thousand INPs L$^{-1}$ at -20 °C and -25 °C, respectively, are consistently emitted from animal feeding operations (Hiranuma et al., 2020).

### **2 Methods**

### *2.1 Precipitation Sampling*

~~In this study, t~~The precipitation samples were collected from different seasons throughout the year during June 2018 – July 2019. Sterilized polypropylene tubes of 50 ml volume~~mL~~ (VWR® Centrifuge Tube) were used as sampling gauges. The gauges were placed at ~50 ft above the ground on the rooftop of the Natural Science Building at West Texas A&M University, Canyon, TX. This particular location was chosen to avoid any obstruction of our sampling activities. The sampling tubes were well exposed to the ambient air without any canopies throughout the sampling process. The sampling gauges were replaced every 24 hours to minimize the

155 effect of dry deposition prior to the precipitation sample collection. A blank dry deposition sample (Sample# 34) was specifically collected for 24 hours from January 2-3, 2019 in order to examine and quantify the effect of dry deposition on $n_{INP}$. The freezing spectrum of this dry deposition sample (suspended in HPLC grade pure water) was later compared with the IN spectra of precipitation samples (see **Sect. 3.3**.1). We note that a volume of pure water (5 ml) for an atmospheric INP estimate based on a dry deposition sample was determined by
160 averaging collected precipitation volumes of all samples prior to this dry deposition sample. For the duration of a given precipitation episode, some amount of sample was accumulated in the tube. The sampling tubes were then capped and stored at $T$ of 4 °C in the refrigerator, following the method described in Petters and Wright (2015), until the droplet-freezing assay experiments were commenced. The effect of storage conditions on the IN activity was not considered in this study. We note that Beall et al. (2020) recently found a decrease
in precipitation $n_{INP}$ by 42% when stored at 4 °C (i.e., Table 5) and suggested correction factors for the $T$ range of -7 °C to -17 °C. After the freezing experiment, a subset of our samples was kept under deep-freeze conditions (-80 °C) for further biological analysis (see **Sect. 2.6**). In total, 42 precipitation samples were collected from different weather systems observed at the surface level. Based on these samples and observationsIn this study, we estimated the $n_{INP}$ values from (1) snows, (2) hails/thunderstorms, (3) long-lasted rains, and (4) weak rains.
More information about the samples used in this study, precipitation types and the amount of the precipitation collected for each sample are provided in the **Supplemental Information (SI) Sect. Table S1** 1.

## 2.2. Disdrometer Measurements of Precipitation Properties

For our precipitation measurements, we used the OTT Parsivel[2] (Particle Size Velocity 2) sensor. This device is a modern laser-optical disdrometer ($\lambda$ = 780 nm) which measures the size and fall velocity of precipitating
particles. The OTT Parsivel[2] was deployed in side-by-side position with the precipitation gauge collector for the duration of our study period. A detailed technical description of OTT Parsivel[2] is given in a previous study (Tokay et al., 2014), so only a brief description is provided here. A combination of the laser transmitter and receiver component was integrated as a single cluster in a weatherproof housing and detects precipitation particles passing through a horizontal strip of light. A nominal cross section area of a laser beam detection was 54 cm[2],
and the system recorded the number of hydrometeors in a 32 x 32 matrix (i.e., fall velocity x diameter) in the ≥ 30 seconds time resolution. The measurable size range of hydrometeor particles was 0.062 - 24.5 mm in diameter ($D_p$) with bin size intervals ($\Delta D_p$) varying from 0.125 to 3.0 mm. Our disdrometer was coupled with an OTT netDL Hydrosystem logger (40 channels). The OTT Parsivel[2] also measured the intensity of precipitation (mm hr[-1]) and the number of precipitation particles passing through the horizontal strip of light in the event of
precipitation. The OTT Parsivel[2] automatically categorized the precipitation type according to the National Weather Service (NWS) weather code based on the measured precipitation properties. Due to the intermittent nature of the precipitation, the OTT Parsivel[2] assigned multiple NWS precipitation codes during a single precipitation event (**Table S1** 1 column 'NWS Code'). We compared our manual observations with the NWS precipitation code assigned by the disdrometer, and we categorized all observed precipitations into four
different types. These four major precipitation types defined in this study included snow, hail/thunderstorm, long-lasted rain, and weak rain, and we collected 6, 18, 13, and 5 samples from each type, respectively, which sum up to a total of 42 samples. More detailed methodology of precipitation categorization is discussed in **SI Sect. S1.1**.

## 2.3 IoT Air Quality Sensor Measurements

A cluster of Arduino-based Internet of Things (IoT) air quality sensors was developed to measure ambient air conditions at our precipitation sampling location. This IoT cluster was deployed alongside the disdrometer and sampling gauge to complement this study. A DFRobot particulate matter (PM) laser dust sensor measured PM with size ranges of < 1 μm ($PM_{1.0}$), < 2.5 μm ($PM_{2.5}$), and < 10 μm ($PM_{10}$) with an estimated uncertainty of ±27% relative to an optical particle counter (Markowiz and Chiliński, 2020). Other ambient conditions, including $T$, barometric pressure, and humidity, were measured with a precision Bosch BME280 environmental sensor. We calibrated our sensors against a commercially available sensor (GlobalSat Inc., LS-113). Our sensors utilized Long Range and Wide Area Network (LoRaWAN) technology for data transmission. A LoRaWAN transceiver is connected to our sensors for wireless data transmission. This small IoT device operated with 915 MHz signal frequency, transmitting encrypted and signed packets of captured air quality data through a hosted LoRa network server to a Kibana visualization server. This data interface enabled in situ monitoring and processing of the data. The PM concentrations were later time-averaged for ~~comparison with the precipitation properties and~~ assessing contribution of wet scavenging of aerosol particles to $n_{INP}$ in the precipitation samples.

## 2.4 Immersion Freezing Experiment

All immersion freezing experiments in this study were conducted using an offline instrument called West Texas - Cryogenic Refrigerator Applied to Freezing Test (WT-CRAFT) system (Hiranuma et al., 2019; Cory et al., 2019~~; Rodriguez et al., 2020~~). The WT-CRAFT system is a cold-stage technique, in which the droplets are placed on an aluminum plate and cooled until they are frozen. A commercially available digital camera was used to record the droplet freezing events, and we visually evaluated the freezing $T$s based on the shift in droplet brightness while freezing. If there was an uncertainty in determining the $T$ at which a droplet was completely frozen, we used the ImageJ software for further image analysis of those droplets (see Table S4 in Hiranuma et al., 2019). This system was used to obtain $T$-resolved $n_{INP}$ in -25 °C < $T$ < 0 °C. The lower $T$ limit was -25 °C to ensure measuring INPs with negligible artefacts (Hiranuma et al., 2019). Our system is susceptible to low INP detection, and the minimum INP detection limit of the WT-CRAFT system for this study was 0.~~002~~006 $L^{-1}$ air. To minimize any contamination during the IN measurement, the WT-CRAFT system was placed in a ventilated fume hood. For each experiment an aluminum plate surface was freshly coated with a thin layer of thermally conductive and IN-inert Vaseline to physically isolate individual droplets from the aluminum surface (otherwise, aluminum can act as a heterogeneous IN surface). A total of 70 suspension droplets of 3μL volume each were prepared for each run. The aluminum plate with the droplets on it was then placed inside a portable cryogenic refrigerator (Cryo-Porter). Freezing $T$s were measured by the sensor taped on the aluminum surface with a resolution of 0.1 °C, and the external keypad controller was used to control cooling rate (°C $min^{-1}$). In this study, the freezing experiments were carried out at a cooling rate of 1 °C $min^{-1}$. The validity of using this cooling rate and another test regarding time trial aspect are demonstrated in **SI Sect.** **S2** (**Figs. S2-1 and S2-2**). The droplets were cooled until all 70 droplets were frozen before warming up the system to 5 °C to be prepared for a subsequent experiment.

If all the droplets were frozen at $T$ > -25 °C, a HPLC-grade ultrapure water was used to prepare different

serial dilutions for the precipitation samples. The diluted suspensions were made to compute the $n_{INP}$ down to -25 °C. Some of our precipitation samples were diluted until the frozen fraction (the ratio of number of droplets frozen to the total number of droplets) curve was conformed to the background curve (i.e., frozen fraction curve for the HPLC ultrapure water). At the end of each WT-CRAFT experiment, the frozen fraction and ambient $n_{INP}$ were estimated as a function of $T$ with an interval of 0.5 °C. The IN measurements from the undiluted and diluted runs were merged by taking the lower $n_{INP}$ values, which typically possess the lowest confidence intervals, for the overlapped $T$ region.

The total systematic $T$ and $n_{INP}$ uncertaintiesy in our experimentsWT-CRAFT for this study arewas ± 0.5 °C and ±23.5% (Hiranuma et al., 2019). For this study, tThe experimental uncertainty in our estimated $n_{INP}$ were estimated was evaluated and reported using the 95% confidence interval method described in Schiebel (2017). Background contamination tests for WT-CRAFT were carried out weekly to make sure negligible background freezing at -25 °C. In this study, we consider the frozen fraction ≤ 0.05, accounting for less than 3% of pure water activation, as negligible background (Hiranuma et al., 2019). For these background tests, only HPLC grade ultrapure water was used for preparing the droplets.

*2.5 IN Parameterization*

Here we describe the parameterization used to estimate ambient $n_{INP}$. Initially, we computed the $C_{INP}(T)$ value, which is the nucleus concentration in precipitation suspension ($L^{-1}$ water) at a given $T$ as described in Vali (1971). This $C_{INP}(T)$ value was calculated as a function of unfrozen fraction, $f_{unfrozen}(T)$ (i.e., the ratio of number of droplets unfrozen to the total number of droplets) as:

$$C_{INP}(T) = -\frac{\ln\left(f_{unfrozen}(T)\right)}{V_d} \tag{1}$$

in which, $V_d$ is the volume of the droplet (3 $\mu$L).

Next, we used the cloud water content (CWC) parameter in order to convert $C_{INP}(T)$ to $n_{INP}(T)$, INP in the unit volume of atmospheric air at standard $T$ and pressure (STP) conditions, which is 273.15 K and 1013 mbar. We assumed CWC to be a constant of 0.4 g $m^{-3}$, following Petters and Wright (2015). This assumption would be reasonable for the following three reasons: (1) Petters and Wright (2015) and references therein showed typical values of CWC for different cloud types could narrowly range from 0.2 g $m^{-3}$ to a factor of few more, (2) the authors also showed that the variation of $n_{INP}$ with CWC values for different cloud types in the atmosphere would typically be limited within a factor of two, and our $n_{INP}$ uncertainties could be larger than that, and (3) based on a parametrization for rainwater evaporation, Zhang et al. (2006) suggests that evaporation does not contribute to $n_{INP}$ bias for both strong convective systems and persistent rain events with cloud base heights of ≈3 km. Thus, the variation of CWC on the $n_{INP}$ was considered to be negligible. Nonetheless, it is necessary in the future to further investigate in cloud specific CWCs incorporating with loss of water through partial evaporation of raindrops during free fall based on vertical vapor deficit profiles to conclusively assess if this assumption is fair or not. Precipitation evaporation rate might introduce bias in $n_{INP}$

for precipitation systems with high cloud base, and the correction can be applied accordingly (Petters and Wright, 2015). Direct comparison between INP measurements in cloud water samples and those in precipitation samples might also be key to answer this question (e.g., Pereira et al., 2020). ~~We presumed CWC to be a constant of 0.4 g m⁻³, covering the continental clouds in our study. Our assumption would be reasonable since Petters and Wright (2015) showed that the variation of $n_{INP}$ with CWC values for different cloud types in the atmosphere would typically be limited within a factor of two, and our $n_{INP}$ uncertainties could be larger than that. Thus, the effect of CWC on the $n_{INP}$ would be negligible.~~

The sample air volume ($V_{air}$) at the cloud level was calculated by converting the volume of the precipitation sample collected ($V_l$) using the Eqn. (2) from Petters and Wright~~,~~ (2015):

$$V_{air} = \frac{V_l \times 1000 \times \rho_w}{CWC}$$

(2)

where $\rho_w$ is a unit density of water (1 g ml⁻¹). $V_{air}$ is in liters (L), whereas $V_l$ is given in ml. The multiplication factor '1000' is used to convert the volume from cubic meter (m³) of air to liter of air. The cumulative $n_{INP}$ per unit volume of sample air, described in the previous study DeMott et al. (2017), was then estimated as:

$$n_{INP}(T) = C_{INP}(T) \times DF~~(DF)~~ \times \frac{V_l}{V_{air}}$$

(3)

where DF is a serial dilution factor (e.g., DF = 1 or 10 or 100 and so on).

## 2.6. Microbiome of Feedlot Dust and Precipitation Samples

The overall goal of our metagenomics analysis was to identify known ice-nucleation-active bacterial and fungal species in feedlot dust and precipitation samples collected in the West Texas region. This biological speciation is also useful to examine if local feedlots can act as a source of bioaerosol particles and/or INPs found in the precipitation samples. In this study, we have examined a heterogeneous set of samples including four feedlot samples locally collected on March 28, 2019 and July 22, 23, and 24, 2018 (see Table 1 of Hiranuma et al., 2020), precipitation samples ~~hail, long-lasted rain~~ (Sample# 1, 2, 7, and 50), and a 24-hour dry deposition sample (Sample# 34). We note that the precipitation Sample# 50 (another hail/thunderstorm sample), which was collected on March 23, 2019 when a tornado warming was issued, was preserved only for metagenomics due to its low volume (≈ 1ml). It is also noteworthy that we attempted to analyze samples of all precipitation types, but acquired quantitative results only for those hail/thunderstorm samples (the reason is unknown). Next, we describe our microbiome analysis procedure in four different steps, including (1) DNA Extraction, (2) 16S rRNA Amplicon Diversity Sequencing, (3) Bioinformatics, and (4) Data Analysis. For DNA extraction, ~~Genomic~~ genomic DNA was first extracted from all samples using PowerSoil DNA Isolation Kits (MoBio Laboratories, Inc., Carlsbad, CA, USA). Extraction proceeded following the manufacturer's protocol, with the following minor changes: solutions C1 and C6 were heated to 65 °C and solution C6 was allowed to remain on the filter membrane for at least one minute before centrifugation. Additionally, the C6 step was repeated. Library preparation for bacterial

16S DNA amplicon sequencing utilized primers for the V1-V3 hypervariable region of the 16S gene. These primers were constructed for the 16S amplicon using a combination of the 28F and Illumina i5 sequencing primer and the Illumina i7 sequencing primer with the 519R primer. Amplifications were performed in 25 µl reactions with Qiagen HotStar Taq master mix (Qiagen Inc, Valencia, CA, USA). Reactions were performed with 1 µl of each 5µM primer and the template DNA. Amplification was performed on an ABI Veriti thermocycler (Applied Biosytems, Carlsbad, CA, USA) under the following thermal profile: 95 °C for 5 min, then 25 cycles of 94 °C for 30 sec, 54 °C for 40 sec, 72 °C for 1 min, followed by one cycle of 72 °C for 10 min and 4 °C hold. An ethidium bromide-stained gel was used to qualitatively determine the amount of the amplification product to add to the second amplification stage. Primers for the second PCR were designed based on the Illumina Nextera PCR primers. The second stage amplification proceeded using the same cycling protocol as the first round, except it was amplified for only 10 cycles. SPRIselect beads (BeckmanCoulter, Indianapolis, IN, USA) were used at a 0.7 ratio to size-select the DNA amplicons from an equimolar pooled sample. Pooled samples were then quantified using a Qubit 2.0 fluorometer (Life Technologies) and loaded on an Illumina MiSeq (Illumina, Inc. San Diego, CA, USA) 2x300 flow cell at 10pM.

For bioinformatics, raw data were initially processed using a standard microbial diversity analysis pipeline (QIIME2-2020). Raw data was first checked for sequencing quality and chimeric sequences, before being parsed through a microbial diversity pipeline. During the cleanup stage; denoising of the raw data was performed using various techniques to remove short sequences, singleton sequences, and reads with poor quality scores. Next, chimera detection software was used to filter out any potentially chimeric sequences. Finally, remaining high-quality sequences were corrected base by base to check for sequencer miscalls. The diversity analysis pipeline clustered all sequences based on 97% similarity to yield operational taxonomic units (OTUs), before running a seed sequence from each OTU through a taxonomic database curated in-house by RTLGenomics. Finally, the taxonomy was assigned to each sequence using a classifier that was pretrained on GreenGenes database with 99% OTUs. The relative abundance of bacterial taxa within each sediment sample was determined by dividing each OTU by the total number of reads. ~~Alpha diversity was carried out by taking phylogenetic distances into account and by looking at how diverse the phylogenetic tree is for each sample. Next, beta diversities were analyzed using weighted (by bacterial abundance) or unweighted Unifrac distances calculated from a mid-point rooted tree. Multivariate differences in beta diversity were analyzed using Permutational Multivariate Analysis of Variance Using Distance Matrices function (ADONIS), which uses an ANOVA-like simulation to test for sampling location differences (McMurdie and Holmes 2013).~~

## 3 Results and Discussion

### 3.1 Ambient and Precipitation Properties

The time series summary of ambient and precipitation properties measured by our disdrometer as well as IoT cluster~~, respectively,~~ is shown in **Fig. 1**. Each data point in **Fig. 1a** shows the average temperature measured over the sampling period of a given precipitation event. A notable seasonal variation of ambient $T$ at our sampling location was observed. ~~All the individual $T$ and relative humidity data points shown in **Fig. 1a and 1b** correspond to the sampling start date of each precipitation event. Each data point in **Fig. 1a** show the average temperature measured over the sampling period of a given precipitation event. A notable seasonal variation of~~

340 ~~atmospheric *T* was observed.~~ The highest average temperature measured during a precipitation event was 34.9~~°C~~ ± 12.2 °C, which was in the summer of 2018 (i.e., ~~16 July;~~ ID# 7; a long-lasted rain sample), while the lowest *T* was -6.5 ~~°C~~ ± 6.7 °C, measured during the winter of 2018 (i.e., ~~28 Dec;~~ ID# 23; a snow sample). The annual mean *T* for Canyon, TX region measured at our sampling site was 17.7 °C. ~~The details of each~~

345 ~~precipitation event and its properties are shown in the **SI Table S1-1 and S1-2**.~~ The diurnal cycles of ambient properties are not shown in **Fig. 1a**. Nevertheless, we typically observed suppression of *T* before precipitation events in our study. It ~~has been understood~~ is known that the *T* gradient plays a major role in the development and growth of the precipitation systems (Vaid and Liang 2015). Next, each relative humidity data point shown in **Fig. 1b** corresponds to the average during each precipitation event. ~~The relative humidity shown in **Fig. 1b** was the averaged value for each precipitation sampling period.~~ With an overall average of 54.0%, ~~t~~The highest

and lowest relative humidity values measured were 70.7~~%~~ ± 2.3 % (~~on 12 March 2019;~~ ID# 26; a weak rain sample) and 30.8~~%~~ ± 0.7 % (~~on 16 July 2018;~~ ID# 7; a long-lasted rain sample). The observed low ground level relative humidities during some precipitation events (**Tables S1 - S2**) may be a concern as loss of water through partial evaporation of hydrometeors during free fall. But, it is noteworthy that the water evaporation might have negligible effect on $n_{INP}$ estimated from precipitation samples as discussed in **Sect. 2.5**. Third, **Fig.~~ure~~ 1c**

displays the time series of the cumulative number of detected precipitation particles in individual precipitation events and the overall mean number of detected particles (dashed line). In our study period, a disdrometer detected a substantial number of precipitation particles with a cumulative number ranging from $1.0 \times 10^4$ to $6.6 \times 10^5$ particles passing through its laser beam cross section per event. More details of each precipitation event and its properties are shown in the **Tables S1 - S3**. ~~In our study period, the precipitation events during~~

~~September 2018 – January 2019 exhibited a substantial number of precipitation particles with a cumulative number of 2E+05 to 6.6E+05 per event.~~ As seen in **Table S3**, ~~This~~ high number~~s~~ of precipitation particles ~~, greater than the overall mean cumulative number (i.e., 7.9E+04) was~~ were observed in conjunction with snow/hail-involving precipitation events during ~~this~~ our study period, which may increase the wet scavenging efficiency of ambient aerosol particles during precipitation (see **Sect. 3.2 and SI Sect. S4**). Out of all the 42 samples, the

highest number of precipitation particles was detected ~~were 6.6E+05 (~~on the 5[th] of Nov, 2018~~;~~ (ID# 19; a snow sample), while the lowest was observed on ~~1.0E+04 (~~the 2[nd] of Sep, 2018~~;~~ (ID# 13; weak rain). ~~Additional information is detailed in **SI Table S1-2 and S1-3**. There were other occasional snow/hail precipitation events in our study period, but the frequency of their occurrence was indeed high in Fall - Winter. Overall, a high average number of precipitation particles were detected for all the snow samples combined, 2E+05 ± 2E+02,~~

~~followed by hail/thunderstorm samples, 7.1E+04 ± 1.9E+04. On the other hand, the weak rain episode had lowest average number of precipitation particles, 1.8E+04 ± 5.4E+03 (**SI Table S1-3**).~~ Finally, **Fig.~~ure~~ 1d** shows the average, maximum, and minimum precipitation intensity (mm hr$^{-1}$) measured during each precipitation event. Due to the intermittent nature of the precipitation, the intensity widely ranged from 1.1 to 129.3 ~~0 to 150~~ mm hr$^{-1}$ per event. ~~The measured lower values of the average intensity were due to the influence of low~~

~~intensities observed over a prolonged period of a given precipitation event.~~ The highest maximum intensity of 129.3 mm hr$^{-1}$ was measured during a hail/thunderstorm event (ID# 40), while the lowest was 1.1 mm hr$^{-1}$ during a snow event (ID# 23). These intensity data were used for our wet deposition analysis (**SI Sect. S4**). ~~The average intensity ± standard error for each precipitation category is shown in the **SI Table S1-3**. Hail/thunderstorm events have recorded the highest average precipitation intensity of 5.3 ± 7E-01 mm hr$^{-1}$,~~

~~which was greater than the average intensity measured for the weak rain episodes, 1.5 ± 3.8E-01 mm hr⁻¹ by a~~ ~~factor of 3 (**SI Table S1-3**).~~

The variation of precipitation properties was further investigated by analyzing the size distribution of precipitation particles measured by the OTT Parsivel$^2$ disdrometer. **Figure 2** shows the precipitation ~~log-normal~~

385 particle size~~diameter~~ distribution for each category of ground level observed precipitation type~~system~~. ~~These size distributions were computed from the size-resolved precipitation particle measurements by the OTT Parsivel² disdrometer during each precipitation event.~~ The size of precipitation particles was represented at the mid-value~~median diameter~~ of the corresponding disdrometer's size bin. As shown in the **Fig. 2a and 2b**, both ~~the~~ snow and hail/thunderstorm samples had particles of diameter greater than 10 mm with~~, and~~ the maximum particle diameter of~~was~~ 17 mm. Although there are three episodes of long-lasted rain with a particle diameter

greater than 14 mm (**Fig. 2c**), a clear trend of overall decrease in the hydrometeor size was seen for this category as well as the weak rain samples (**Fig. 2d**). ~~Even though the number of samples in each precipitation category was different, 66.7% (4 samples) of the total snow samples (n=6) had precipitation particles of diameter ≥ 8.5 mm. Less compared to snow, but 55.6% (10 samples) and 46% (6 samples) of the hail/thunderstorms (n=18) and long-lasted rain (n=13) samples, respectively, had recorded precipitation particles of diameter ≥ 8.5 mm.~~

~~In contrast, none of the weak rain samples (n=5) had hydrometeors of diameter ≥ 8.5 mm, and~~ In fact, all weak rain samples contained particles only smaller than 6.5 mm. Moreover, the mode precipitation particle diameter for the snow, hail/thunderstorm, and long-lasted rain samples was 0.44 mm, whereas it was 0.31 mm for the weak rain samples (see **SI Table S3~~1-3~~**). This variation in mode diameter along with the results shown in **Fig. 2** generally exhibited the shift in hydrometeor particle size distribution towards a ~~higher~~ larger diameter with an

increased intensity of precipitation at the ground level. ~~Further discussion regarding the variation of $n_{INP}$ with the severity of precipitation was analyzed and is followed in **Section 3.3**.~~

*3.2 IoT Air Quality Sensor Results and Implication of Wet Deposition*

The overall mean PM concentrations (± standard error) measured by an IoT air quality sensor for our study

period were 3.9 ± 9.2 x 10⁻² $\mu$g m⁻³ (PM$_{1.0}$), 4.0 ± 4.5 x 10⁻² $\mu$g m⁻³ (PM$_{2.5}$), and 10.0 ± 2.2 x 10⁻¹ $\mu$g m⁻³ (PM$_{10}$). Although there was an inconsistent variation of PM concentrations with precipitation type, we observed a substantial increase in all PM values for the period July – Aug 2018 and May 2019. In contrast, a decrease in all PM concentrations was observed during Sep 2018 – Mar 2019. This increase in PM values during summer and decrease during winter suggested a seasonal variation at the sampling site. The seasonal variation in PMs may

be indicative of different aerosol particle sources or the local meteorological conditions. In the Southern Great Plains, the local sources include harvesting crop fields and agricultural burning (Garcia et al., 2012; DeMott et al., 2015). Based on the long-term measurements of aerosol particle composition at Southern Great Plains (SGP), Parworth et al. (2015) found a seasonally varying interstate transport of biogenic aerosols to the SGP site. The authors also observed a springtime increase in biomass burning organic aerosols at SGP, which were

mainly associated with local fires. The long-distance dispersion of *Juniperus ashei* pollen into the SGP area by the southern winds was previously observed by Van de Water et al. (2003). Elevated layers of haze have been observed over the same site due to the inter-oceanic and intercontinental transport of smoke from intense Siberian fires (Arnott et al., 2006; Damoah et al., 2004). It was also evident from previous observation and simulation modeling studies that Saharan dust can reach southeastern parts of USA through the transatlantic

long-range transport (Weinzierl et al., 2017). Thus, PMs observed in the West Texas region may be a mixture of aerosol particles from different sources and spatial scales of transport.

  **Table 1** shows the hourly time-averaged PM data measured prior to vs. after precipitation. During intense precipitation, aerosol particle concentrations below cloud tend to decrease due to the wet scavenging effect (Hanlon et al., 2017). In fact, the reduction in our hourly averaged $PM_1$, $PM_{2.5}$, and $PM_{10}$ after precipitation is apparent in **Table 1**, presumably because of scavenging in part at least. Note that any counter mechanisms, such as primary biological aerosol particles and surface material rupture after rainfall (e.g., Huffman et al., 2013), were not considered in our data interpretation. The first order calculations are performed to understand implications of scavenging processes towards the reduction in the PM after rain event (**SI Sect. S4**). These calculations contain ±61.5% uncertainty and can be further extended with some assumptions to estimate INP. However, to better constrain these estimates, direct vertical INP (He et al., 2020) and scavenging measurements (Hanlon et al., 2017) are needed. A total of 28 precipitation events was analyzed, and our estimated $n_{INP}(T)$ of scavenged aerosol particles appeared to be constantly an order magnitude lower as compared to total $n_{INP}(T)$ measured in our precipitation samples (**Fig. S3**). This trend is true across all ranges of examined $T$s (> -25 °C). Nevertheless, our estimates imply some (but negligible) contributions of scavenged aerosol particles on $n_{INP}(T)$ in our precipitation samples.

### 3.2 IoT Air Quality Sensor Results

  An IoT air quality sensor-measured PM concentrations were also analyzed for each precipitation sampling period to understand the effect of wet deposition of PMs on INPs. Figure 3 shows the time-series of average PM concentrations observed during each precipitation episode, overall mean PM values, and the hourly PM data. The overall mean ± standard error PM concentrations calculated from over one year of data were 3.9E+00 ± 9.2E-02 ($PM_{1.0}$), 4E+00 ± 4.5E-02 ($PM_{2.5}$), and 1E+01 ± 2.2E-01 ($PM_{10}$) $\mu$g m$^{-3}$. Although, there was an inconsistent variation of PM concentrations with precipitation type, we observed a substantial increase in all PM values for the period July – Aug 2018 and May 2019. In contrast, a decrease in all PM concentrations was observed during Sep 2018 – Mar 2019. This increase in PM values during summer and decrease during winter suggested the seasonal variation at the sampling site. In addition, the influence of PM values on $n_{INP}$ from each precipitation event was analyzed at -10°C, -15°C, -20°C, and -25°C. The Pearson correlation coefficients (R-value) at -10°C, -15°C, -20°C, and -25°C were statistically insignificant and negative for all PM types (SI Fig. S3). These results suggested no strong positive correlation between the PM and $n_{INP}$ for our sampling period. Moreover, we did not observe a clear sign of wet deposition during a given precipitation event (Stopelli et al., 2015), as there was no decrease in the original hourly PM concentrations during or after the precipitation. Overall, our PM analysis had suggested a local seasonal variation in PM concentrations, and no significant relation between PM and $n_{INP}$ values from our precipitation samples.

### 3.3 INP Results

The time series of cumulative $n_{INP}$ from precipitation samples at different $T$s (i.e., -5, -10, -15, -20, and -25 °C) are shown in **Fig. 3**. The $T$-resolved averaged cumulative $n_{INP}$ ± standard error is also presented in **Fig. 3**. Note that **Fig. 3b** shows $n_{INP}$ for two precipitation samples (ID# 26 and 27) observed on the same day of 12 March

2019. Overall, three orders of magnitude variations of averaged cumulative $n_{INP}$ values were observed between -10 °C (0.17 ± 0.04 L$^{-1}$) and -25 °C (74.74 ± 28.28 L$^{-1}$) for our precipitation samples.  Occasionally, we observed $n_{INP}$ detected at ≥ -5 °C, but such a high $T$ INPs was randomly found in only 7 out of 42 samples within our detection capability.

Attempts to examine the distribution of $n_{INP}$ based on the precipitation type, meteorological season, and maximum precipitation intensity (mm hr$^{-1}$) were made (see **SI Sect. S5**). Due to the limited total number of samples we collected, we cannot conclusively state anything regarding seasonal variations of $n_{INP}$ in our precipitation samples. Nonetheless, our INP results showed that the lowest $n_{INP}$ at -25 °C (3.0 L$^{-1}$) was found in a hail/thunderstorm sample (ID#37; no inclusion of large hydrometeors as seen in **Fig. 2b**) Likewise, the highest $n_{INP}$ at -25 °C (1,130 L$^{-1}$) was found in a hail-involved severe thunderstorm sample (ID# 1) collected in summer 2018. This observation is interesting because the measured PM$_{10}$ of ~6.2 µg m$^{-3}$ prior to precipitation of ID# 1 (**Table 1**) is not the highest PM$_{10}$ recorded in 2018-2019, suggesting wet scavenging does not control the total INPs in precipitation samples. The fact that the second lowest $n_{INP}$ (-25 °C), which is 3.2 L$^{-1}$, is from the snow sample (ID# 23) also supports a negligible contribution of scavenging in our INP data. Moreover, our results showed that cumulative $n_{INP}$ below -20 °C in our precipitation samples could be high in the samples collected while observing > 10 mm hr$^{-1}$ hail/thunderstorm and snow precipitation with notably large hydrometeor sizes.

**Figure 4** shows a compilation of $n_{INP}(T)$ spectra of each precipitation type in comparison to previously reported precipitation $n_{INP}(T)$. In general, most of $n_{INP}$ spectra fall in the upper range of the previous precipitation $n_{INP}$ data presented in Petters and Wright (2015) and Vali (1968). INP humps shaping the reference spectra (i.e., one below -20 °C and another at > -20 °C) are also found in our spectra. The observed hump is especially obvious for $n_{INP}$ at $T$ above -20 °C, and some of our spectra exceed the upper bound of the reference spectra in any precipitation types. For $T$s below -20 °C, our $n_{INP}(T)$ data match fairly well within the range of the reference $n_{INP}(T)$ for all four precipitation types. Thus, the precipitation type observed at the ground level would not have any relationships with INP propensity at least for our 42 samples collected for this study. However, it is interesting that most of our $n_{INP}$ data points above -15 °C fall within the range of estimated $n_{INP}$ at cloud height with < 50% storm efficiency, reported in Vali (1968). In fact, regardless of precipitation type, we see reasonable overlaps of our $n_{INP}(T)$ with Vali (1968). The author stated that the large differences in IN content among precipitation samples were mainly caused by differences in the nucleus content of the air entering the storm. This implies that the cloud level dynamics like cloud entrainment impact the cloud level INP concentrations. Hence, we compared our precipitation INP data with the lower and upper limits of the IN concentrations in the air entering the storm given by Vali (1968) (Table 2, Chapter# 9). These cloud level INP concentrations given by Vali (1968) were for two different storm efficiencies, which is the ratio of mass of precipitation to the mass of water input. The storm efficiency of 10% represents the time when high concentrations of precipitation inside the storm begins to develop. Likewise, 50% is at the peak intensity of the storm. These different combinations of storm efficiencies and water content accounted for a tenfold variation in the ice nucleus content. As more air is entered into the storm with 50% efficiency, more IN concentrations are observed at cloud level. Nonetheless, there is still indeed the need for cloud level INP measurements to define the relationship between the ground level INP concentrations and precipitation intensity.

In addition, **Fig. 4** also shows the $n_{INP}$ result of our 24-hour dry deposition blank sample. For the measured $T$ range, $n_{INP}$ values from the dry deposition blank sample were at least an order of magnitude lower than that from our precipitation samples. This finding corroborated our assumption of negligible contribution of dry deposition in our WT-CRAFT estimated $n_{INP}$ from precipitation samples.

**Figure 5** shows another compilation plot of our precipitation $n_{INP}(T)$ spectra compared to ambient $n_{INP}(T)$ data of local agricultural dusts from Fig. 3 of Hiranuma (2020). As seen, most of our precipitation INP spectra are accumulated near the lower end of the feedlot IN spectra, implying some inclusion of these local dusts as INPs in our samples. Although we are not certain if these local dusts play a role in precipitation, and assessing the potential of locally emitted aerosol particles to precipitation formation is beyond the scope of the current study, it is important to study the contribution of local agricultural dust in wet scavenging and INP formation at cloud height separately in the future. It is noteworthy that adjacent feedlots (> 45,000 head capacity) are located within 33 miles of our sampling site, and the role of feedlot dusts in atmospheric INPs is described in more detail in Hiranuma et al. (2020). Further discussion regarding the feedlot contribution in INPs in our precipitation samples is provided in **Sect. 3.4**.

~~*3.3.1 $n_{INP}(T)$ spectra of each precipitation type*~~

~~Figure 4 shows the IN spectra for different precipitation types analyzed in this study superposed on the IN spectral boundaries adapted from a previous precipitation INP study (Petters and Wright, 2015). This figure also displays other reference IN spectra, including our 24-hour dry deposition blank sample (collected from January 2 – 3, 2019 at our sampling site) and IN spectra measured for dust suspension samples collected from the downwind side of a local feedlot (identity purposely concealed), where substantial and consistent dust emission historically persists (Whiteside et al., 2018). For the measured $T$ range, $n_{INP}$ values from dry deposition blank sample were at least an order of magnitude lower than that from our precipitation samples. This finding corroborated our assumption of negligible contribution of dry deposition in our WT-CRAFT estimated $n_{INP}$ from precipitation samples. Interestingly, the feedlot IN spectra and most of our precipitation samples shown in Fig. 4 were greater than the previously derived precipitation IN upper limit, implying abundant IN active feedlot dusts, which might be involved in the precipitation formation and thereby our samples. It is noteworthy that adjacent feedlots (> 45,000 head capacity) are located within 33 miles of our sampling site. We observed approximately a two orders of magnitude increase in the upper limit of feedlot $n_{INP}$ compared to previous precipitation study at -15°C. For $T \geq$ -15°C, there was at least one sample from each precipitation category falling in the IN spectra region of feedlot dust. Some of the hail/thunderstorm type samples had $n_{INP}$ values in the range of feedlot samples even for the entire $T$ range of 0°C to -25°C (Fig. 4b). For example, at -5°C, $n_{INP}$ from our precipitation samples were in the range of 0.01 – 0.11 L$^{-1}$ of air in the atmosphere. These findings suggest the influence of local feedlot dust on precipitations and on increase of $n_{INP}$ upper limit for precipitation samples. Furthermore, for 0°C $\leq T \leq$ -25°C, we found no precipitation sample with $n_{INP}$ values below the lower limit from the Petters and Wright (2015) study.~~

~~Compared to all other precipitation types, hail/thunderstorm type had the highest average $n_{INP} \pm$ standard error of 0.1 ± 0.01 and 118 ± 68.1 L$^{-1}$ at -5°C and -25°C. In addition, the snow type had the highest average $n_{INP}$ of 0.4 ± 0.3, 0.8 ± 0.5, and 5.7 ± 2.5 L$^{-1}$ at -10°C, -15°C, and -20°C (SI Table S3-1). The lowest $n_{INP}$ values~~

were observed in both the long-lasted and weak rain samples at most of the temperatures. Interestingly, we observed an order of magnitude increase in the maximum $n_{INP}$ calculated at -5°C and -25°C in hail/thunderstorm type compared to long-lasted and weak rains. Such high values of $n_{INP}$ in snow and hail/thunderstorm samples suggested that the INPs impact the severity of a precipitation at least in the West Texas region. These feedlot dusts could reach cloud height and be involve in local aerosol-cloud-precipitation interactions, influencing the local hydrological cycle. Further discussion regarding the feedlot contribution in INPs in our precipitation samples are provided in Section 3.4. We observed a reduced uncertainty in $n_{INP}$ from precipitation samples at $T > -10$°C. For example, a two order magnitude difference was estimated at -8°C in this study, which is lower than previously reported $n_{INP}$ uncertainty at the same temperature (Petters and Wright, 2015). Nonetheless, the discrepancy in $n_{INP}$ still remains at high $T$s. Furthermore, the lower $n_{INP}$ values from this study were greater than the lower limit presented in Petters and Wright (2015). The upper and lower $n_{INP}$ limit derived from this precipitation study could help in comparison studies of $n_{INP}$ at the cloud level to the observed ice-crystal concentrations. Overall, our findings imply that the local feedlot dust contribute to the regional INPs, with an increase in the high $T$ $n_{INP}$ in our precipitation samples.

### 3.3.2. Seasonal variability and INP-precipitation relationship

The time series of cumulative $n_{INP}$ from precipitation samples at different $T$s (i.e., -5°C, -10°C, -15°C, -20°C, and -25°C) is shown in Fig. 5. The overall average, cumulative $n_{INP}$ ± standard error is also presented in Fig. 5. For example, we observed an average $n_{INP}$ of 0.17 ± 0.04 L$^{-1}$ at high $T$ such as -10°C. Figure 5b shows $n_{INP}$ for two precipitation samples (ID# 26 and 27) observed on the same day of 12 March 2019. In total, 24 precipitation samples were collected in the year 2018 and 18 in 2019. A clear variation in high $T$ $n_{INP}$ at -5°C was observed with seasons due to variation in the occurrence of severe precipitations which is discussed below. Furthermore, an increase in $n_{INP}$ at -25°C was observed in the summer of both 2018 and 2019 (Fig. 5a). Overall, 2018 had recorded the highest maximum $n_{INP}$ at -5°C and -10°C (0.11 and 1.62 L$^{-1}$) compared to 2019 (0.06 and 0.65 L$^{-1}$). This high INPs in the year 2018 than in 2019 might be due the presence of more snow and hail/thunderstorms in the year 2018 compared to 2019. A combined total of 14 snow and hail/thunderstorm samples were collected in 2018 and a total of 10 in the year 2019. In order to elucidate this seasonal variation, we further subcategorized our sampling period into four different periods; i.e., May-August (May-Aug; which is a summer season at Canyon, TX), September-October (Sep-Oct), November-January (Nov-Jan; which is a winter season at Canyon, TX), and February-April (Feb-Apr), shown in Fig. 6a. Most of the high $T$ (-5°C) $n_{INP}$ were observed during May – Aug, while there was a decrease in the following seasons, with no INPs at -5°C in the Feb – Apr period. The May - Aug season was dominated by hail/thunderstorms, whereas Feb – Apr had seen mostly long-lasted and weak rains. Likewise, significantly (p-value = 0.09; student's independent t-test) higher INPs were measured at -10°C during Nov – Jan than in Feb – Apr. These findings suggested a strong seasonal variation in INPs, specifically in the high $T$ (≥ -15°C) INPs in Canyon, TX.

The variation of INPs among the precipitation types is shown in Fig. 6b. A statistically significant (p-value ≤ 0.01; student's independent t-test) increase in high $T$ (-5°C) $n_{INP}$ was found in hail/thunderstorm samples compared to long-lasted rains. Additionally, we observed only one sample from weak rain type with $n_{INP}$ at -5°C, supporting the decrease of high $T$ INPs in the less severe precipitation types. Similarly, the

575 ~~distribution of $n_{INP}$ at -25°C for weak rain type was shifted towards relatively lower values than compared with more severe precipitation types, such as hail/thunderstorm. For example, at -25°C, hail/thunderstorm type had a median $n_{INP}$ of 22.44 L$^{-1}$, which was greater than what was measured in weak rain (6.19 L$^{-1}$) type. These results of increase in severity of precipitation with an increase in INPs were further corroborated by our findings from maximum intensity range based $n_{INP}$ analysis (shown in Fig. 6c). For this intensity - $n_{INP}$ analysis, we grouped all our precipitation samples into three different categories based on the observed maximum intensity (mm hr$^{-1}$) in each precipitation event. A significant (p-value ≤ 0.01) increase in INPs at -5°C was found when the maximum intensity was > 50 mm hr$^{-1}$ compared to the range of 10 – 50 mm hr$^{-1}$. The samples from this high intensity~~
580 ~~range (> 50 mm hr$^{-1}$) were mostly coincided with the hail/thunderstorm precipitation types, supporting our previous findings of increase in severity of precipitation with INPs. It is also important to note that there was only one hail/thunderstorm sample which fell in the low intensity range (< 10 mm hr$^{-1}$). Overall, we found a strong seasonal variation in INPs, especially in the high $T$ (-5°C and -10°C) INPs from our yearlong precipitation study. Moreover, we observed an increase in the severity of precipitation with INPs, which highlights the~~
585 ~~importance of INPs in the development and growth of severe precipitation systems in the West Texas region.~~

### 3.4. Microbiome of Feedlot and Precipitation Samples

We carried out a metagenomics analysis of the bacterial microbiome of a subset of our precipitation samples and ambient dust samples collected at commercial feedlots in West Texas to identify 1) potential biological sources of INPs in our precipitation samples, 2) similarities in the microbiome of feedlot dust and precipitation
samples.

We successfully generated data on the bacterial microbiome of our precipitation and feedlot dust samples. Unfortunately, our attempt to extract fungal ~~microbe~~ DNA was not successful due to the limitation in sample amount. Thus, we focus on the bacterial microbiome ~~bacterial discussions~~ hereafter. In most cases, bacterial phyla were classified to the level of genus. The majority of bacteria in all feedlot and precipitation
samples belonged to phyla *Proteobacteria* and *Bacteroidetes* (**Fig. ~~7a~~ 6** and **Table S~~4~~ 5**). In hailstorm samples, the main taxa of *Proteobacteria* were *Massilia* (a genus found in clinical samples and~~,~~ mammals, but also the soil, rhizosphere, and even aerosols), genera belonging to the order *Sphingomonadales* (bacteria with wide metabolic abilities), *Caulobacterales* (bacteria living in diverse terrestrial and aquatic habitats;~~,~~ some are minor human pathogens), and *Rhizobiales* (nitrogen-fixing bacteria forming symbioses with the roots of legumes).
Among the *Bacteroidetes* phylum, the genus *Marinoscillum* was relatively the most abundant. This genus is a recently described marine bacterium, and it is interesting that it was found in hailstorm samples at percentages from 17.3% to 3.2% of the microbiome. Our results perhaps indicate some connection with storms or winds originating from the North Atlantic Ocean (back-trajectory~~trajectories~~ analyses done, but not shown). Other *Bacteroidetes* taxa with notable presence in hailstorm microbiome included *Saprospirales* and *Chitinophagales*
orders with bacteria living on animals and in the gut of animals as expected.

The microbiome~~s of one long-lasted rain sample shared members also found in hailstorms:~~commonly found in our precipitation samples included the genus *Massilia* in significant numbers (11.3% of the microbiome), bacteria of the Proteobacterial orders *Rhizobiales*, *Sphingomonadales*, and *Burkholderiales*; a

significant percentage (8.5%) of the marine genus *Marinoscillum* and bacteria in order *Saprospirales* of phylum *Bacteroidetes*. Our results suggest that no known ~~ice nucleation~~IN active species ~~were~~ detected in precipitation microbiomes. The order *Pseudomonadales*, which includes most known ~~ice nucleation~~IN active species, was found at the limit of detection.

*Massilia* and other unidentified genera of the family *Oxalobacteraceae* were~~was~~ also relatively dominant in all four feedlot samples with percentages from 6.5% to 65.4% of the microbiome. *Marinoscillum*, a marine bacterium surprisingly found in all precipitation samples, was also found in all feedlot samples from 3% to 8.5% of the microbiome. These similarities of the predominant bacteria in the microbiome of four feedlot dust samples and of four precipitation samples taken at an area distant from the feedlots, perhaps indicate~~indicating~~ some connection of the feedlot dust and precipitation microbiomes, ~~these genera~~ either with the formation of precipitation or with their presence in aerosols during precipitation events. Although we cannot rule out the possibility that scavenging of aerosolized bacteria explains the presence of these bacteria both in feedlot and precipitation samples taken even at a distance from feedlots, our dry deposition background result shows complete different biological composition (**Fig. 6**). It is also noteworthy to mention that neither of the~~these two~~ genera (*Massilia* and *Marinoscillum*) were ~~was~~ detected in the background deposition blank sample and it is not known whether they have any ~~ice nucleation~~IN activity. Therefore, the scavenging may not be the main reason for the presence of *Massilia* and *Marinoscillum* found in our precipitation samples. Other bacterial taxa with a significant presence in feedlot samples included members of orders *Caulobacterales* and *Burkholderiales*.

~~Alpha diversity analysis (Shannon's Faith PD index of diversity) indicated that feedlot and hailstorm samples had a lower bacterial diversity than the long-lasting rain sample (**Fig. 7b**). We sought to identify a possible connection between the feedlot microbiome and the microbiome of hail and rain. Beta diversity analysis compared the microbiome diversity distance of feedlot samples between themselves, as well as the microbiome diversity distance of the background deposition, hailstorm and long-lasted rain samples to feedlot samples (**Fig. 7c**). In all comparisons, the distance was at least 0.70, a high value not indicating a "cause and effect" connection between the feedlot and precipitation microbiomes. However, our detailed phylogenetic analysis showed evidence of such a connection by identifying common bacterial taxa in feedlot and precipitation microbiomes. Their absence from the background deposition blank sample may indicate local aerosol-cloud interactions leading to precipitation events, but it is not known if this is a result of any bacterial traits such as ice nucleation activity.~~

### 3.5. Caveats and Future Studies

A surface level air mass on a plain is not necessarily the same as the air mass where precipitation forms at the cloud level. Studying the vertical gradient in INP concentrations in this region would hint at the link between these two vertical zones (e.g., He et al., 2020). The future investigation should also include investigations in physicochemical transformation of hydrometers and INPs, which might occur between the cloud height and the ground (e.g., Pereira et al., 2020), impact of aerosol dynamics and processing, effect of solutes to alter the freezing point (Whale et al., 2018), secondary ice formation, and cloud macrophysics addressed in Wright and Petters (2015 - Sects. 4.1 to 4.3).

The precipitation intensity strongly depends on several other dynamical factors and thermodynamic conditions, including the land use, moisture levels, land surface temperatures, and convective available potential energy. For instance, recent observational study showed that the irrigation practices in the Great Plains region had enhanced summer precipitation intensity (Alter et al., 2015) resulting an increase in the total precipitation received. Hence, it is not straightforward to link the precipitation intensity to the estimated INP concentrations and more future studies involving cloud level and surface level INP measurements might help in elucidating this problem. To assess the impact of INPs on precipitation properties (and vice versa), it is necessary to conduct the INP measurement of cloud water samples, aerosol particle characterizations below cloud, and more detailed analysis of precipitation-forming cloud properties as well as cloud height. More detailed scavenging analysis without many assumptions and limitations, such as assuming a constant scavenging rate over precipitation, limited particle size distributions, and assuming a well-mixed boundary layer, is also necessary to connect the surface observation to cloud level phenomenon. Diffusional scavenging of small particles may not contribute to IN unless they are highly ice active macromolecules or other small biological species. Regardless, robust aerosol particle size distribution data across the ground to cloud base segment would definitely complement to accurately and precisely estimate scavenging efficiencies. Some previous studies support the assumption of a well-mixed boundary layer near the study area. Further effort may be needed to characterize the climatology of boundary layer height in the West Texas region at different times of a day, as demonstrated in Schmid and Niyogi (2012) and Zhu et al. (2001). Incorporating more local specific vertical ambient profiles (lapse rate, Dong et al., 2008) for further analysis would also be helpful.

As for more future studies, INPs derived from precipitation samples collected over multiple years would give comprehensive insight into their impact on local precipitation systems. This work highlights this need for more precipitation-based INP studies from different geographical locations. The reduced uncertainties in $n_{INP}$ along with the high INP detection sensitivity could help in addressing the long-debated issue of INP rarity at $T$s $\geq$ -10 °C.

## 4. Conclusion

We have successfully estimated $n_{INP}$ (per liter of air) in the immersion freezing mode from different precipitation samples collected in Canyon, TX, USA during June 2018 – July 2019. IN spectra were derived for MPC $T$ range (0 °C to -25 °C) from four different precipitation types (snows, thunder/hailstorms, long-lasted rains, and weak rains) using a cold-stage instrument (WT-CRAFT). Our disdrometer measurements showed a clear variation in the precipitation properties among the four different categories of precipitation samples. Severe precipitation, such as hail/thunderstorms, had the highest rainfall intensity (mm hr$^{-1}$) and the number of precipitation particles were highest in the snow samples. We also found an increased number of large hydrometeors (> 108.5 mm in diameter) in both the snow and hail/thunderstorm samples. In contrast, there were no precipitation particles > 6.5 mm in diameter observed in the weak rain samples. Our PM concentration measurements impliedshowed no strong correlation with the measured INPs from precipitation samples some possibilities of wet deposition (but neglected). The IN spectra from each precipitation category in this study were compared with the nucleus IN spectra from previous precipitation-based INP studiesy (Petters and Wright, 2015; Vali, 1986). Previously derived IN spectra from local feedlot dust samples (Whiteside et al., 2018) was also used for comparing $n_{INP}$ from precipitation samples. We have found that $n_{INP}$ values from our precipitation samples

~~were greater than~~match or exceed previously derived $n_{INP}$~~IN upper limits~~ from precipitation. ~~Especially~~Notably, the high $T$ (≥ -15 °C) INPs in some of our precipitation samples are~~were~~ in the same order of magnitude as ~~of local feedlot dust samples~~what is reported in Vali (1986). ~~These findings suggested the importance of local feedlot dusts as INP sources. Moreover, we have observed a strong seasonal variation in $n_{INP}$ in this precipitation based INP study. The May – Aug period had seen the most INPs at -5°C, while none during Feb – Apr season. It is important to note that, hail/thunderstorms were predominantly observed in the May – Aug season. A statistically significant increase in high $T$ (-5°C) INPs was observed in hail/thunderstorms compared to long-lasted rains. Except in one case, we observed no weak rain samples with INPs at -5°C. These findings suggested an increase in high $T$ (-5°C and -10°C) INPs in severe precipitation systems like hail/thunderstorm and snow in the West Texas region. These results were further supported by our findings of increased high $T$ INPs when the rainfall intensity was > 50 mm hr⁻¹. Overall, our results showed that the INPs impact the severity of precipitation systems observed in Texas Panhandle, which represents the importance of more precipitation based INP studies in the future.~~ Although we found no clear seasonal variations in $n_{INP}$ values, in part due to the limited number of samples, the analysis of yearlong ground level precipitation observations as well as INPs for the precipitation samples showed that the highest $n_{INP}$ at -25 °C of 1,130 L⁻¹ coincided with a hail-involved severe thunderstorm event observed during the summer in 2018 (ID# 1). Similarly, the lowest cumulative INP at the same temperature, 3.0 INP L⁻¹, was found in another hail/thunderstorm samples collected in June, 2019 (ID# 37). The second lowest $n_{INP}$ (-25 °C) was found in one of our snow samples collected during the winter (ID# 23 = 3.2 INP L⁻¹). Overall, our results showed that cumulative $n_{INP}$ in our precipitation samples below -20 °C could be high in the samples collected while observing > 10 mm hr⁻¹ precipitation with the presence of notably large hydrometeor sizes. While our results cannot conclusively define the relationship between INPs and precipitation, our precipitation INP data is an important asset for understanding ambient INPs in the West Texas region, where a rural agricultural environment prevails.

We also identified the similarity in bacterial microbiomes between our ~~hailstorm~~ precipitation and local feedlot dust samples.~~,~~ While we cannot conclude if local feedlot dust contributes to precipitation formation, we find some indications of the inclusion of agricultural dust in our precipitation samples~~nevertheless, it is not known whether these microbiomes are IN active~~. Regardless, we did not find the previously known bacterial INPs, such as *Pseudomonas* and *Xanthomonas* (Morris et al., 2004) in either~~both~~ the precipitation or~~and~~ feedlot samples. To further seek a connection between local dust and precipitation,~~Our preliminary analysis showed that organic component was predominant in our precipitation residuals (>70%, not shown in this study), which is similar to the composition of local animal feeding dust (Hiranuma et al., 2011). This similarity might explain the observed increase in $n_{INP}$ from precipitation samples at high $T$s (≥ -15°C). Therefore,~~ it is worthwhile to characterize the local feedlot dust in cloud water samples, as it~~they~~ can be the source~~s~~ of INPs and may~~can~~ impact the local hydrological cycle. Collecting long-term pollen and other biogenic aerosol particles samples and associated observational data for multiple years may add important knowledge regarding the role of local bioaerosols on precipitation INPs.

~~As for more future studies, INPs derived from precipitation samples collected over multiple years would give comprehensive insight into their impact on local precipitation systems. This work highlights this need for more precipitation based INP studies from different geographical locations. Precipitation category based $n_{INP}$ parametrizations can be applicable to any future studies as demonstrated in this study. The reduced uncertainties in $n_{INP}$ along with the observed increase in the lower $n_{INP}$ values from this study could help in~~

**Author Contributions**

Research design: NH, JW; Measurements: HSKV, CAR, GDM, DH, JW, NH; Analysis: HSKV, DGG, NH; Writing: HSKV, NH, DGG. GDM conducted the metagenomics investigation without knowing the identity of samples.

**Competing Interests**

The authors declare that they have no conflict of interest.

**Data Availability**

Original data created for the study will be available in a persistent repository upon publication within www.wtamu.edu.

**Acknowledgements**

The authors acknowledge the financial support by Killgore Graduate Student Research Grant (WT20-017) provided by West Texas A&M University.  This material is based upon work supported by the U.S. Department of Energy, Office of Science, Office of Biological and Environmental Research under Award Number DE-SC-0018979. We also acknowledge Drs. Gourihar Kulkarni for useful discussions regarding implications of scavenging processes on our data.

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

## List of Abbreviations

| | |
|---|---|
| $C_{INP}$ | Nucleus concentration in Precipitation suspension |
| CWC | Cloud water content |
| DF | Dilution Fold |
| $D_p$ | Precipitation particle diameter |
| $f_{unfrozen}$ | Unfrozen fraction (ratio of number of unfrozen droplets to total number of droplets) |
| IoT | Internet of Things |
| IN | Ice-nucleation |
| INP | Ice-nucleating Particle |
| LoRaWAN | Long Range and Wide Area Network |
| MPC | Mixed-phase cloud |
| $n_{INP}$ | INP concentration |
| NWS | National Weather Service |
| PM | Particulate Matter |
| $\rho_w$ | Unit density of water |
| SI | Supplemental Information |
| STP | Standard Temperature and Pressure |
| $T$ | Temperature |
| $V_{air}$ | Sample air volume at cloud level |

$V_d$           Volume of the droplet

$V_t$           Precipitation sample volume

WT-CRAFT           West Texas Cryogenic Refrigerator Applied to Freezing Test

**Figures**

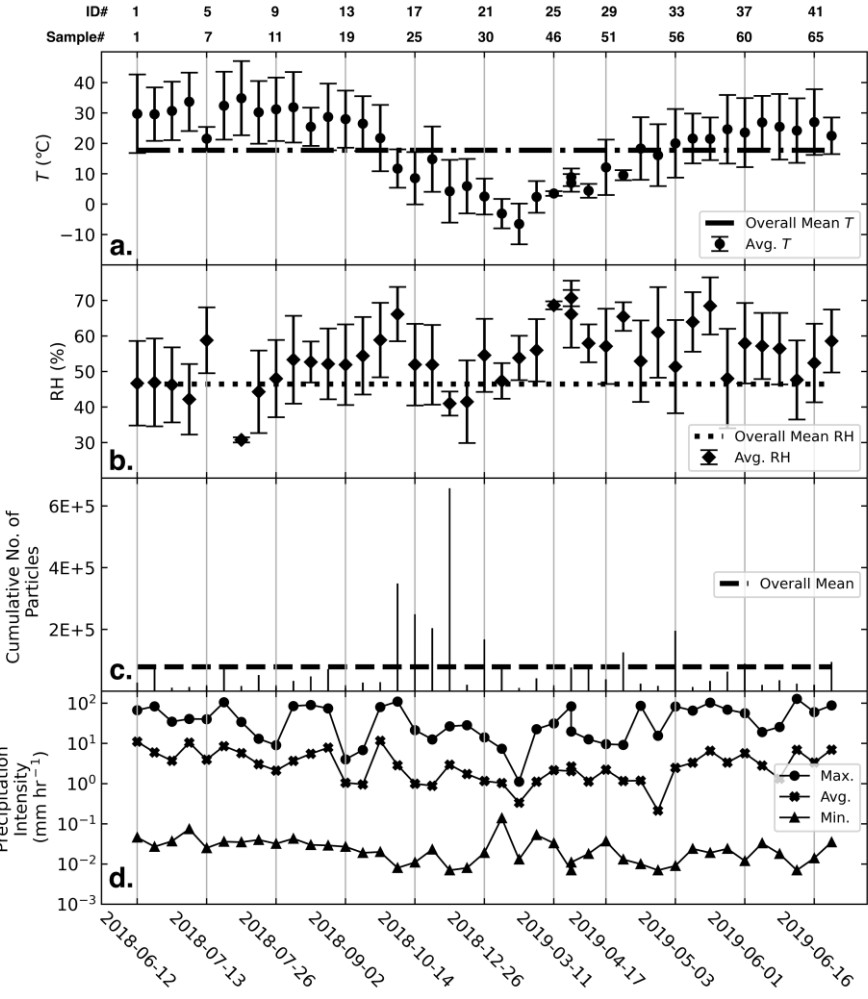

**Figure 1.** Time series of disdrometer and IoT sensor measurements ~~for~~of (a) average *T* ± standard deviation, (b) average relative humidity ± standard deviation, (c) cumulative number of detected hydrometeors in each precipitation event, and (d) maximum, average, and minimum precipitation intensity ~~for each precipitation sample~~. Each data point corresponds to the sampling start time for each precipitation event.

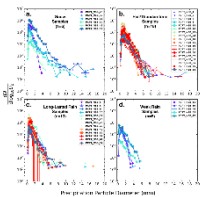

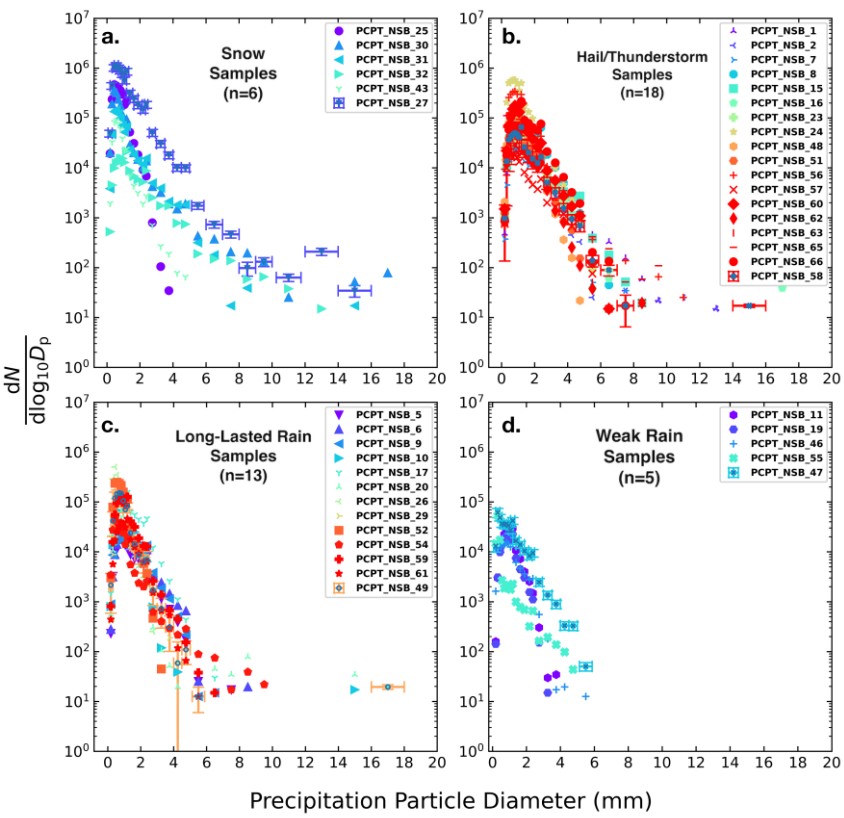

**Figure 2.** Size distribution of precipitation particles detected in (a) Snow, (b) Hail/Thunderstorm, (c) Long-lasted rain, and (d) Weak rain samples. A subset of distributions shows with varying uncertainty in diameter (mm). The X-axis error bars are ± 1.0 mm of size class for diameter < 2mm and ±0.5 mm of size class for diameter > 2mm. The Y-axis error bars represent standard errors at each diameter. The sub-total number of precipitation samples in each category is shown by the value of 'n'.

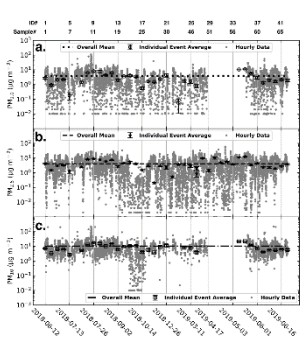

**Figure 3.** Time series of IoT air quality sensor measurements of (a) $PM_{1.0}$, (b) $PM_{2.5}$, and (c) $PM_{10}$ for each precipitation event. Hourly data include the non-precipitation periods (grey dots). The Y-axis error bars are standard errors measured for each precipitation event.

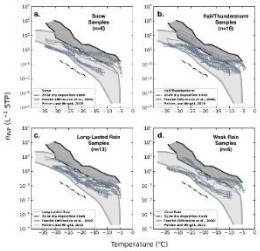

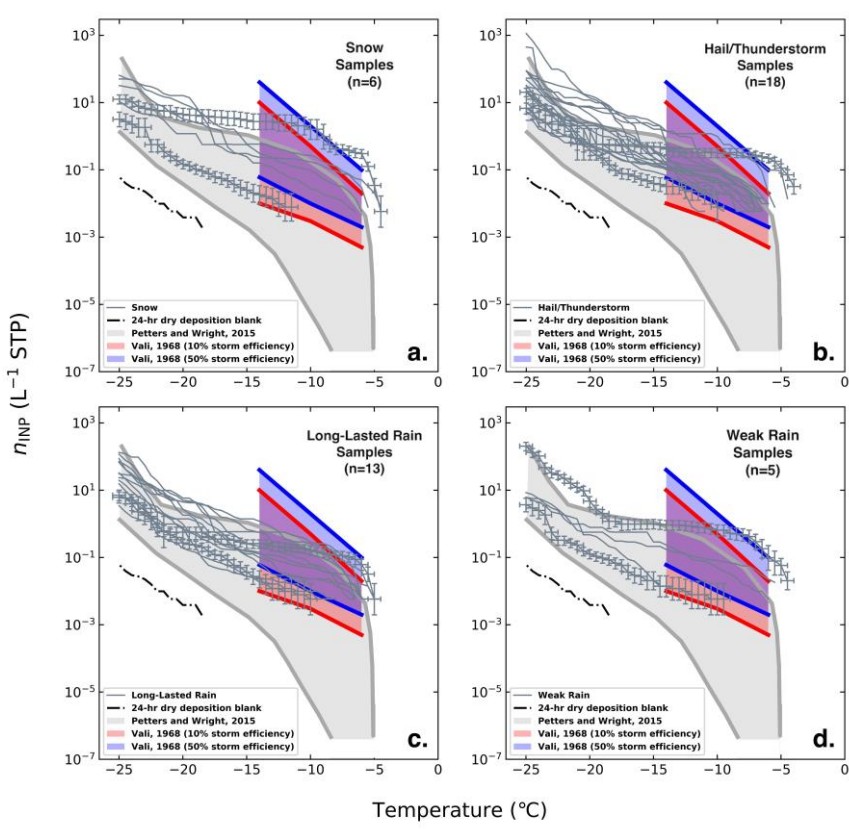

**Figure 44.** IN spectra of (a) Snow, (b), Hail/Thunderstorm, (c) Long-Lasted rain, and (d) Weak rain samples superposed on nucleation spectra from previous precipitation INP studies (shaded areas). A subset of spectra shows error bars. The X-axis error bars represent constant uncertainty of ±0.5 °C in temperature. The Y-axis error bars are 95% confidence interval for $n_{INP}$ shown only for two samples from each category. The number of precipitation samples in each category is shown by the value of 'n'.

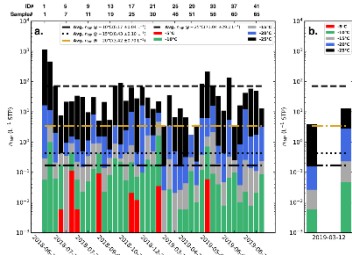

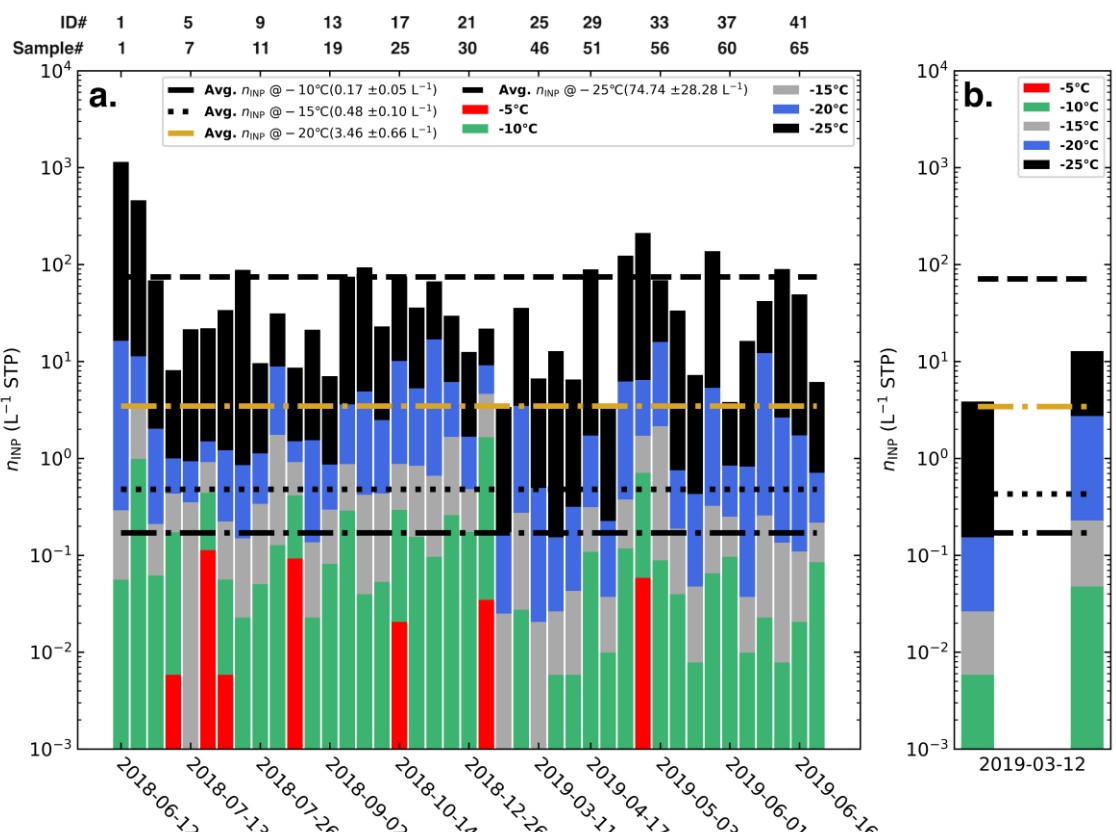

**Figure 5̶3.** (a) Time series of cumulative $n_{INP}$ (L$^{-1}$ air) in each precipitation sample at different temperatures. (b) $n_{INP}$ for two precipitation samples (ID# 26 and 27) observed on the same day of 12 March 2019. The uncertainty in the average $n_{INP}$ at each temperature (± numbers in parentheses) is the standard error calculated for 42 samples.

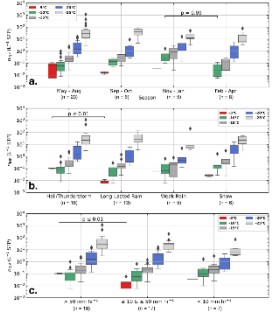

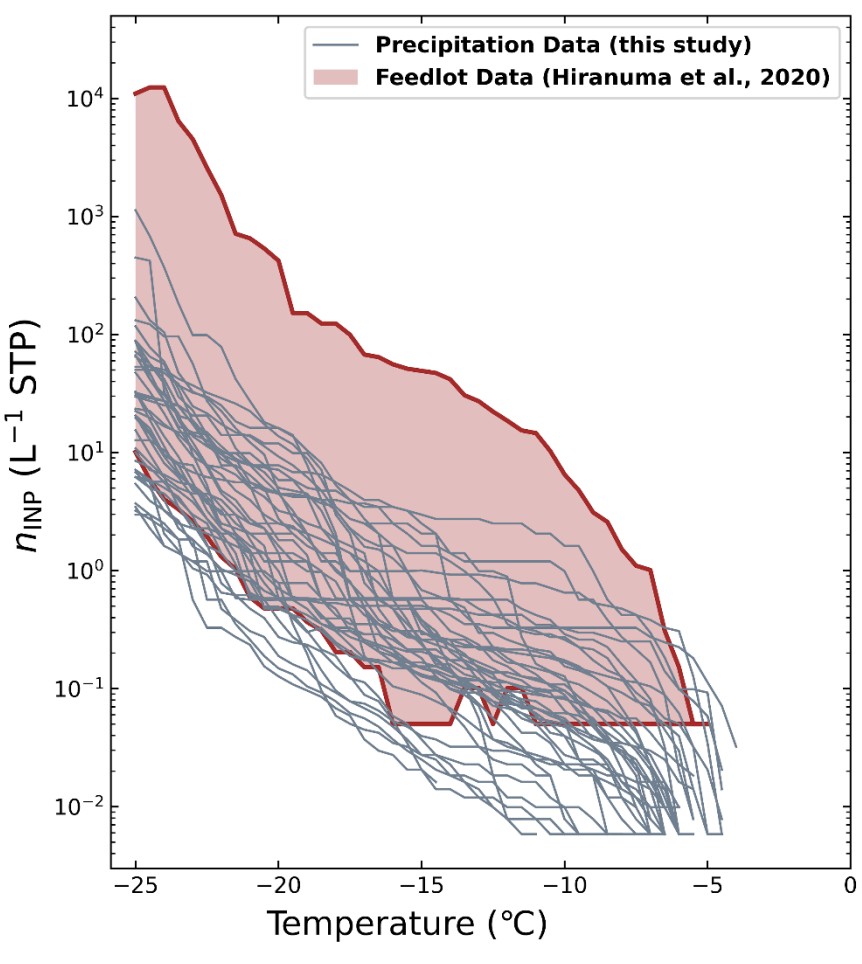

~~**Figure 6.** The distribution of $n_{INP}$ for different (a) seasons, (b) precipitation types, and (c) range of maximum intensity observed in precipitation sample. The p-value from student's independent t-test is also shown by 'p' at different $T$s between different categories. The solid line in the box is median $n_{INP}$ and the box edges are the inter-quartile ranges q1 (25 percentile) and q3 (75 percentile). The diamond markers are the outliers with maximum $n_{INP}$ values.~~

**Figure 5.** Compiled IN spectra of our precipitation samples superposed on nucleation spectra from local feedlot dust study (shaded area). The feedlot INP data are adapted from Fig. 3 of Hiranuma et al. (2020).

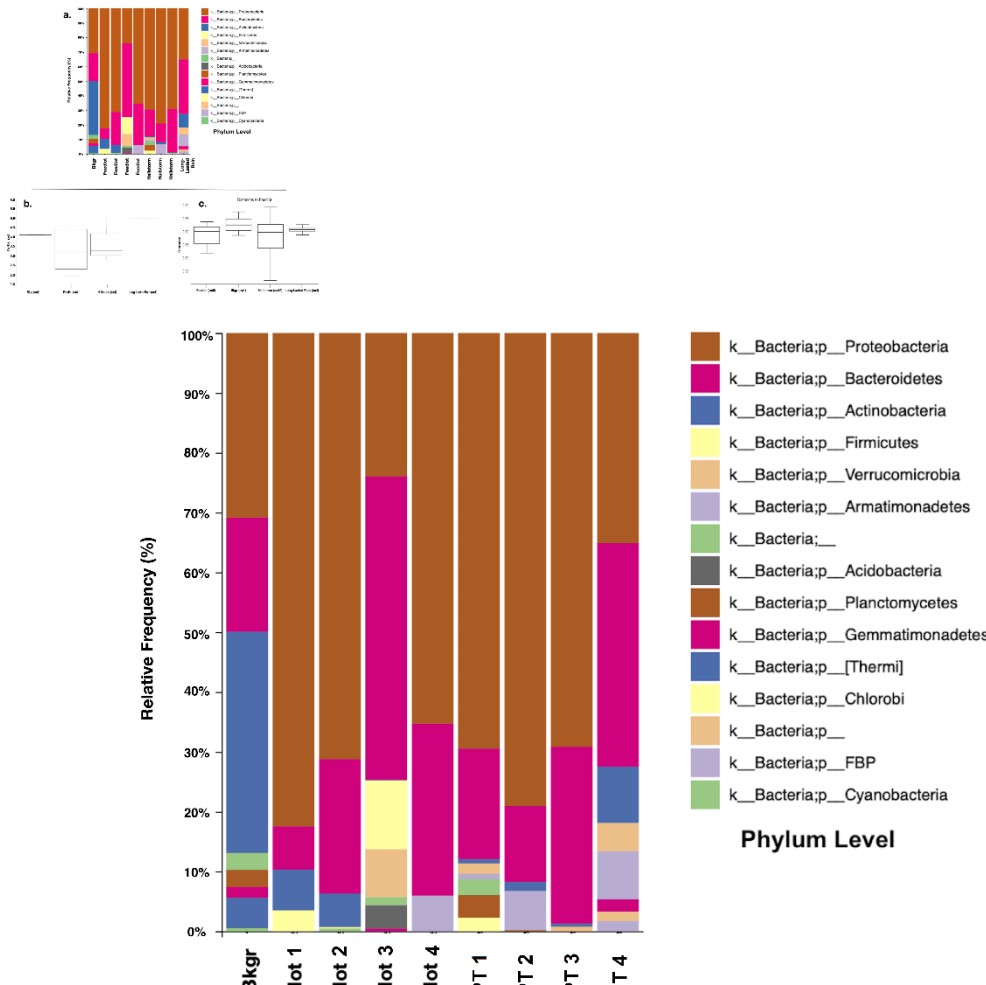

**Figure ~~7~~ 6.** Metagenomics analysis of precipitation and feedlot dust samples showing ~~(a)~~ Relative Frequency (%) or abundance of Bacterial taxonomy.~~, (b) alpha-diversity analysis with Faith's PD index of diversity (Y-axis), and (c) beta-diversity analysis comparing microbial distance of feedlot samples between themselves, as well as the microbiome diversity distance of other samples.~~ 'Bkgr' represents the 24-hour dry deposition blank sample (Sample# 34). Our feedlot samples are collected

locally on March 28, 2019 (1), July 22, 2018 (2), July 23, 2018 (3), and July 24, 2018 (4) – see Hiranuma et al. (2020). PCPT 1-4 corresponds to our Sample# 1, 2, 50, and 7, respectively.

**Table 1**. Adjacent hourly averaged PM values before and after each precipitation event. We excluded 14 data where PM data were not recorded due to technical issues etc. (ID# of 6-7, 17, 20, 22-24, 26, 28-33).

| ID# | Sample# | Precipitation type | | PM$_1$ (µg m$^{-3}$) | | | PM$_{2.5}$ (µg m$^{-3}$) | | | PM$_{10}$ (µg m$^{-3}$) | |
|---|---|---|---|---|---|---|---|---|---|---|---|
| | | | | Before | After | | Before | After | | Before | After |
| 1 | PCPT_NSB_1 | Hail/Thunderstorm | | 1.969 | 0.111 | | 4.090 | 1.693 | | 6.188 | 1.990 |
| 2 | PCPT_NSB_2 | Hail/Thunderstorm | | 0.010 | 0 | | 1.811 | 0.001 | | 2.111 | 0.001 |
| 3 | PCPT_NSB_5 | Long-Lasted Rain | | 4.667 | 0.660 | | 5.734 | 1.947 | | 10.790 | 3.690 |
| 4 | PCPT_NSB_6 | Long-Lasted Rain | | 3.755 | 3.755 | | 5.956 | 5.721 | | 8.867 | 8.580 |
| 5 | PCPT_NSB_7 | Hail/Thunderstorm | | 0 | N/A | | 0.557 | N/A | | 0.723 | N/A |
| 8 | PCPT_NSB_10 | Long-Lasted Rain | | 7.479 | 1.495 | | 9.894 | 3.409 | | 14.771 | 4.742 |
| 9 | PCPT_NSB_11 | Weak Rain | | 5.760 | 3.812 | | 8.165 | 6.190 | | 12.770 | 9.436 |
| 10 | PCPT_NSB_15 | Hail/Thunderstorm | | 14.289 | 4.020 | | 16.078 | 5.124 | | 30.794 | 9.277 |
| 11 | PCPT_NSB_16 | Hail/Thunderstorm | | 4.913 | N/A | | 5.423 | N/A | | 10.534 | N/A |
| 12 | PCPT_NSB_17 | Long-Lasted Rain | | 4.551 | N/A | | 6.414 | N/A | | 10.633 | N/A |
| 13 | PCPT_NSB_19 | Weak Rain | | 0.049 | N/A | | 1.283 | N/A | | 6.301 | N/A |
| 14 | PCPT_NSB_20 | Long-Lasted Rain | | 1.780 | N/A | | 4.312 | N/A | | 5.890 | N/A |
| 15 | PCPT_NSB_23 | Hail/Thunderstorm | | 3.867 | 2.167 | | 5.740 | 5.740 | | 9.551 | 7.235 |
| 16 | PCPT_NSB_24 | Hail/Thunderstorm | | 1.592 | 0 | | 4.984 | 0.003 | | 5.786 | 0.003 |
| 18 | PCPT_NSB_26 | Long-Lasted Rain | | 0.657 | 0 | | 2.830 | 0 | | 3.192 | 0 |
| 19 | PCPT_NSB_27 | Snow Sample | | 0 | N/A | | 0.011 | N/A | | 0.080 | N/A |
| 21 | PCPT_NSB_30 | Snow Sample | | 0.760 | 0 | | 2.627 | 0.275 | | 3.180 | 0.275 |
| 25 | PCPT_NSB_46 | Weak Rain | | 1.461 | 0 | | 4.525 | 1.233 | | 5.449 | 1.233 |
| 27 | PCPT_NSB_48 | Hail/Thunderstorm | | 0 | 0 | | 0.427 | 0.002 | | 0.427 | 0.002 |
| 34 | PCPT_NSB_57 | Hail/Thunderstorm | | 29.649 | 13.515 | | 29.649 | 13.770 | | 58.946 | 26.604 |
| 35 | PCPT_NSB_58 | Hail/Thunderstorm | | 12.450 | 0.680 | | 13.245 | 1.400 | | 24.390 | 2.860 |
| 36 | PCPT_NSB_59 | Long-Lasted Rain | | 10.515 | 6.912 | | 11.516 | 7.918 | | 21.192 | 12.892 |
| 37 | PCPT_NSB_60 | Hail/Thunderstorm | | 9.740 | 3.423 | | 10.661 | 4.396 | | 18.750 | 7.269 |
| 38 | PCPT_NSB_61 | Long-Lasted Rain | | 4.396 | 0.192 | | 5.912 | 1.215 | | 10.069 | 2.051 |
| 39 | PCPT_NSB_62 | Hail/Thunderstorm | | 0.039 | N/A | | 1.555 | N/A | | 1.804 | N/A |
| 40 | PCPT_NSB_63 | Hail/Thunderstorm | | 2.217 | 1.365 | | 4.348 | 2.479 | | 6.533 | 4.781 |
| 41 | PCPT_NSB_65 | Hail/Thunderstorm | | 1.694 | 0 | | 3.994 | 0.316 | | 5.306 | 0.316 |
| 42 | PCPT_NSB_66 | Hail/Thunderstorm | | 1.750 | 0.080 | | 2.881 | 1.459 | | 5.771 | 1.530 |

NOTE: N/A: either below detection limit of our PM sensor (< 0.001 µg m$^{-3}$) or sensor failure return values.