# Peer review of "Ice-nucleating particles in precipitation samples from West Texas"

_Atmospheric Chemistry and Physics, 2020_

## Referee Comment (RC1) · Anonymous Referee #1 · 2 Oct 2020

Throughout the course of 13 months the authors have sampled 42 precipitation events in the Northwest of Texas and analysed INP concentration in the hydrometeors. Parallel observations included the size distribution of hydrometeors, airborne particulate matter, air temperature and humidity. Precipitation samples were further subjected to metagenomic analysis, together with a dry deposition sample collected at the same site and suspended dust samples from a cattle feedlot about 50 km away. Data on this variety of parameters was then combined in an interpretation involving numerous implicit and some explicit assumptions, but neglegting two important issues: (a) that surface level air mass on a plain is not necessarily the same as the air mass where precipitation forms (typically "... 2 km to 9 km above ground level ..." (line 66); for vertical gradients in INP concentrations see He et al. (2020)), and (b) that hydrometeors scav-

[Figure]

enge particles between cloud and ground level. Latter was clearly demonstrated by Hanlon et al. (2017), who produced showers of artificial, sterile rain from a road bridge and collected the artificial hydrometeors, including microbial ice nucleators scavenged during 55 m of free fall, on the field below.

Figure 4 exemplifies the problem I see with the combination of little-related data and ignored processes. The Figure combines INP data on airborne dust samples near ground (feedlot, 50 km away from other observations), INP estimates of atmospheric INP concentrations at cloud height derived from precipitation samples and an assumed cloud water content (ignoring scavenging of particles and loss of water through partial evaporation of raindrops during free fall (ground level RH during rainfall 31% to 71% (line 309)), and an atmospheric INP estimate based on a dry deposition sample suspended in an (arbitrary?) volume of pure water and transformed into an atmospheric concentration value. I think the data from these three kinds of sources can not be directly compared because of mentioned issues.

However, the paper definitely contains new and interesting observations that may be interpreted to a certain extent, without making too many implicit or explicit assumptions. These observations are foremost the INP concentrations in precipitation samples combined with the precipitation properties, including kind of precipitation, size spectra of the hydometeors, precipitation duration and intensity. Such an interpretation needs to address the issues of below-cloud scavenging and also the higher scavenging efficiency of snow as compared to rain (Wang et al., 2014). I also found interesting the occurence of marine bacteria in the precipitation samples. In contrast, data on particulate airborne matter near ground level is something I would put aside when revising the manuscript.

Minor issues

I found it tedious to read through listings of data in the Results and Discussion section. Somehow, I missed a clear storyline. It would have been a more engaging reading

experience, if Figures were not introduced by full sentences that resemble Figure legends. To give an example (lines 372 and following): "Figure 4 shows the IN spectra for different precipitation types analyzed in this study superposed on the IN spectral boundaries adapted from a previous precipitation INP study (Petters and Wright, 2015). This figure also displays other reference IN spectra, including our 24-hour dry deposition blank sample (collected from January 2 – 3, 2019 at our sampling site) and IN spectra measured for dust suspension samples collected from the downwind side of a local feedlot (identity purposely concealed), where substantial and consistent dust emission historically persists (Whiteside et al., 2018). For the measured T range, nINP values from dry deposition blank sample were at least an order of magnitude lower than that from our precipitation samples." This entire section could simply be replaced with: "For the measured T range, nINP values from dry deposition blank sample were at least an order of magnitude lower than that from our precipitation samples (Figure 4)."

What is meant by (line 313): "...substantial number of precipitation particles with a cumulative number of 2E+05 to 6.6E+05 per event." Perhaps "...precipitation particles recorded by the disdrometer...", or "...precipitation particles per square metre..."?

Lines 323-327: It is not clear why the range of intensities is indicated as "0 to 150 mm hr-1", when maximum intensity was 129.3 mm hr-1 and minimum intensity 1.1 mm hr-1?

References: Whiteside et al. 2018: I would have liked to learn more about this study, but could not find it. A link to the paper, if available, would have been usefull.

Gabor Vali determined INP spectra in rain and hail samples from numerous storms in various parts of North America (Vali, 1968). The authors may find it helpful to have a look at his work when revising their manuscript.

References

[Figure]

Hanlon et al. (2017) Microbial ice nucleators scavenged from the atmosphere duringsimulated rain events scavenging of particles by hydrometeors. http://dx.doi.org/10.1016/j.atmosenv.2017.05.030

He et al. (2020) Aircraft observations of ice nucleating particles over the Northern China Plain: Two cases studies. https://doi.org/10.1016/j.atmosres.2020.105242

Vali (1968) Ice nucleation relevant to formation of hail. https://escholarship.mcgill.ca/concern/theses/h702q709t?locale=en

Wang et al. (2014) Development of a new semi-empirical parameterization for below-cloud scavenging of size-resolved aerosol particles by both rain and snow. https://doi.org/10.5194/gmd-7-799-2014

---

## Referee Comment (RC2) · Anonymous Referee #2 · 14 Oct 2020

The authors present a set of observation of INP concentrations from rainwater samples collected over West Texas during a 13 months period. They also measure data of precipitation properties, atmospheric temperature, relative humidity and air quality. In addition to that, they performed a metagenomics analysis to obtain information about bacteria present in the rain samples. I personally think that the technical methods used are correctly presented and used. However, the interpretation of their results and the relations they claim be proving between INP and precipitation intensity seems to me completely inaccurate and not backed by their method nor their data at all. There are 4 major points why I think this paper should be rejected:

-Correlation does not imply causality: The intensity of rain is subject to change due

to dynamical a thermodynamical factors. For example, I would expect that the large increases in CAPE over the summer season make convective clouds much more intense due to stronger updrafts than those observed on the more stratiform precipitation characteristic of winter like cyclone driven rain. None of these points is addressed minimally in the paper although they are the main drivers of precipitation intensity. INP concentrations can also change seasonally due to a variety of factors (dust transport, dryer conditions, higher biological productivity etc... ), such factors are also barely mentioned. Finding a correlation between these 2 variables does not imply any type of causality between them. Also, in case you find a strong correlation, you should attempt to see what direction is this going, is it INP affecting rain or rain affecting INP? You could likely do a similar study with any variable, and you might likely find similar correlations.

-Wet deposition on rain particles is not properly addressed: Whereas they mention that surface PM does not correlate with INP, this does not discard that wet deposition might be affecting their results. Surface PM is not necessarily a measure of free tropospheric aerosol concentrations, and it is well known that during strong precipitation, aerosol concentrations tend to decrease due to wet deposition. The non correlation between PM and INP is perhaps showing that INP concentrations might be independent on the total aerosol concentration, which is likely given their rareness. The authors could measure the importance of wet scavenging by analysing the number of particles in their rain samples collected at the surface and just below cloud. There are strong evidences in their data that point towards wet scavenging being critical, such as how their largest INP concentrations occur on snow samples, which are best at wet scavenging.

-Their statistical analysis is not presented in detail and strongly limited to a few self-selected data samples: The two-sample t-test is a parametric test. Therefore, first they need to show that their distributions are normal, which I think they probably are in logarithmic scale but not on linear scale. Then, they need to present their results clearly and broadly in a reproducible manner, showing the number of datapoints going

in each of the calculations and which dataset are you comparing. Currently they only show the final p-value for a couple of comparisons at high temperature which to me seems not valid at all for a scientific publication.

-Their data seems to show many times the opposite to what they claim: Looking at the available data in the supplementary, I can see that intensity of the rain types increases from snow to weak rain to long-lasted rain to hailstorm (being this last one the most intense) (Table S1-3). The INP values presented in table S3-1 do not correlate at all with their conclusions, being typically snow the precipitation category with the highest INP measured (at -10, -15, -20 and -25C) while having the weakest intensity. Of course, this is not the same analysis as performed by the authors, but given the data available in the paper, it seems that the conclusions should, in any case, go the other way around.

Extra comments.

Section 3.2. I like that the authors address the wet deposition factor in this section. However, I do not understand why they relate directly wet deposition with the ambient PM. Wet deposition depends on many factors (size distribution of particles, height from where the droplet falls, etc...) It could have been much more accurate to measure directly the number of particles in each of their precipitation samples.

L366. Whereas a measurable decrease in surface PM during rain suggests a clear removal by wet deposition, a non-measurable decrease in surface PM does not discard wet deposition as the particles could have been absorbed higher up and in amounts below the detection limit.

L393. Snow is a much better scavenger of aerosols than rain. This might be a likely explanation on why you get higher INPs in snow samples. You could test this by measuring the number of particles (and their size distribution) in your snow samples.

L397-397. I do not see the link here between these 2 ideas.

L426-429. This observation goes against the conclusions of the paper. You observed

lower -10C INP values during the May-Aug season when precipitation is stronger (due to the appearance of convective storms) than in the Nov-Jan season.

L430-433 How many points with -5C INPs in the hail/thunderstorm type where included in the analysis. It seems from the plot that there was only 1 point or that all points had the same value. In the supplementary this information is not included.

L440 As per my previous comments, I am not sure how many points are included in this analysis.

L445 what are the results of the statistical analysis over the other temperatures? Showing only the -5C p-values is not enough.

L447 I don't think this statement is backed by your results.

L515-517 Showing some correlations that might be affected by seasonal variations is not enough to claim such a statement.

---

## Referee Comment (RC3) · Anonymous Referee #3 · 20 Oct 2020

The paper is not appropriate for publication. The paper tries to link INP properties and precipitation events of different strength. I was expecting at least some interesting results in Sect. 3.3 (INP results), after reading 8 pages of introductory and technical aspects. . . and after further reading of the result sections 3.1 and 3.2. But at the end there were no solid findings and convincing results. The paper contains many figures and many speculative statements. This not sufficient and satisfactory.

My main problem with the manuscript: I am not convinced that one can try to simply link INP concentration measurements at ground with rain events. You need to know cloud base where most of the aerosol particle enter the rain-producing cloud, you need to know cloud top height where ice nucleation typically starts, there may be entrainment of INP from the side. . . The strength of the thunderstorm or more generally of the rain

event depends on the water vapor reservoir and meteorological conditions (sounds trivial), all this is not known here. Furthermore, on the way to the surface the rain drops collect a lot of aerosol particles (scavenging of pollution, biological and dust particles). All this material you will finally find in the collected rain water. So many questions, I got during reading and reviewing, remained open.

The paper must be rejected.

---

## Author Comment (AC1) · 22 Dec 2020

*Response to Referee #1*

First of all, the authors thank the referee for submitting helpful and meaningful comments, which lead to improvements and clarifications within the manuscript.

Below, we provide our point-by-point responses. For clarity and easy visualization, the Referee's comments (*RC*) are shown from here on in black. The authors' responses (*AR*) are in blue color below each of the referee's statement. In addition to the responses to referees' comments, we further modified the manuscript to increase its clarity and readability. The summary of major and minor changes is included at the end of this document. We introduce the revised materials in green color along/below each one of your response (otherwise directed to the Track Changes version manuscript).

*RC:* Throughout the course of 13 months the authors have sampled 42 precipitation eventsin the Northwest of Texas and analysed INP concentration in the hydrometeors. Par-allel observations included the size distribution of hydrometeors, airborne particulatematter, air temperature and humidity. Precipitation samples were further subjected tometagenomic analysis, together with a dry deposition sample collected at the samesite and suspended dust samples from a cattle feedlot about 50 km away.

*AR:* The authors appreciate these general remarks regarding our manuscript by Referee #1. Below, we provide our point-by-point responses. To reflect our changes and articulate what is truly presented in the revised version paper, the authors have decided to change the title of manuscript to "**Ice-nucleating particles in precipitation samples from West Texas**". We have also revised our abstract as well as the conclusion to reflect all of our major revisions (please see the Track Changes version paper).

*RC:* Data on this variety of parameters was then combined in an interpretation involving numerous implicit and some explicit assumptions, but neglecting two important issues: (a) that surface level air mass on a plain is not necessarily the same as the air mass where precipitation forms (typically "... 2 km to 9 km above ground level ..." (line 66); for vertical gradients in INP concentrations see He et al. (2020)), and (b) that hydrometeors scavenge particles between cloud and ground level. Latter was clearly demonstrated by Hanlon et al. (2017), who produced showers of artificial, sterile rain from a road bridge and collected the artificial hydrometeors, including microbial ice nucleators scavenged during 55 m of free fall, on the field below.

*AR:* The authors highly appreciate these general and intuitive remarks regarding our manuscript by Referee #1. We also thank the referee for providing us with the references. The reviewer is absolutely right about (a). The INP concentration in general decreases from near

ground to cloud height over plains as reported in He et al. (2020). It is clear that using our $n_{INP}$ values in precipitation samples collected at the ground level to assess the impact on precipitation properties at cloud height (and vice versa) is not appropriate. Concerning this and many other issues raised by all peer reviewers (e.g., cloud water ≠ precipitation water), the authors decided to substantially revise the manuscript to focus on presenting the observed variation in precipitation properties but somewhat similar $n_{INP}(T)$ in different precipitation systems and their metagenomics analysis.

The authors agree with the referee's point (b). We provide our new interpretation and example to explore potential implications of wet scavenging on our data in the new **Sect. 3.2** and **SI Sect. S4** and to further motivate the research. Please see the Track Changes version of the manuscript and SI. Briefly, **Fig. 1** below shows the estimated INP concentration of scavenged aerosol particles at four different $T$s, $n_{INP,sv}(T)$, based on scavenged mass simulated with column-integrated mean PM$_{10}$ (see **SI Sect. S4**). We also show the measured INP concentrations of our precipitation samples, $n_{INP,pcpt}(T)$ [L$^{-1}$], for comparison. As seen in this figure, our estimated $n_{INP,sv}(T)$ values are constantly much lower as compared to $n_{INP,pcpt}(T)$. This trend is true across all ranges of examined $T$s even if we used the ground level PM$_{10}$ as for scavenging inputs. As noted in **Sect. 3.5**, due to many assumptions we made for this analysis, our results of $n_{INP,sv}(T)$ being much smaller than $n_{INP,pcpt}(T)$ may not be conclusive and indeed requires further detailed study. Nevertheless, our estimates suggest the

[Figure]

**Figure 1.** (a) Time series of cumulative $n_{INP}$ (L$^{-1}$ air) in each precipitation sample (ID# shown on the x-axis) at different temperatures. (b) Estimated $n_{INP,sv}$ for a total of 28 samples analyzed based on $M_{sv,cm}$. All data above our $n_{INP}$ detection limit of > 0.006 L$^{-1}$ are shown. The average $n_{INP}$ values at -25 °C (74.7 L$^{-1}$) and -20 °C (3.5 L$^{-1}$) in all precipitation samples are shown to guide the reader's eye.

presence of $n_{INP,sv}(T)$ in our precipitation samples. Though the estimated $n_{INP,sv}(T)$ values may be negligible, the authors respectfully take the reviewer's words and removed all discussions associated with influence of INP on precipitation intensity etc.

*RC:* Figure 4 exemplifies the problem I see with the combination of little-related data and ignored processes. [1] The Figure combines INP data on airborne dust samples near ground (feedlot, 50 km away from other observations), [2] INP estimates of atmospheric INP concentrations at cloud height derived from precipitation samples and an assumed cloud water content (ignoring scavenging of particles and loss of water through partial evaporation of raindrops during free fall (ground level RH during rainfall 31% to 71% (line 309)), and [3] an atmospheric INP estimate based on a dry deposition sample suspended in an (arbitrary?) volume of pure water and transformed into an atmospheric concentration value. I think the data from these three kinds of sources cannot be directly compared because of mentioned issues.

*AR:* RE [1]: Agricultural dust is a predominant local dust source in West Texas throughout the year as a number of feedlots exist "within" 33 miles of our precipitation sampling location. Thus, feedlots might act as multiple roles, such as locally emitted INPs, precipitation INPs (if they reach the cloud height), or scavenged aerosol particles. Our result of a dry deposition sample (Sample# 34 – see **Fig. 4** in the revised manuscript) suggests the limited contribution of local aerosol particles, including feedlot dust. Likewise, our assessment of wet deposition, now presented in **SI Sect. S4**, shows $n_{INP,PCPT}(T)$ being much larger than $n_{INP,sv}(T)$. As the possibility of feedlot dust entering clouds cannot be ruled out, the authors would like to retain the discussion of potential contributions of local agricultural dust to precipitation INPs. To clarify our point of including feedlot dust INP data, we have added the following sentences in **Sect. 3.3**;

"Although we are not certain if these local dusts play a role in precipitation, and assessing the potential of locally emitted aerosol particles to precipitation formation is beyond the scope of the current study, it is important to study the contribution of local agricultural dust in wet scavenging and INP formation at cloud height separately in the future. It is noteworthy that adjacent feedlots (> 45,000 head capacity) are located within 33 miles of our sampling site, and the role of feedlot dusts in atmospheric INPs is described in more detail in Hiranuma et al. (2020). Further discussion regarding the feedlot contribution in INPs in our precipitation samples is provided in **Sect. 3.4**."

Please also see our new **Sect. 3.4**;
"…Although we cannot rule out the possibility that scavenging of aerosolized bacteria explains the presence of these bacteria both in feedlot and precipitation samples taken even at a distance from feedlots, our dry deposition background result shows different biological

composition (**Fig. 6**). It is also noteworthy to mention that neither of the genera (*Massilia* and *Marinoscillum*) were detected in the background deposition blank sample and it is not known whether they have any IN activity. Therefore, the scavenging may not be the main reason for the presence of *Massilia* and *Marinoscillum* found in our precipitation samples…"

***AR:*** RE [2]: The authors revised the text in **Sect. 2.5** to clarify our points regarding CWC as follows.

"We presumed CWC to be a constant of 0.4 g m$^{-3}$, covering the continental clouds in our study. Our assumption would be reasonable since Petters and Wright (2015) showed that the variation of $n_{INP}$ with CWC values for different cloud types in the atmosphere would typically be limited within a factor of two, and our $n_{INP}$ uncertainties could be larger than that. Thus, the effect of CWC on the $n_{INP}$ would be negligible."
→
"We assumed CWC to be a constant of 0.4 g m$^{-3}$, following Petters and Wright (2015). This assumption would be reasonable for the following three reasons: (1) Petters and Wright (2015) and references therein showed typical values of CWC for different cloud types could narrowly range from 0.2 g m$^{-3}$ to a factor of few more, (2) the authors also showed that the variation of $n_{INP}$ with CWC values for different cloud types in the atmosphere would typically be limited within a factor of two, and our $n_{INP}$ uncertainties could be larger than that, and (3) based on a parametrization for rainwater evaporation, Zhang et al. (2006) suggests that evaporation does not contribute to $n_{INP}$ bias for both strong convective systems and persistent rain events with cloud base heights of ≈3 km. Thus, the variation of CWC on the $n_{INP}$ was considered to be negligible. Nonetheless, it is necessary in the future to further investigate in cloud specific CWCs incorporating with loss of water through partial evaporation of raindrops during free fall based on vertical vapor deficit profiles to conclusively assess if this assumption is fair or not. Precipitation evaporation rate might introduce bias in $n_{INP}$ for precipitation systems with high cloud base, and the correction can be applied accordingly (Petters and Wright, 2015). Direct comparison between INP measurements in cloud water samples and those in precipitation samples might also be key to answer this question (e.g., Pereira et al., 2020)."

***AR:*** RE [3]: Thanks for asking. The volume of pure water used to assess our dry deposition sample (Sample# 34), which is 5 mL, was arbitrarily determined by averaging collected precipitation volumes of all prior samples (Sample# 1 to 33). A new sentence is now added in P4L141 to clarify this as follows:
"We note that a volume of pure water (5 ml) for an atmospheric INP estimate based on a dry deposition sample was determined by averaging collected precipitation volumes of all samples prior to this dry deposition sample."

*RC:* However, the paper definitively contains new and interesting observations that may be interpreted to a certain extent, without making too many implicit or explicit assumptions. These observations are foremost the INP concentrations in precipitation samples combined with the precipitation properties, including kind of precipitation, size spectra of the hydometeors, precipitation duration and intensity. Such an interpretation needs to address the issues of below-cloud scavenging and also the higher scavenging efficiency of snow as compared to rain (Wang et al., 2014).

*AR:* Thank you – We also believe that the data presented in the revised version manuscript are unique and analysis is robust. We have very good data. We have improved the clarity of precipitation properties and how we interpret potential impacts of scavenging on our data etc. in **Sects. 3.1, 3.2, and S4** (please see the Track Changes version paper and SI). As discussed in these sections, the estimated scavenging efficiencies of snow are relatively high compared to those of rain as expected (ID# 19 and #21 in **SI Table S6** – almost all scavenged). However, we note that the $M_{sv}$ values of these IDs are not substantially higher compared to those of other rain samples in part due to low $M_0$. Some implications and examples of potential wet scavenging in our INP data are given **in Sect 3.3**.

Our finding on maritime bacteria in West Texas adds an important caveat for the precipitation INP study – a link between microphysics and dynamics beyond regional scale. The authors now extended this discussion in **Sect. 3.4** (please see the Track Changes version paper). The authors indeed wish to continue including this part in the manuscript. We now included the discussion of local and long-range PM sources/transport in **Sect. 3.2**.

*RC:* In contrast, data on particulate airborne matter near ground level is something I would put aside when revising the manuscript.

*AR:* The authors agree and deleted former Fig. 3. We also excluded the PM data collected during precipitation from **SI Table S2**. We note that we used our PM data collected before precipitation to assess the scavenging efficiency of PMs and its impact/implication on our precipitation INP estimation.

*RC:* I found it tedious to read through listings of data in the Results and Discussion section. Somehow, I missed a clear storyline. It would have been a more engaging reading experience, if Figures were not introduced by full sentences that resemble Figure legends.

*AR:* The authors apologize for all of our confusing and cumbersome statements, resulted in an unclear story, in the original discussion manuscript. We gave careful re-interpretation of our data and revisions to remove all logical leaps and insufficient discussions.

*RC:* To give an example (lines 372 and following): "Figure 4 shows the IN spectra for different precipitation types analyzed in this study superposed on the IN spectral boundaries adapted from a previous precipitation INP study (Petters and Wright, 2015). This figure also displays other reference IN spectra, including our 24-hour dry deposition blank sample (collected from January 2 – 3, 2019 at our sampling site) and IN spectra measured for dust suspension samples collected from the downwind side of a local feedlot (identity purposely concealed), where substantial and consistent dust emission historically persists (Whiteside et al., 2018). For the measured T range, nINP values from dry deposition blank sample were at least an order of magnitude lower than that from our precipitation samples." This entire section could simply be replaced with: "For the measured T range, nINP values from dry deposition blank sample were at least an order of magnitude lower than that from our precipitation samples (Figure 4)."

*AR:* The authors took the reviewer's word for it. Thank you.

*RC:* What is meant by (line 313): "...substantial number of precipitation particles with a cumulative number of 2E+05 to 6.6E+05 per event." Perhaps "...precipitation particles recorded by the disdrometer...", or "...precipitation particles per square metre..."?

*AR:* The former of the reviewer's comments is correct. These are the absolute number of precipitation particles passing through the laser beam cross section of and detected by our disdrometer. We have rephrased the manuscript text as follows;
"In our study period, a disdrometer detected a substantial number of precipitation particles with a cumulative number ranging from $1.0 \times 10^4$ to $6.6 \times 10^5$ particles passing through its laser beam cross section per event."

*RC:* Lines 323-327: It is not clear why the range of intensities is indicated as "0 to 150 mm hr-1", when maximum intensity was 129.3 mm hr-1 and minimum intensity 1.1 mm hr-1?

*AR:* Thank you for catching this. The numbers are corrected.

*RC:* References: Whiteside et al. 2018: I would have liked to learn more about this study, but could not find it. A link to the paper, if available, would have been useful.

*AR:* As per request, the authors provide the doi link of Whiteside et al. poster.

Whiteside, C. L., Auvermann, B. W., Bush, J., Goodwin, C., McFarlin, R., and Hiranuma, N.: Ice nucleation activity of dust particles emitted from cattle feeding operations in

the Texas Panhandle, Poster, AMS - 10th Symposium on Aerosol-Cloud-Climate Interactions, Austin, TX, USA, doi: 10.13140/RG.2.2.29505.38248, 2018.

The authors, however, note that more exclusive feedlot INP data (over 2016-2019) generated using the same immersion freezing assay has recently become available in the following ACPD (e.g., Fig. 3):

Hiranuma, N., Auvermann, B. W., Belosi, F., Bush, J., Cory, K. M., Fösig, R., Georgakopoulos, D., Höhler, K., Hou, Y., Saathoff, H., Santachiara, G., Shen, X., Steinke, I., Umo, N., Vepuri, H. S. K., Vogel, F., and Möhler, O.: Feedlot is a unique and constant source of atmospheric ice-nucleating particles, Atmos. Chem. Phys. Discuss., https://doi.org/10.5194/acp-2020-1042, in review, 2020.

Since the data presented in Whiteside et al. 2018 are adapted and merged in this new manuscript, we have replaced Whiteside et al. with Hiranuma et al.

*RC:* Gabor Vali determined INP spectra in rain and hail samples from numerous storms in various parts of North America (Vali, 1968). The authors may find it helpful to have a look at his work when revising their manuscript.

*AR:* The authors appreciate the referee for providing a useful reference. It indeed helped us in better understanding the previous INP measurements from hail and rain samples. We have modified our **Fig. 4**, and added the following discussions in the new **Sect. 3.3**;
"**Figure 4** shows a compilation of $n_{INP}(T)$ spectra of each precipitation type in comparison to previously reported precipitation $n_{INP}(T)$. In general, most of $n_{INP}$ spectra fall in the upper range of the previous precipitation $n_{INP}$ data presented in Petters and Wright (2015) and Vali (1968). INP humps shaping the reference spectra (i.e., one below -20 °C and another at > -20 °C) are also found in our spectra. The observed hump is especially obvious for $n_{INP}$ at $T$ above -20 °C, and some of our spectra exceed the upper bound of the reference spectra in any precipitation types. For $T$s below -20 °C, our $n_{INP}(T)$ data match fairly well within the range of the reference $n_{INP}(T)$ for all four precipitation types. Thus, the precipitation type observed at the ground level would not have any relationships with INP propensity at least for our 42 samples collected for this study. However, it is interesting that most of our $n_{INP}$ data points above -15 °C fall within the range of estimated $n_{INP}$ at cloud height with < 50% storm efficiency, reported in Vali (1968). In fact, regardless of precipitation type, we see reasonable overlaps of our $n_{INP}(T)$ with Vali (1968). The author stated that the large differences in IN content among precipitation samples were mainly caused by differences in the nucleus content of the air entering the storm. This implies that the cloud level dynamics like cloud entrainment impact the cloud level INP concentrations. Hence, we compared our precipitation INP data with the

lower and upper limits of the IN concentrations in the air entering the storm given by Vali (1968) (Table 2, Chapter# 9). These cloud level INP concentrations given by Vali (1968) were for two different storm efficiencies, which is the ratio of mass of precipitation to the mass of water input. The storm efficiency of 10% represents the time when high concentrations of precipitation inside the storm begins to develop. Likewise, 50% is at the peak intensity of the storm. These different combinations of storm efficiencies and water content accounted for a tenfold variation in the ice nucleus content. As more air is entered into the storm with 50% efficiency, more IN concentrations are observed at cloud level. Though our data are comparable to Vali (1968), there is still indeed the need for cloud level INP measurements to define the relationship between the ground level INP concentrations and precipitation intensity."

[Figure]

Figure 4. IN spectra of (a) Snow, (b), Hail/Thunderstorm, (c) Long-Lasted rain, and (d) Weak rain samples superposed on nucleation spectra from previous precipitation INP studies (shaded areas). A subset of spectra shows error bars. The X-axis error bars represent constant uncertainty of ±0.5 °C in temperature. The Y-axis error

bars are the 95% confidence interval for $n_{INP}$ shown only for two samples from each category. The number of precipitation samples in each category is shown by the value of 'n'.

*Summary of Major Changes*

- Our abstract has been revised to reflect all major revisions.
- **Sect. 1.3**: Ambiguous/speculative statements referring to the cloud height condition vs. ground level have been removed; i.e., P3L100-102 and P4L117-120.
- **Sect. 1.4**: Now the study focus is on presenting the ground level observations and measurements, and it is reflected in this particular section with reduced tones.
- **Sect. 3.1**: All repetitive and insufficient statements have been removed or rephrased (e.g., P9L317-322). The authors believe that the readability of this section has improved.
- **Sect. 3.2**: The main focus of this section has been changed to mainly discuss on the wet deposition based on our Air Quality PM sensor data.
- **Sect. 3.3**: Our new data interpretation and comparison to Vali (1968) are now introduced, and our previous statistical analysis has been remove. We re-analyze the $n_{INP}$, precipitation type observed at the ground level, meteorological season, and precipitation intensity data entirely using histograms (new **SI Sect. S5**).
- **Sect. 3.4**: The authors clarified the connection between feedlot and precipitation samples. We have removed some ambiguous results out of a limited number of samples (i.e., previous Figs. 7b and 7c). All associated texts have been modified, and an unnecessary reference has been removed.
- **Sect. 3.5**: Major caveats and limitations are discussed in this new section. After going through the revision process, the authors realize that including caveats for the reader is as important as offering scientific findings.
- Conclusion is also revised to reflect all major changes addressed above.
- **SI Sect. S4**: Detailed discussion of our interpretation of wet scavenging and its impact on our precipitation INP measurements are discussed in this new SI section. The overview is provided in the main manuscript **Sect. 3.2**.

*Minor/technical Changes*

- P1 L3: Dimitri → Dimitrios as per request.
- P6 L195-197: The authors realized that removing the frozen fraction ≤ 0.05, accounting for less than 3% of pure water activation (see **Sect. 2.4**), as an artifact shifts our minimum detection to 0.006 $L^{-1}$ for the current study. This detection limit shift has changed a few INP data (but not a substantial amount). The change has been reflected in **Figs. 1-3**, **S1-S2**, and **Table S4**.

- **Sect. 2.4**: Systematic and experimental uncertainties of WT-CRAFT and our experiments are clarified in more intuitive manner.
- **Sect. 2.6**: Identification of our samples for metagenomics is now provided. Note that the precipitation Sample# 50 (another hail/thunderstorm sample) was preserved only for metagenomics.
- **Fig. 2**: Replaced – all the data connecting lines are now removed to increase the visibility of data points.
- Former Fig. 4: Subdivided into two separate figures (**Figs. 4 and 5**) to clarify the associated discussion (new **Fig. 4**: our precipitation INP vs. previous precipitation INP & new **Fig. 5**: precipitation INP vs. local dust INP). All WT-CRAFT data were presented down to -25 °C.
- **Table S1**: Replaced – meteorological seasons are now used to categorize the sampling season instead of previously introduced arbitrary seasonal categories.
- Former Figs. 3, 6, and S3: Deleted as these figures were misleading/oversimplifying the relevant discussion.
- The reference sections have been updated for both main manuscript and SI. The abbreviation sections have been removed as they might not add much value.
- Cory et al. (2019b) and Rodriguez et al. (2020) are removed from the reference list and the main manuscript as Cory et al. (2019a) can represent a single good reference.
- A new reference (Markowicz and Chiliński, 2020) is added for showing an uncertainty of our PM measurements (see **Sect. 2.3**).
- A new acknowledgement is added for useful scientific discussion for the manuscript revision, "We also acknowledge Drs. Gourihar Kulkarni for useful discussions regarding implications of scavenging processes on our data."

*References*

He, C., Yin, Y., Wang, W., Chen, K., Mai, R., Jiang, H., Zhang, X. and Fang, C.: Aircraft observations of ice nucleating particles over the Northern China Plain: Two cases studies, Atmospheric Research, 248, 105242, 2020.

Hiranuma, N., Auvermann, B. W., Belosi, F., Bush, J., Cory, K. M., Fösig, R., Georgakopoulos, D., Höhler, K., Hou, Y., Saathoff, H., Santachiara, G., Shen, X., Steinke, I., Umo, N., Vepuri, H. S. K., Vogel, F., and Möhler, O.: Feedlot is a unique and constant source of atmospheric ice-nucleating particles, Atmos. Chem. Phys. Discuss., https://doi.org/10.5194/acp-2020-1042, in review, 2020.

Markowicz, K.M., and Chiliński, M.T.: Evaluation of two low-cost optical particle counters for the measurement of ambient aerosol scattering coefficient and Ångström exponent, Sensors, 20, 2617, 2020.

Pereira, D. L., Silva, M. M., García, R., Raga, G. B., Alvarez-Ospina, H., Carabali, G., Rosas, I., Martinez, L., Salinas, E., Hidalgo-Bonilla, S. and Ladino, L. A.: Characterization of ice nucleating particles in rainwater, cloud water, and aerosol samples at two different tropical latitudes, Atmospheric Research, 105356, 2020.

Petters, M. D., and Wright, T. P.: Revisiting ice nucleation from precipitation samples, Geophysical Research Letters, 42, 8758-8766, 2015.

Vali, G.: Ice nucleation relevant to formation of hail, Stormy Weather Group, Ph.D. thesis, McGill University, Montreal, Quebec, Canada, available at https://central.bac-lac.gc.ca/.item?id=TC-QMM-73746&op=pdf&app=Library&oclc_number=894992919 (last accessed on December 21, 2020), 1968.

Whiteside, C. L., Auvermann, B. W., Bush, J., Goodwin, C., McFarlin, R., and Hiranuma, N.: Ice nucleation activity of dust particles emitted from cattle feeding operations in the Texas Panhandle, Poster, AMS - 10th Symposium on Aerosol-Cloud-Climate Interactions, Austin, TX, USA, doi: 10.13140/RG.2.2.29505.38248, 2018.

Zhang, G. F., J. Z. Sun, and E. I. A. Brandes.: Improving parameterization of rain microphysics with disdrometer and radar observations, Journal of the Atmospheric Sciences, 63, 1273-1290, 2006.

---

## Author Comment (AC2) · 22 Dec 2020

*Response to Referee #2*

First of all, the authors thank the referee for submitting helpful and meaningful comments, which lead to improvements and clarifications within the manuscript.

Below, we provide our point-by-point responses. For clarity and easy visualization, the Referee's comments (**RC**) are shown from here on in black. The authors' responses (**AR**) are in blue color below each of the referee's statement. In addition to the responses to referees' comments, we further modified the manuscript to increase its clarity and readability. The summary of major and minor changes is included at the end of this document. We introduce the revised materials in green color along/below each one of your response (otherwise directed to the Track Changes version manuscript).

**RC:** The authors present a set of observation of INP concentrations from rainwater samples collected over West Texas during a 13 months period. They also measure data of precipitation properties, atmospheric temperature, relative humidity and air quality. In addition to that, they performed a metagenomics analysis to obtain information about bacteria present in the rain samples. I personally think that the technical methods used are correctly presented and used. However, the interpretation of their results and the relations they claim be proving between INP and precipitation intensity seems to me completely inaccurate and not backed by their method nor their data at all.

**AR:** The authors appreciate these general remarks and constructive criticisms regarding our manuscript by Referee #2. We believe that the data presented in the revised manuscript are unique and analysis is robust. We have very good data. The authors sincerely hope the referee considers reading the revised manuscript. We respectfully admit that we have made some insufficient discussions, leading some of our data interpretations in an original manuscript to be speculative. Based on the peer-review comments, we removed/modified them to motivate the research. To allay the reviewer's concerns and mitigate any misgivings, the authors have decided to change the title of manuscript to "**Ice-nucleating particles in precipitation samples from West Texas**", reflecting our changes and articulate what is truly presented in the revised version paper. We have also revised our abstract as well as the conclusion to reflect all of our major revisions (please read the Track Changes version paper). Below, we provide our point-by-point responses in hopes of our manuscript being considered for another review by the reviewer. Please know that problems are not stop signs for the authors. We consider these as important guidelines, and we again thank the reviewer for providing us with ones.

**RC:** There are 4 major points why I think this paper should be rejected: -Correlation does not imply causality: The intensity of rain is subject to change due to dynamical a thermodynamical factors. For example, I would expect that the large increases in CAPE over the summer season

make convective clouds much more intense due to stronger updrafts than those observed on the more stratiform precipitation characteristic of winter like cyclone driven rain. None of these points is addressed minimally in the paper although they are the main drivers of precipitation intensity. INP concentrations can also change seasonally due to a variety of factors (dust transport, dryer conditions, higher biological productivity etc...), such factors are also barely mentioned. Finding a correlation between these 2 variables does not imply any type of causality between them. Also, in case you find a strong correlation, you should attempt to see what direction is this going, is it INP affecting rain or rain affecting INP? You could likely do a similar study with any variable, and you might likely find similar correlations.

*AR:* We apologize for extending the analysis to interpret the implications towards raised research questions. The authors concur with the reviewer that using our $n_{INP}$ values in precipitation samples collected at the ground level to assess the impact on precipitation properties at cloud height (and vice versa) is not appropriate. Concerning this and many other issues raised by all peer reviewers (e.g., cloud water ≠ precipitation water), the authors decided to substantially revise the manuscript to focus on presenting the observed variation in precipitation properties and somewhat similar $n_{INP}(T)$ in different precipitation systems and their metagenomic analysis. To address what is raised by the reviewer, the authors decided to discuss all caveats (including CAPE) in the new **Sect. 3.5** - Please see the Track Changes version of the manuscript.
"The precipitation intensity strongly depends on several other dynamical factors and thermodynamic conditions, including the land use, moisture levels, land surface temperatures, and convective available potential energy…"

The seasonal variation in aerosol episodes is now discussed as part of **Sect. 3.2**.
"The seasonal variation in PMs may be indicative of different aerosol particle sources or the local meteorological conditions. In the Southern Great Plains, the local sources include harvesting crop fields and agricultural burning..."

The potential impact of dry condition (plus other ambient and precipitation properties) is now discussed in **Sect. 3.1** - e.g.;
"With an overall average of 54.0%, the highest and lowest relative humidity values measured were 70.7 ± 2.3 % (ID# 26; a weak rain sample) and 30.8 ± 0.7 % (ID# 7; a long-lasted rain sample). The observed low ground level relative humidities during some precipitation events (**Tables S1 - S2**) may be a concern as loss of water through partial evaporation of hydrometeors during free fall. But, it is noteworthy that the water evaporation might have negligible effect on $n_{INP}$ estimated from precipitation samples as discussed in **Sect. 2.5**."

The authors revised the text in **Sect. 2.5** to clarify our points regarding CWC as follows.

"We assumed CWC to be a constant of 0.4 g m$^{-3}$, following Petters and Wright (2015). This assumption would be reasonable for the following three reasons: (1) Petters and Wright (2015) and references therein showed typical values of CWC for different cloud types could narrowly range from 0.2 g m$^{-3}$ to a factor of few more, (2) the authors also showed that the variation of $n_{INP}$ with CWC values for different cloud types in the atmosphere would typically be limited within a factor of two, and our $n_{INP}$ uncertainties could be larger than that, and (3) based on a parametrization for rainwater evaporation, Zhang et al. (2006) suggests that evaporation does not contribute to $n_{INP}$ bias for both strong convective systems and persistent rain events with cloud base heights of ≈3 km. Thus, the variation of CWC on the $n_{INP}$ was considered to be negligible. Nonetheless, it is necessary in the future to further investigate in cloud specific CWCs incorporating with loss of water through partial evaporation of raindrops during free fall based on vertical vapor deficit profiles to conclusively assess if this assumption is fair or not. Precipitation evaporation rate might introduce bias in $n_{INP}$ for precipitation systems with high cloud base, and the correction can be applied accordingly (Petters and Wright, 2015). Direct comparison between INP measurements in cloud water samples and those in precipitation samples might also be key to answer this question (e.g., Pereira et al., 2020)."

Addressing the impact of higher biological productivity on precipitation $n_{INP}$ is a stimulating but tricky task. Agricultural dust is a predominant local dust source in West Texas throughout the year as a number of feedlots exist "within" 33 miles of our precipitation sampling location. Thus, feedlots might act as multiple roles, such as locally emitted INPs, precipitation INPs (if they reach the cloud height), or scavenged aerosol particles. Our result of a dry deposition sample (Sample# 34 – see **Fig. 4**) suggests the limited contribution of local aerosol particles, including feedlot dust. In addition, the authors investigated other dry deposition blank samples (Sample# 35, 38, 39, 40, and 41), and found negligible contribution to $n_{INP}$. We are not including these results because we ran metagenomics only on Sample# 34. Likewise, our assessment of wet deposition, now presented in the new **SI Sect. S4**, shows negligible impact of scavenged PMs on INPs. As the possibility of feedlot dust entering clouds cannot be ruled out, the authors decided to add the discussion of potential contributions of local agricultural dust to precipitation INPs. We have added the following sentences in **Sect. 3.3**;
"Although we are not certain if these local dusts play a role in precipitation, and assessing the potential of locally emitted aerosol particles to precipitation formation is beyond the scope of the current study, it is important to study the contribution of local agricultural dust in wet scavenging and INP formation at cloud height separately in the future. It is noteworthy that adjacent feedlots (> 45,000 head capacity) are located within 33 miles of our sampling site, and the role of feedlot dusts in atmospheric INPs is described in more detail in Hiranuma et al. (2020). Further discussion regarding the feedlot contribution in INPs in our precipitation samples is provided in **Sect. 3.4**."

As further investigation in the bioaerosol impact on precipitation $n_{INP}$ is indeed needed, the authors added the following paragraph in our **Sect. 4**;

"We also identified the similarity in bacterial microbiomes between our precipitation and local feedlot dust samples. While we cannot conclude if local feedlot dust contributes to precipitation formation, we find some indications of the inclusion of agricultural dust in our precipitation samples. Regardless, we did not find the previously known bacterial INPs, such as *Pseudomonas* and *Xanthomonas* (Morris et al., 2004) in either the precipitation or feedlot samples. To further seek a connection between local dust and precipitation, it is worthwhile to characterize the local feedlot dust in cloud water samples, as it can be the source of INPs and may impact the local hydrological cycle. Collecting long-term pollen and other biogenic aerosol particles samples and associated observational data for multiple years may add important knowledge regarding the role of local bioaerosols on precipitation INPs."

Please also see our new **Sect. 3.4**;

"…Although we cannot rule out the possibility that scavenging of aerosolized bacteria explains the presence of these bacteria both in feedlot and precipitation samples taken even at a distance from feedlots, our dry deposition background result shows different biological composition (**Fig. 6**). It is also noteworthy to mention that neither of the genera (*Massilia* and *Marinoscillum*) were detected in the background deposition blank sample and it is not known whether they have any IN activity. Therefore, the scavenging may not be the main reason for the presence of *Massilia* and *Marinoscillum* found in our precipitation samples…"

*RC:* -Wet deposition on rain particles is not properly addressed: Whereas they mention that surface PM does not correlate with INP, this does not discard that wet deposition might be affecting their results.

*AR:* This is absolutely a valid point. We provide our new interpretation and example to explore potential implications of wet scavenging on our data in **Sect. 3.2** and **SI Sect. S4** and to further motivate the research. Some implications and examples of potential wet scavenging in our INP data are given **in Sect 3.3**. Please see the Track Changes version of the manuscript and SI. Briefly, **Fig. 1** (in the next page) shows the estimated INP concentration of scavenged aerosol particles at four different $T$s, $n_{INP,sv}(T)$, based on scavenged mass simulated with column-integrated mean $PM_{10}$ (see **SI Sect. S4**). We also show the measured INP concentrations of our precipitation samples, $n_{INP,pcpt}(T)$ [$L^{-1}$], for comparison. As seen in this figure, our estimated $n_{INP,sv}(T)$ values are constantly much lower as compared to $n_{INP,pcpt}(T)$. This trend is true across all ranges of examined $T$s even if we used the ground level $PM_{10}$ as for scavenging inputs. As noted in **Sect. 3.5**, due to many assumptions we made for this analysis, our results of $n_{INP,sv}(T)$ being much smaller than $n_{INP,pcpt}(T)$ may not be conclusive and indeed requires further detailed

study. Nevertheless, our estimates suggest the presence of $n_{INP,sv}(T)$ in our precipitation samples. Though the estimated $n_{INP,sv}(T)$ values may be negligible, the authors respectfully take the reviewer's words and removed all discussions associated with influence of INP on precipitation intensity etc.

[Figure]

**Figure 1.** (a) Time series of cumulative $n_{INP}$ ($L^{-1}$ air) in each precipitation sample (ID# shown on the x-axis) at different temperatures. (b) Estimated $n_{INP,SV}$ for a total of 28 samples analyzed based on $M_{sv,cm}$. All data above our $n_{INP}$ detection limit of > 0.006 $L^{-1}$ are shown. The average $n_{INP}$ values at -25 °C (74.7 $L^{-1}$) and -20 °C (3.5 $L^{-1}$) in all precipitation samples are shown to guide the reader's eye.

*RC:* Surface PM is not necessarily a measure of free tropospheric aerosol concentrations, and it is well known that during strong precipitation, aerosol concentrations tend to decrease due to wet deposition. The non correlation between PM and INP is perhaps showing that INP concentrations might be independent on the total aerosol concentration, which is likely given their rareness.

*AR:* The reviewer is right. We have re-assessed our hourly averaged PM values right before vs. after precipitation (instead of comparing $n_{INP}$ vs. PM measured 'during' precipitation, as previously offered in Fig. S3). Our measurements of $PM_1$, $PM_{2.5}$, and $PM_{10}$ are summarized in

our new **Table 1**. As can be seen in the table, we confirm the trend of PM reduction for all three PM categories after precipitation in part because of scavenging. Therefore, the authors excluded former Fig. S3 and associated discussions from the revised manuscript. The **Sect. 3.2** was revised accordingly. Please see the Track Changes version of the manuscript.

*RC:* The authors could measure the importance of wet scavenging by analysing the number of particles in their rain samples collected at the surface and just below cloud.

*AR:* Unfortunately, we do not own cloud water samples. Sampling these as demonstrated in previous studies (e.g., Pereira et al., 2020) and investigating their properties would be an important future work, which is now included in **Sect. 3.5**. Please see the Track Changes version of the manuscript.

An approach of measuring the number of particles in suspension samples to assess the importance of wet scavenging is valid, but requires a hydrodynamic light scattering instrument, which any of the authors do not possess. Instead, as presented above, the authors took a different approach to investigate the "first order" impact of wet deposition - that is, to estimate the amount of scavenged aerosol particle mass, $M_{sv}$ (µg m$^{-3}$), for each precipitation event using the PM data (i.e., **SI Sect. S4**).

*RC:* There are strong evidences in their data that point towards wet scavenging being critical, such as how their largest INP concentrations occur on snow samples, which are best at wet scavenging.

*AR:* This is a valid question. Please see the Track Changes version of the manuscript - **SI Sect. S4**. As discussed in this new section, the estimated scavenging efficiencies of snow are relatively high compared to those of rain as expected (ID# 19 and 21 in **SI Table S6** - almost all PM$_{10}$ scavenged). However, we note that the scavenged PM values of these IDs are not substantially higher compared to those of other rain samples in part due to low measured PM.

*RC:* -Their statistical analysis is not presented in detail and strongly limited to a few self-selected data samples: The two-sample t-test is a parametric test. Therefore, first they need to show that their distributions are normal, which I think they probably are in logarithmic scale but not on linear scale. Then, they need to present their results clearly and broadly in a reproducible manner, showing the number of data points going in each of the calculations and which dataset are you comparing. Currently they only show the final p-value for a couple of comparisons at high temperature which to me seems not valid at all for a scientific publication. -Their data seems to show many times the opposite to what they claim: Looking at the available data in the supplementary, I can see that intensity of the rain types increases from snow to weak rain to long-lasted rain to hailstorm (being this last one the most intense) (Table S1-3).

*AR:* We agree. Concerning our limited $n_{INP}$ data (especially thin at -5 °C), we have removed discussions involving p-values and associated with $n_{INP}$(-5°C). The changes have been reflected in the revised manuscript and SI. Instead, the authors re-analyzed the $n_{INP}(T)$ distribution histogram, categorized based on the season, precipitation type, and precipitation intensity, at -10, -15, -20, and -25 °C. The results are presented in **Figs. 2-4** below. Briefly, we first binned our $n_{INP}$ values at each temperature (i.e., -10, -15, -20, and -25 °C) into five equally sized bins by dividing the $n_{INP}$ range (i.e., max - min) at that temperature by the number six. Subsequently, we visualized the frequency distribution of $n_{INP}$ across different bins on a log scale based on the meteorological season in the U.S. (**Fig. 2**), precipitation type (**Fig. 3**), and maximum precipitation intensity (**Fig. 4**). From these results, we found the followings:

- While no clear seasonal variations of $n_{INP}$ values were apparent in part due to the limited number of samples, the analysis of yearlong ground level precipitation observation as well as INPs in the precipitation samples showed that the highest $n_{INP}$ at -25 °C of 1,130 L$^{-1}$ coincided with a hail-involved severe thunderstorm event in the summer.
- On the other hand, the lowest cumulative INP at the same temperature, 3.2 INP L$^{-1}$, was found in one of our snow samples collected during the winter.
- Cumulative $n_{INP}$ in our precipitation samples below -20 °C could be high in the samples collected while observing > 10 mm hr$^{-1}$ precipitation with notably large hydrometeor sizes.

These three findings are now included in our main manuscript text. We include these figures in our **SI Sect. S5** to visualize the data in **Tables S1-2**.

[Figure]

**Figure 2.** The $n_{INP}(T)$ distribution histogram over different $T$s. The histogram frequency is color-categorized for different meteorological seasons (see **Table S1**). The vertical dashed lines and solid line represent 95% confidence intervals and mean $n_{INP}(T)$ value, respectively.

[Figure]

**Figure 3.** The $n_{INP}(T)$ distribution over different $T$s. The histogram frequency is color-categorized for different types of precipitation, including snow, hail/thunderstorm rain, long-lasted ran, and weak rain, observed at the ground level (see **Table S2**).

[Figure]

**Figure 4.** The $n_{INP}(T)$ distribution histogram color-categorized based on three maximum precipitation intensity categories, < 10 mm hr$^{-1}$, 10-50 mm hr$^{-1}$, and > 50 mm hr$^{-1}$ (see **Table S2**).

*RC:* The INP values presented in table S3-1 do not correlate at all with their conclusions, being typically snow the precipitation category with the highest INP measured (at -10, -15, -20 and

-25C) while having the weakest intensity. Of course, this is not the same analysis as performed by the authors, but given the data available in the paper, it seems that the conclusions should, in any case, go the other way around.

*AR:* The reviewer is right. We removed all insufficient discussion regarding the $n_{INP}$-intensity relationship. Again, as discussed above, our new interpretation of data only suggests that cumulative $n_{INP}$ in our precipitation samples below -20 °C could be high in the samples collected while observing > 10 mm hr$^{-1}$ precipitation when we observed large hydrometeor size (i.e., Sample# 1 >> Sample# 60 in **Fig. 2b**).

*RC:* Section 3.2. I like that the authors address the wet deposition factor in this section. However, I do not understand why they relate directly wet deposition with the ambient PM. Wet deposition depends on many factors (size distribution of particles, height from where the droplet falls, etc...)

*AR:* The authors agree that atmospheric deposition depends on many factors. We now include the wet deposition discussion in **Sect. 3.2** and **SI Sect. S4**. The authors included major caveats and limitations of our study in **Sect. 3.5**. Please see the Track Changes version of the manuscript.

*RC:* It could have been much more accurate to measure directly the number of particles in each of their precipitation samples.

*AR:* Measuring the number of particles in our suspension samples to assess the importance of wet scavenging is a valid idea, but requires a hydrodynamic light scattering instrument. Unfortunately, the authors do not own a right instrument, and doing such a rigorous measurements is beyond the scope of the current study.

*RC:* L366. Whereas a measurable decrease in surface PM during rain suggests a clear removal by wet deposition, a non-measurable decrease in surface PM does not discard wet deposition as the particles could have been absorbed higher up and in amounts below the detection limit.

*AR:* The authors agree. Our interoperation of wet deposition is presented in **SI Sect. S4**. Please see the Track Changes version of the manuscript.

*RC:* L393. Snow is a much better scavenger of aerosols than rain. This might be a likely explanation on why you get higher INPs in snow samples. You could test this by measuring the number of particles (and their size distribution) in your snow samples.

*AR:* Discussed above.

*RC:* L397-397. I do not see the link here between these 2 ideas.

*AR:* The authors admit that the analysis/writing was not done properly. We apologize and deleted this sentence.

*RC:* L426-429. This observation goes against the conclusions of the paper. You observed lower -10C INP values during the May-Aug season when precipitation is stronger (due to the appearance of convective storms) than in the Nov-Jan season.

*AR:* The authors agree, and this part is removed from the paper.

*RC:* L430-433 How many points with -5C INPs in the hail/thunderstorm type where included in the analysis. It seems from the plot that there was only 1 point or that all points had the same value. In the supplementary this information is not included.

*AR:* The reviewer is right about invalidity of these thin data. The discussion on -5 °C INPs is removed in the revised main manuscript.

*RC:* L440 As per my previous comments, I am not sure how many points are included in this analysis.

*AR:* Again, the reviewer is right about invalidity of these thin data. The discussion on -5 °C INPs is removed in the revised main manuscript.

*RC:* L445 what are the results of the statistical analysis over the other temperatures? Showing only the -5C p-values is not enough.

*AR:* Discussed above.

*RC:* L447 I don't think this statement is backed by your results.

*AR:* Deleted. We apologize for including such an ambiguous statement.

*RC:* L515-517 Showing some correlations that might be affected by seasonal variations is not enough to claim such a statement.

*AR:* Deleted. We apologize for including such an ambiguous statement.

*Summary of Major Changes*

- Our abstract has been revised to reflect all major revisions.
- **Sect. 1.3**: Ambiguous/speculative statements referring to the cloud height condition vs. ground level have been removed; i.e., P3L100-102 and P4L117-120.
- **Sect. 1.4**: Now the study focus is on presenting the ground level observations and measurements, and it is reflected in this particular section with reduced tones.
- **Sect. 3.1**: All repetitive and insufficient statements have been removed or rephrased (e.g., P9L317-322). The authors believe that the readability of this section has improved.
- **Sect. 3.2**: The main focus of this section has been changed to mainly discuss on the wet deposition based on our Air Quality PM sensor data.
- **Sect. 3.3**: Our new data interpretation and comparison to Vali (1968) are now introduced, and our previous statistical analysis has been remove. We re-analyze the $n_{INP}$, precipitation type observed at the ground level, meteorological season, and precipitation intensity data entirely using histograms (new **SI Sect. S5**).
- **Sect. 3.4**: The authors clarified the connection between feedlot and precipitation samples. We have removed some ambiguous results out of a limited number of samples (i.e., previous Figs. 7b and 7c). All associated texts have been modified, and an unnecessary reference has been removed.
- **Sect. 3.5**: Major caveats and limitations are discussed in this new section. After going through the revision process, the authors realize that including caveats for the reader is as important as offering scientific findings.
- Conclusion is also revised to reflect all major changes addressed above.
- **SI Sect. S4**: Detailed discussion of our interpretation of wet scavenging and its impact on our precipitation INP measurements are discussed in this new SI section. The overview is provided in the main manuscript **Sect. 3.2**.

*Minor/technical Changes*

- P1 L3: Dimitri → Dimitrios as per request.
- P6 L195-197: The authors realized that removing the frozen fraction ≤ 0.05, accounting for less than 3% of pure water activation (see **Sect. 2.4**), as an artifact shifts our minimum detection to 0.006 $L^{-1}$ for the current study. This detection limit shift has changed a few INP data (but not a substantial amount). The change has been reflected in **Figs. 1-3**, **S1-S2**, and **Table S4**.
- **Sect. 2.4**: Systematic and experimental uncertainties of WT-CRAFT and our experiments are clarified in more intuitive manner.

- **Sect. 2.6**: Identification of our samples for metagenomics is now provided. Note that the precipitation Sample# 50 (another hail/thunderstorm sample) was preserved only for metagenomics.
- **Fig. 2**: Replaced – all the data connecting lines are now removed to increase the visibility of data points.
- Former Fig. 4: Subdivided into two separate figures (**Figs. 4 and 5**) to clarify the associated discussion (new **Fig. 4**: our precipitation INP vs. previous precipitation INP & new **Fig. 5**: precipitation INP vs. local dust INP). All WT-CRAFT data were presented down to -25 °C.
- **Table S1**: Replaced – meteorological seasons are now used to categorize the sampling season instead of previously introduced arbitrary seasonal categories.
- Former Figs. 3, 6, and S3: Deleted as these figures were misleading/oversimplifying the relevant discussion.
- The reference sections have been updated for both main manuscript and SI. The abbreviation sections have been removed as they might not add much value.
- Cory et al. (2019b) and Rodriguez et al. (2020) are removed from the reference list and the main manuscript as Cory et al. (2019a) can represent a single good reference.
- A new reference (Markowicz and Chiliński, 2020) is added for showing an uncertainty of our PM measurements (see **Sect. 2.3**).
- A new acknowledgement is added for useful scientific discussion for the manuscript revision, "We also acknowledge Drs. Gourihar Kulkarni for useful discussions regarding implications of scavenging processes on our data."

*References*

He, C., Yin, Y., Wang, W., Chen, K., Mai, R., Jiang, H., Zhang, X. and Fang, C.: Aircraft observations of ice nucleating particles over the Northern China Plain: Two cases studies, Atmospheric Research, 248, 105242, 2020.

Hiranuma, N., Auvermann, B. W., Belosi, F., Bush, J., Cory, K. M., Fösig, R., Georgakopoulos, D., Höhler, K., Hou, Y., Saathoff, H., Santachiara, G., Shen, X., Steinke, I., Umo, N., Vepuri, H. S. K., Vogel, F., and Möhler, O.: Feedlot is a unique and constant source of atmospheric ice-nucleating particles, Atmos. Chem. Phys. Discuss., https://doi.org/10.5194/acp-2020-1042, in review, 2020.

Markowicz, K.M., and Chiliński, M.T.: Evaluation of two low-cost optical particle counters for the measurement of ambient aerosol scattering coefficient and Ångström exponent, Sensors, 20, 2617, 2020.

Morris, C. E., Georgakopoulos, D. G., and Sands, D. C.: Ice nucleation active bacteria and their potential role in precipitation, J. Phys. IV France, 121, 87-103, 2004.

Pereira, D. L., Silva, M. M., García, R., Raga, G. B., Alvarez-Ospina, H., Carabali, G., Rosas, I., Martinez, L., Salinas, E., Hidalgo-Bonilla, S. and Ladino, L. A.: Characterization of ice nucleating particles in rainwater, cloud water, and aerosol samples at two different tropical latitudes, Atmospheric Research, 105356, 2020.

Petters, M. D., and Wright, T. P.: Revisiting ice nucleation from precipitation samples, Geophysical Research Letters, 42, 8758-8766, 2015.

Vali, G.: Ice nucleation relevant to formation of hail, Stormy Weather Group, Ph.D. thesis, McGill University, Montreal, Quebec, Canada, available at https://central.bac-lac.gc.ca/.item?id=TC-QMM-73746&op=pdf&app=Library&oclc_number=894992919 (last accessed on December 21, 2020), 1968.

Whiteside, C. L., Auvermann, B. W., Bush, J., Goodwin, C., McFarlin, R., and Hiranuma, N.: Ice nucleation activity of dust particles emitted from cattle feeding operations in the Texas Panhandle, Poster, AMS - 10th Symposium on Aerosol-Cloud-Climate Interactions, Austin, TX, USA, doi: 10.13140/RG.2.2.29505.38248, 2018.

Zhang, G. F., J. Z. Sun, and E. I. A. Brandes.: Improving parameterization of rain microphysics with disdrometer and radar observations, Journal of the Atmospheric Sciences, 63, 1273-1290, 2006.

---

## Author Comment (AC3) · 22 Dec 2020

*Response to Referee #3*

First of all, the authors thank the referee for submitting helpful and meaningful comments, which lead to improvements and clarifications within the manuscript.

Below, we provide our point-by-point responses. For clarity and easy visualization, the Referee's comments (***RC***) are shown from here on in black. The authors' responses (***AR***) are in blue color below each of the referee's statement. In addition to the responses to referees' comments, we further modified the manuscript to increase its clarity and readability. The summary of minor changes is included at the end of this document. We introduce the revised materials in green color along/below each one of your response (otherwise directed to the Track Changes version manuscript).

The paper is not appropriate for publication. The paper tries to link INP properties and precipitation events of different strength. I was expecting at least some interesting results in Sect. 3.3 (INP results), after reading 8 pages of introductory and technical aspects...and after further reading of the result sections 3.1 and 3.2. But at the end there were no solid findings and convincing results. The paper contains many figures and many speculative statements. This not sufficient and satisfactory.

***AR:*** The authors appreciate these general remarks and diplomatic criticisms regarding our manuscript by Referee #3. We believe that the data presented in the revised manuscript are unique and analysis is robust. We have very good data. The authors sincerely hope the referee considers reading the revised manuscript. We respectfully admit that we have made some insufficient discussions, leading some of our data interpretations in an original manuscript to be speculative. Based on the peer-review comments, we removed/modified them to motivate the research. To allay the reviewer's concerns and mitigate any misgivings, the authors have decided to change the title of manuscript to "**Ice-nucleating particles in precipitation samples from West Texas**", reflecting our changes and articulate what is truly presented in the revised version paper. We have also revised our abstract as well as the conclusion to reflect all of our major revisions - **please read the Track Changes version paper**. Below, we provide our point-by-point responses in hopes of our manuscript being considered for another review by the reviewer. Please know that problems are not stop signs for the authors. We consider these as important guidelines, and we again thank the reviewer for providing us with ones.

***RC:*** My main problem with the manuscript: I am not convinced that one can try to simply link INP concentration measurements at ground with rain events. You need to know cloud base where most of the aerosol particle enter the rain-producing cloud, you need to know cloud top height where ice nucleation typically starts, there may be entrainment of INP from the side...

*AR:* We apologize for extending the analysis to interpret the implications towards raised research questions. The discussion on raised topics is removed in the revised manuscript. It is clear that using our $n_{INP}$ values in precipitation samples collected at the ground level to assess the impact on precipitation properties at cloud height (and vice versa) is not appropriate. Concerning this and many other issues raised by all peer reviewers (e.g., cloud water ≠ precipitation water), the authors decided to substantially revise the manuscript to focus on presenting the observed variation in precipitation properties but somewhat similar $n_{INP}(T)$ in different precipitation systems and metagenomics analysis. Our major revisions include the following:

- Our abstract has been revised to reflect all major revisions.
- **Sect. 1.3**: Ambiguous/speculative statements referring to the cloud height condition vs. ground level have been removed; i.e., P3L100-102 and P4L117-120.
- **Sect. 1.4**: Now the study focus is on presenting the ground level observations and measurements, and it is reflected in this particular section with reduced tones.
- **Sect. 3.1**: All repetitive and insufficient statements have been removed or rephrased (e.g., P9L317-322). The authors believe that the readability of this section has improved.
- **Sect. 3.2**: The main focus of this section has been changed to mainly discuss on the wet deposition based on our Air Quality PM sensor data.
- **Sect. 3.3**: Our new data interpretation and comparison to Vali (1968) are now introduced, and our previous statistical analysis has been remove. We re-analyze the $n_{INP}$, precipitation type observed at the ground level, meteorological season, and precipitation intensity data entirely using histograms (new **SI Sect. S5**).
- **Sect. 3.4**: The authors clarified the connection between feedlot and precipitation samples. We have removed some ambiguous results out of a limited number of samples (i.e., previous Figs. 7b and 7c). All associated texts have been modified, and an unnecessary reference has been removed.
- **Sect. 3.5**: Major caveats and limitations are discussed in this new section. After going through the revision process, the authors realize that including caveats for the reader is as important as offering scientific findings.
- Conclusion is also revised to reflect all major changes addressed above.
- **SI Sect. S4**: Detailed discussion of our interpretation of wet scavenging and its impact on our precipitation INP measurements are discussed in this new SI section. The overview is provided in the main manuscript **Sect. 3.2**.

*RC:* The strength of the thunderstorm or more generally of the rain event depends on the water vapor reservoir and meteorological conditions (sounds trivial) all this is not known here.

*AR:* The reviewer is right. To address what is raised by the reviewer, the authors decided to discuss all caveats (including dynamical factors and thermodynamic conditions) in the new **Sect. 3.5** - Please see the Track Changes version of the manuscript. In addition, the authors also include the discussion of the potential impact of dry conditions (plus other ambient and precipitation properties) on our observations as well as the seasonal variation in aerosol episodes near our study area in **Sects. 3.1 and 3.2**, respectively.

*RC:* Furthermore, on the way to the surface the rain drops collect a lot of aerosol particles (scavenging of pollution, biological and dust particles). All this material you will finally find in the collected rain water.

*AR:* Thanks for clarifying. To explore the scavenging question, we provide examples and implications of scavenging towards our results in **Sect. 3.2** and **SI Sect. S4**. Some implications and examples of potential wet scavenging in our INP data are given **in Sect 3.3**. Please see the Track Changes version of the manuscript and SI.

*RC:* So many questions, I got during reading and reviewing, remained open. The paper must be rejected.

*AR:* The authors hope our responses mitigate the referee's misgivings. We hope this does not end just as an educational opportunity. The authors consider the integration of research and education as an important part of science, and we hope we share the same philosophy with the referee and beyond. Every successful person has a painful story, and we are strong believers that all painful stories deserve successful endings if proper and persistent efforts are made. Please know that we are ready to do what is further required if given the second chance, and we hope our scientific responses prove that we are determined to make it so.

*Minor/technical Changes*

- P1 L3: Dimitri → Dimitrios as per request.
- P6 L195-197: The authors realized that removing the frozen fraction ≤ 0.05, accounting for less than 3% of pure water activation (see **Sect. 2.4**), as an artifact shifts our minimum detection to 0.006 L$^{-1}$ for the current study. This detection limit shift has changed a few INP data (but not a substantial amount). The change has been reflected in **Figs. 1-3**, **S1-S2**, and **Table S4**.
- **Sect. 2.4**: Systematic and experimental uncertainties of WT-CRAFT and our experiments are clarified in more intuitive manner.

- **Sect. 2.6**: Identification of our samples for metagenomics is now provided. Note that the precipitation Sample# 50 (another hail/thunderstorm sample) was preserved only for metagenomics.
- **Fig. 2**: Replaced – all the data connecting lines are now removed to increase the visibility of data points.
- Former Fig. 4: Subdivided into two separate figures (**Figs. 4 and 5**) to clarify the associated discussion (new **Fig. 4**: our precipitation INP vs. previous precipitation INP & new **Fig. 5**: precipitation INP vs. local dust INP). All WT-CRAFT data were presented down to -25 °C.
- **Table S1**: Replaced – meteorological seasons are now used to categorize the sampling season instead of previously introduced arbitrary seasonal categories.
- Former Figs. 3, 6, and S3: Deleted as these figures were misleading/oversimplifying the relevant discussion.
- The reference sections have been updated for both main manuscript and SI. The abbreviation sections have been removed as they might not add much value.
- Cory et al. (2019b) and Rodriguez et al. (2020) are removed from the reference list and the main manuscript as Cory et al. (2019a) can represent a single good reference.
- A new reference (Markowicz and Chiliński, 2020) is added for showing an uncertainty of our PM measurements (see **Sect. 2.3**).
- A new acknowledgement is added for useful scientific discussion for the manuscript revision, "We also acknowledge Drs. Gourihar Kulkarni for useful discussions regarding implications of scavenging processes on our data."

*References*

Markowicz, K.M., and Chiliński, M.T.: Evaluation of two low-cost optical particle counters for the measurement of ambient aerosol scattering coefficient and Ångström exponent, Sensors, 20, 2617, 2020.

Vali, G.: Ice nucleation relevant to formation of hail, Stormy Weather Group, Ph.D. thesis, McGill University, Montreal, Quebec, Canada, available at https://central.bac-lac.gc.ca/.item?id=TC-QMM-73746&op=pdf&app=Library&oclc_number=894992919 (last accessed on December 21, 2020), 1968.

---

## Author Response (AR2)

**Response to Referee #1**

The authors again thank the referee for submitting helpful and constructive comments, which lead to further improvements and clarifications within the manuscript. Below, we provide our point-by-point responses. For clarity and easy visualization, the Referee's comments (*RC*) are shown from here on in black. The authors' responses (*AR*) are in blue color below each of the referee's statement. In addition to the responses to referees' comments, we further modified the manuscript to increase its clarity and readability. The summary of major and minor changes is included at the end of this document. We introduce the revised materials in green color along/below each one of your response (otherwise directed to the Track Changes version manuscript). All references are available in the end of this AR document.

*RC:* Concerns I had raised in my earlier review were taken seriously. The manuscript is is better shape now than initially. Still, there are numerous issues left to solve before it might be in a form where it can recommended for publication. I have to say that my earlier review did not address all issues in detail because the manuscript seemed to require more than one iteration anyway, initially a coarse one and then one to solve remaining intermediate and minor issues.

My main concern with the revised manuscript is its still insufficient focus. It should focus on the immediate objective of the study and the progress made towards achieving it. New insights generated by the study is what the reader is interested in. Currently, these insights are diluted with lengthy explanations of general issues that are well known to most of those who chose to read papers on ice-nucleating particles. For readers, who are not yet familiar with INPs, there are numerous excellent review papers available for a general introduction. I see no need for explaining in the Introduction sections of a specific study in detail what INPs are or why immersion freezing is important.

*AR:* The authors agree with the reviewer that there are many other great review articles regarding atmospheric INPs. Nevertheless, our general introduction of INPs (Sect. 1.1), immersion freezing (Sect. 1.2), and precipitation INPs (Sect. 1.3) contain invaluable information for the reader to follow the story without heavily reading and referring to other papers in parallel. The authors would like to retain these information. In any case, the referee makes a good point, and the authors took the referee's word for it to clarify and extend our study focus in Sect. 1.4.

*RC:* Also section 2.5 can be shortened substantially. I would not call it "IN Parameterization" because what the section describes is the estimation of INP concentration in cloud volume from INP concentration measured in precipitation samples.

*AR:* The authors concur. We now rephrased the subsection title to "*2.5 Precipitation $n_{INP}(T)$ Estimation,*" and shortened the contents accordingly. We keep the most discussion of CWC as this part was extended based on the previous review comments, but decided to move the following content to Sect. 3.5:

"…while assuming a constant CWC may be reasonable to study precipitation INPs (i.e., Sect. 2.5), it is necessary in the future to further investigate in cloud specific CWCs incorporating with loss of water through partial evaporation of raindrops during free fall based on vertical vapor deficit profiles to conclusively assess if this assumption is fair or not. Precipitation evaporation rate might introduce bias in $n_{INP}$ for precipitation systems with high cloud base, and the correction can be applied accordingly (Petters and Wright, 2015). Direct comparison between INP measurements in cloud water samples and those in precipitation samples might also be key to answer this question (e.g., Pereira et al., 2020)."

*RC:* If I was to revise the manuscript, I would start with Section 1.4, Study Objectives. This section summarises what has been done (not necessary in such a section), but does not convincingly state why it has been done. The only statement somewhat pointing in such a direction needs to be specified ("...help

understanding of ambient INPs in the West Texas region..."). Perhaps it means something similar to: "...to understand whether the high density of animals on dusty ground in large feed yards (feedlot), which are typical for Western Texas, has a discernible impact on regional atmospheric INP concentration and composition near the gound and in clouds." If this was indeed the objective, it should be introduced by a short description of feedlot operations, the extent of these operations in terms on number of animals and their spatial extent in Western Texas to give the reader a flavour of these features in the landscape. In most other regions of the world feedlot operations of that kind are unknown. Once the objective is clear, all other changes to the manuscript follow naturally from that point onwards. As I am not sure whether my idea of the specific objective of this study is correct, it makes little sense for me to datail all changes that would follow.

*AR:* Indeed, cattle feedyards can act as a significant point source of local PMs, and they can represent important perturbations to other agricultural INPs - thereby potentially influencing INPs in precipitation if PM reached out to the cloud height and fell back to the ground as part of precipitation particles. Investigating in the potential contributions of cattle feedyard PMs to INPs was definitely one of major research objectives and motivations of the presented study. To incorporate with the reviewer's comment, the authors extended our motivation by revising Sect. 1.4 as follows:

"It is noteworthy that adjacent cattle feedyards (> 45,000 head capacity) are located within 33 miles of our sampling site, and the role of cattle feedyard dusts in atmospheric INPs is described in more detail in Hiranuma et al. (2020)."

→

"In this study, we characterized properties of INPs in precipitation samples collected in the Texas Panhandle region to understand whether the high density of cattle in large open-lot concentrated feeding operation facilities (cattle feedyards hereafter), where often >45,000 head capacity can be seen in a single facility in this region, has a discernible impact on regional atmospheric INP concentration and composition near the ground and in clouds. This region significantly contributes to U.S. cattle production, and the total cattle population of 11 million head accounts for 42% of cattle in the U.S. (according to cattle feedyard research experts at Texas A&M AgriLife Research). Adjacent cattle feedyards are located within 33 miles of our sampling site, and the impact of cattle feedyard dusts in ambient particulate matter (PM), frequently exceeding 1200 µg m$^{-3}$ (24-hour averaged-basis), and aerosol particle composition as well as an overall regional air quality is described in Hiranuma et al. (2011) and Von Essen and Auvermann (2005). Moreover, the emission flux of PM smaller than < 10 µm diameter (PM$_{10}$) is typically high in the range of 4.5 µg m$^{-2}$ s$^{-1}$ up to 23.5 µg m$^{-2}$ s$^{-1}$ depending on stocking density, creating PM-laden ambient conditions in this particular region (Bush et al., 2014)."

The authors also added the following sentence in the second paragraph of Sect. 1.4 to clarify that we compared the metagenomics result of precipitation to that of cattle feedyard PM samples in this study:
"Some of water-suspended cattle feedyard PM samples were also analyzed with metagenomics to find bacterial microbiome that may appear in precipitations."

In Sect. 2.6, the authors clarified the focus of our metagenomics analysis - "The overall goal of our metagenomics analysis was to identify known ice-nucleation-active bacterial species in cattle feedyard dust, collected in commercial cattle feedyards located within 33 miles from the precipitation sampling site and suspended in the high-performance liquid chromatography grade water (Hiranuma et al., 2020), …"

The authors replaced "feedlot" with "cattle feedyard" to specify the type of animal feeding activities discussed in this manuscript in more accurate and descriptive manner (based on advice provided by a local cattle feedyard research expert). Additionally, we also renamed the subsection title to "*1.4. Study Motivation and Objectives*"

*RC:* Line 17: Please specify that "nINP" stands for INP concentration in air.
*AR:* Specified and corrected – it now reads INP concentrations per unit volume of air ($n_{INP}$).
The authors also reflected this change to its first appearance in the main text (Sect. 1.3);
increased $n_{INP}$ → ambient INP concentration ($n_{INP}$)

*RC:* Lines 78 to 82: I do not understand the argument. There is a high number concentration of INPs (up to 20'000 per litre) and at the same time there are liquid droplets. Why should this be an argument for immersion freezing to be important? If it had happened in these clouds, there would have been up to 20'000 ice crystals formed per litre and the clouds would quickly have completely glaciated, so liquid droplets would not have been observed.
*AR:* The discussion on raised topics is removed in the revised manuscript. Instead, we have added a more relevant reference (Westbrook and Illingworth, 2011) in Sect. 1.2.
"Similarly, an importance and predominance of supercooled liquid droplets as for a prerequisite of atmospheric ice formation is reported in Westbrook and Illingworth (2011). The authors verified it based on radar and lidar observations of clouds over the U.K. at temperatures relevant to immersion freezing."

*RC:* Line 92: What is an "INP episode"?
*AR:* We have removed the "INP episode" word, and clarify the sentence by extending the sentence as follows:
This latent heat is further influenced by INP episode, thus affecting the dynamics of the precipitation system.
→
When immersion freezing occurs, the latent heat of freezing energy can be released. Thus, INPs themselves can impact the dynamics of the precipitation system.

*RC:* Lines 98 to 100: Do you mean "...have shown that the addition of INPs at the base of warm clouds would result in stronger updrafts and lead to increased amounts of precipitation..."?
*AR:* Yes. The referee is right. Thank you. We corrected the sentence as suggested.
"…have shown that the addition of INPs at the base of warm clouds results in stronger updrafts and lead to increased amounts of precipitation…"

*RC:* Line 121: "studied" or "compared"?
*AR:* "Compared" would be more appropriate word choice. Thank you.

*RC:* Line 123: What is "bio-speciation"?
*AR:* We meant taxonomic identification. We have replaced all other "speciation" words to either analysis or characterization for clarity.
e.g., P1L27: metagenomics analysis of ambient dust samples → metagenomics characterization of the bacterial microbiome in suspended ambient dust samples

*RC:* Line 263/264: What are the feedlot samples, airborne dust collected next to a feedlot or soil samples taken from the surface of a feedlot?
*AR:* These are airborne particulate matter collected at the downwind location of typical large commercial cattle feedyards in West Texas (1.5 m above ground level). We have revised the sentence as:
"…we have examined a heterogeneous set of samples including four airborne PM samples locally collected at the downwind location of typical commercial cattle feedyards in West Texas on March 28, 2019 and…"

*RC:* Line 267: "tornado warning", not "tornado warming"
*AR:* Corrected. Thank you.

*RC:* Lines 218 to 220: I do not understand why in the case of overlapping INP spectra those with the "lower nINP values" were taken for merging when they had the "lowest confidence intervals". Did you mean "lower uncertainty"?
*AR:* Yes. Thanks for catching this. We meant lower uncertainty. Corrected as suggested/stated.

*RC:* Line 229: Please say what "CINP" stands for.
*AR:* It is defined in the second sentence in Sect. 2.5 as "the nucleus concentration in precipitation suspension ($L^{-1}$ water) at a given *T*."

*RC:* Line 239: What is a "factor of few more"?
*AR:* It means 0.2 to 0.8 g $m^{-3}$. We rephrased the senescence as:
"…typical values of CWC for different cloud types could narrowly range within a factor of two from 0.4 g $m^{-3}$…"

*RC:* Line 346: A standard deviation that is larger than the mean implies that there is a substantial fraction of data with negative values. This is not possible when the unit of the data is mass per volume. Most likely, the data does not have a normal, but a log-normal distribution. If so, it should be treated and reported accordingly (cf. Limpert et al, 2001, https://doi.org/10.1641/0006-3568(2001)051[0341:LNDATS]2.0.CO;2, or Limpert et al., 2008, DOI 10.1007/s10453-008-9092-4)
*AR:* Those negative powers of hundredth and tenth are only effective for the standard error values. So all means are larger than error values. We apologize for causing this confusion. We have corrected our text and numerical expressions as:
$3.9 \pm 0.0_9$ $\mu$g $m^{-3}$ ($PM_{1.0}$), $4.0 \pm 0.0_5$ $\mu$g $m^{-3}$ ($PM_{2.5}$), and $10.0 \pm 0.2_2$ $\mu$g $m^{-3}$ ($PM_{10}$).

*RC:* Lines 350 to 362: A range of potential sources of particulate matter in the sampled air are discussed, but why not soil dust from feedlot? I thought this study intends to understand the possible impact of this source on regional INP concentrations in air and precipitation (or clouds)?
*AR:* The referee is right about PM from cattle feedyard to be another significant PM source in this region. Admittedly, we overlooked to mention it. We have updated the text as:
"Besides the local PMs originating from cattle feedyards as described in Sect. 1.4, other prominent local sources include harvesting crop fields and agricultural burning In the Great Plains region nearby West Texas..."

*RC:* Line 367: What do you mean with "surface material rupture"?
*AR:* We meant "surface material ejected by water impaction of rainfall (e.g., Huffman et al., 2013; Wang et al., 2016)." For clarity, we added another reference (Wang et al., 2016) for supporting this sentence.

*RC:* Section 3.4: Here, I am missing a discussion of the results of the present study in a wider context. There have been others, who analysed the microbiome of precipitation samples (e.g. Woo and Yamamoto, 2020, https://doi.org/10.1186/s40793-020-00369-4). How do their results compare to your results. Is there much similarity? Are there major differences, perhaps pointing at the influence of dust from feedlot operations? I am not expecting a list of names and percentages of microorganisms found in numberous other places ot the world, but a discussion focused on what I presume is the aim of your study: do feedlot operations affect what is found in terms of microorganisms and INPs in air and precipitation in Western Texas.

*AR:* In Abstract, we added "…Some key bacterial phyla present in cattle feedyard samples appeared in precipitation samples. However, no known ice nucleation active species were detected in our samples."
IN activity of *Massilia* reported in Woo and Yamamoto is not conclusive yet, but the authors added the following sentence along with another new citation (Jimenez-Sanchez et al., 2018).
In Sect. 3.4, we added "Genera *Massilia* and *Sphingomonas* have been reported as weak IN active species (Jimenez-Sanchez et al., 2018), but these results are inconclusive and the discussion is ongoing at this stage (Woo and Yamamoto, 2020)."

*RC:* Line 460: I do not understand the statement in brackets: Did the back-trajectories show a possible marine influence or not?
*AR:* Yes, they did. For our back-trajectory analysis we used the HYSPILT-READY model with Global Data Assimilation System (1 degree) meteorological data as input (Stein et al., 2015; Rolph et al., 2017). The back-trajectory analysis for our precipitation sampling periods (i.e., PCPT 1-4 in Fig. 6) was carried out at different heights over our precipitation sampling location; i.e., 500, 1000, and 3000 m above ground level (assuming these as the typical cloud heights). For example, as represented in the PCPT 4 example shown below, our back-trajectory analysis indicates a possible maritime influence from the Atlantic Ocean through the Caribbean Sea and Gulf of Mexico.

[Figure]

**Figure.** NOAA HYSPLIT 7 day backward trajectory calculations are given for our precipitation samples of PCPT 4 which corresponds to our Sample# 7 for 500, 1000, and 3000 m above ground level (AGL).

Furthermore, for the cattle feedyard samples 1-4 (Fig. 6), the back-trajectory analysis was carried out at 1.5 m above ground level (i.e., our sampling height at cattle feedyards). The authors note that the back-trajectories for the cattle feedyard samples are intentionally not included anywhere in this response and our manuscript to protect the royalty of our commercial cattle feedyard partners and not to disclose any geographical information of the cattle feedyards. Regardless, the results from all of our back-trajectory analyses indicated that the air masses were originated from the Atlantic Ocean (through Gulf of Mexico and Caribbean Sea) and/or the Pacific Ocean. Overall, these results support a possible marine influence in our cattle feedyard and precipitation samples.

We have extended the back-trajectory discussion in Sect. 3.4 as:

"Additionally, in one hailstorm sample, we also identified *Gilvimarinus*, which is another marine genus of *γ-Proteobacteria* (**Table S9**). These results indicate some connection with air mass originating from ocean. To verify this point, we performed back-trajectory analysis using the HYSPILT-READY model with Global Data Assimilation System (1 degree) meteorological data as input (Stein et al., 2015; Rolph et al., 2017). The analysis for our precipitation sampling periods (i.e., PCPT 1-4 in **Fig. 6**) was carried out at different heights over our precipitation sampling location; i.e., 500, 1000, and 3000 m above ground level (assuming these as the typical cloud heights). Furthermore, for the cattle feedyard samples 1-4 (**Fig. 6**), the back-trajectory analysis was carried out at the sampling height, which is 1. 5m above ground level. Overall, all these back-trajectories indicate a possible maritime influence through the Caribbean Sea, Gulf of Mexico and/or the Pacific Ocean (not shown). Thus, these results support a possible marine influence in our precipitation and cattle feedyard samples."

*RC:* Line 459: What was the limit of detection again?
*AR:* Theoretically, there is no detection limit in any metagenomics studies. Even if the DNA from a single cell is extracted and amplified, these species would be detected. In our supplemental table S9, the 0% of several bacteria phyla is exactly this case: a very low percentage of these phyla in the microbiome, which nevertheless were detected. For this study, we only report >0.1% OTU/ASV.

*RC:* Section 4. Conclusion: The first paragraph (lines 510 to 533) is a summary of results and only the second, much shorter paragraph contains conclusions. I would suggest to shorten the first and to strengthen the second paragraph.
*AR:* The authors agree. We changed the section tile to Summary and Conclusion, and excluded the following sentences from the first paragraph to shorten texts:

"Our disdrometer measurements showed a clear variation in the precipitation properties among the four different categories of precipitation samples. Severe precipitation, such as hail/thunderstorms, had the highest rainfall intensity (mm hr$^{-1}$) and the number of precipitation particles were highest in the snow samples. We also found an increased number of large hydrometeors (> 10 mm in diameter) in both the snow and hail/thunderstorm samples. In contrast, there were no precipitation particles > 6.5 mm in diameter observed in the weak rain samples. Our PM concentration measurements implied some possibilities of wet deposition (but neglected). The IN spectra from each precipitation category in this study were compared with the IN spectra from previous precipitation-based INP studies (Petters and Wright, 2015; Vali, 1986)."

Furthermore, as suggested, we extended our second paragraph as follows; "Our metagenomics results suggest the presence of marine genera *Marinoscillum* and *Gilvimarinus* in precipitation and cattle feedyard PM samples. These genera may have derived by an influence of air mass originating from maritime regions. Marine bacteria in inland sampling sites have been identified in previous studies (e.g., Cho and Jang, 2014). We also identified bacterial genera common in our precipitation as well as the local cattle feedyard dust samples, while the microbiome composition in one feedyard sample (Feedyard 3 **in Fig. 6**) was considerably different from the microbiome composition in precipitation samples. The difference of the microbiomes in dry and wet deposition samples, suggesting a non-local origin of bioaerosols in precipitation, has also been observed previously over crops (Constantinidou et al., 1990), as well as in urban precipitation samples (Cho and Jang, 2014; Woo and Yamamoto, 2020). While we cannot conclude if local cattle feedyard dust contributes to precipitation formation, we also found some indications of the inclusion of agricultural dust in our precipitation samples. Regardless, we did not find previously known bacterial INPs, such as *Pseudomonas* and *Xanthomonas* (Morris et al., 2004) in either the precipitation or cattle feedyard samples. To further seek a connection between local dust and precipitation, it is worthwhile to characterize the local cattle feedyard dust in cloud water samples, as it

can be the source of INPs and may impact the local hydrological cycle. Collecting long-term pollen and other biogenic aerosol particles samples (i.e., *Fungi* and *Archaea*) and associated observational data for multiple years may add important knowledge regarding the role of local bioaerosols on precipitation INPs. Besides DNA analysis, analysis of RNA by metatranscriptomics will provide insights on the active life of the microbiome in clouds and precipitation. Ultimately, both DNA and RNA analysis of the microbe in ice crystal residuals would offer a direct link between naturally-occurring biological particles and INPs."

*RC:* Lines 514 to 515: Is it not self-evident that severe precipitation has the highest rainfall intensity? It is measured rainfall intensity that leads to the categorisation of an event as severe rainfall. Or, was there any other criteria for that category?
*AR:* As summarized in Table S3, the Hail/Thunderstorm precipitation type showed the highest average intensity as well as maximum intensity when compared to other precipitation types beyond standard error.

*RC:* Table 1: Are the PM really reliable to a precision of 1 ng/m3? If not, reduce the number of digits.
*AR:* This is a good question and suggestion. A previous publication reported only one decimal digit for PM in $\mu g\ m^{-3}$ measured by an identical sensor to correctly represent the detectable PM by this particular sensor (Hegde et al., 2020). The authors decided to follow the same procedure to report only one decimal place and changed all numbers accordingly as shown in the table below:

**Table 1**. Adjacent hourly averaged PM values (with one decimal point) before and after each precipitation event. We excluded 14 data where PM data were not recorded due to technical issues etc. (ID# of 6-7, 17, 20, 22-24, 26, 28-33).

| ID# | Sample# | Precipitation type | $PM_1$ ($\mu g\ m^{-3}$) Before | After | $PM_{2.5}$ ($\mu g\ m^{-3}$) Before | After | $PM_{10}$ ($\mu g\ m^{-3}$) Before | After |
|---|---|---|---|---|---|---|---|---|
| 1 | PCPT_NSB_1 | Hail/Thunderstorm | 2 | 0.1 | 4.1 | 1.7 | 6.2 | 2 |
| 2 | PCPT_NSB_2 | Hail/Thunderstorm | <0.1 | 0 | 1.8 | <0.1 | 2.1 | <0.1 |
| 3 | PCPT_NSB_5 | Long-Lasted Rain | 4.7 | 0.7 | 5.7 | 1.9 | 10.8 | 3.7 |
| 4 | PCPT_NSB_6 | Long-Lasted Rain | 3.8 | 3.8 | 6 | 5.7 | 8.9 | 8.6 |
| 5 | PCPT_NSB_7 | Hail/Thunderstorm | 0 | N/A | 0.6 | N/A | 0.7 | N/A |
| 8 | PCPT_NSB_10 | Long-Lasted Rain | 7.5 | 1.5 | 9.9 | 3.4 | 14.8 | 4.7 |
| 9 | PCPT_NSB_11 | Weak Rain | 5.8 | 3.8 | 8.2 | 6.2 | 12.8 | 9.4 |
| 10 | PCPT_NSB_15 | Hail/Thunderstorm | 14.3 | 4 | 16.1 | 5.1 | 30.8 | 9.3 |
| 11 | PCPT_NSB_16 | Hail/Thunderstorm | 4.9 | N/A | 5.4 | N/A | 10.5 | N/A |
| 12 | PCPT_NSB_17 | Long-Lasted Rain | 4.6 | N/A | 6.4 | N/A | 10.6 | N/A |
| 13 | PCPT_NSB_19 | Weak Rain | <0.1 | N/A | 1.3 | N/A | 6.3 | N/A |
| 14 | PCPT_NSB_20 | Long-Lasted Rain | 1.8 | N/A | 4.3 | N/A | 5.9 | N/A |
| 15 | PCPT_NSB_23 | Hail/Thunderstorm | 3.9 | 2.2 | 5.7 | 5.7 | 9.6 | 7.2 |
| 16 | PCPT_NSB_24 | Hail/Thunderstorm | 1.6 | 0 | 5 | <0.1 | 5.8 | <0.1 |
| 18 | PCPT_NSB_26 | Long-Lasted Rain | 0.7 | 0 | 2.8 | 0 | 3.2 | 0 |
| 19 | PCPT_NSB_27 | Snow Sample | 0 | N/A | <0.1 | N/A | 0.1 | N/A |
| 21 | PCPT_NSB_30 | Snow Sample | 0.8 | 0 | 2.6 | 0.3 | 3.2 | 0.3 |
| 25 | PCPT_NSB_46 | Weak Rain | 1.5 | 0 | 4.5 | 1.2 | 5.4 | 1.2 |
| 27 | PCPT_NSB_48 | Hail/Thunderstorm | 0 | 0 | 0.4 | <0.1 | 0.4 | <0.1 |
| 34 | PCPT_NSB_57 | Hail/Thunderstorm | 29.6 | 13.5 | 29.6 | 13.8 | 58.9 | 26.6 |
| 35 | PCPT_NSB_58 | Hail/Thunderstorm | 12.5 | 0.7 | 13.2 | 1.4 | 24.4 | 2.9 |
| 36 | PCPT_NSB_59 | Long-Lasted Rain | 10.5 | 6.9 | 11.5 | 7.9 | 21.2 | 12.9 |
| 37 | PCPT_NSB_60 | Hail/Thunderstorm | 9.7 | 3.4 | 10.7 | 4.4 | 18.8 | 7.3 |
| 38 | PCPT_NSB_61 | Long-Lasted Rain | 4.4 | 0.2 | 5.9 | 1.2 | 10.1 | 2.1 |
| 39 | PCPT_NSB_62 | Hail/Thunderstorm | <0.1 | N/A | 1.6 | N/A | 1.8 | N/A |
| 40 | PCPT_NSB_63 | Hail/Thunderstorm | 2.2 | 1.4 | 4.3 | 2.5 | 6.5 | 4.8 |
| 41 | PCPT_NSB_65 | Hail/Thunderstorm | 1.7 | 0 | 4 | 0.3 | 5.3 | 0.3 |
| 42 | PCPT_NSB_66 | Hail/Thunderstorm | 1.8 | 0.1 | 2.9 | 1.5 | 5.8 | 1.5 |

NOTE: N/A: either below detection sensor failure return values (i.e., detection limit of our PM sensor).

In addition, the authors made a few technical language/grammar changes without changing context since the referee rated the presentation quality fair but not good yet. Our changes include:

- Throughout the manuscript, the use of appropriate articles, plural nouns, and pre-position words has been corrected wherever applicable. An appropriate use of English language was re-checked by native English speakers.
- P1L4: Greg D. Mayer → Gregory D. Mayer
- P1L23: lowest at -25 °C → lowest $n_{INP}$ values at -25 °C
- P1L28: to check… → to ascertain whether local cattle feedyards can act as
- P4L143: examine → determine
- P4L147: → ranging from several hundred to several thousand
- P8L303: → 5μM primer mix
- P27L890: Metagenomics analysis → Bacterial community analysis

The authors will use an external data depository to increase a public awareness of our data (e.g., https://issues.pangaea.de/). Therefore, we rephrased our Data Availability from "Original data created for the study will be available in a persistent repository upon publication within www.wtamu.edu. → Original data created for the study are or will be available in a persistent repository (pangaea.de) upon publication.

Naruki Hiranuma now acts as a single corresponding author for the revised manuscript as he led the revision effort.

**Refrences:**

Bush, J., Heflin, K. R., Marek, G. W., Bryant, T. C., and Auvermann, B. W.: Increasing stocking density reduces emissions of fugitive dust from cattle feedyards, Applied Engineering in Agriculture, 30, 815-824, 2014.

Cho, B. C., and Jang, G. I. : Active and diverse rainwater bacteria collected at an inland site in spring and summer 2011, Atmospheric Environment, 94, 409-416, 2014.

Constantinidou, H. A., Hirano, S. S., Baker, L. S., and Upper, C. D.: Atmospheric dispersal of ice nucleation-active bacteria: the role of rain, Phytopathology, 80, 934-97, 1990.

Hegde, S., Min, K.T., Moore, J., Lundrigan, P., Patwari, N., Collingwood, S., Balch, A. and Kelly, K.E.: Indoor Household Particulate Matter Measurements Using a Network of Low-cost Sensors. *Aerosol Air Qual. Res.,* 20, 381-394, https://doi.org/10.4209/aaqr.2019.01.0046, 2020.

Jimenez-Sanchez, C., Hanlon, R., Aho, K. A., Powers, C., Morris, C. E., and Schmale, D. G. III: Diversity and ice nucleation activity of microorganisms collected with a small Unmanned Aircraft System (sUAS) in France and the United States. Front. Microbiol., 9, 1667, 2018.

Rolph, G., Stein, A., and Stunder, B.: Real-time Environmental Applications and Display sYstem: READY. Environmental Modelling & Software, 95, 210–228, 2017.

Stein, A.F., Draxler, R.R, Rolph, G.D., Stunder, B.J.B., Cohen, M.D., and Ngan, F.: NOAA's HYSPLIT atmospheric transport and dispersion modeling system, Bull. Amer. Meteor. Soc., 96, 2059–2077, 2015.

Von Essen, S. G. and Auvermann, B. W.: Health effects from breathing air near CAFOs for feeder cattle or hogs, J Agromedicine, 10, 55-64, 2005.

Wang, B., Harder, T. H., Kelly, S. T., Piens, D. S., China, S., Kovarik, L., Keiluweit, M., Arey, B. W., Gilles, M.K., and Laskin, A.: Airborne soil organic particles generated by precipitation, Nature Geosci., 9, 433-437, 2016.

Westbrook, C. D., and Illingworth, A. J.: Evidence that ice forms primarily in supercooled liquid clouds at temperatures > −27 °C, Geophys. Res. Lett., 38, L14808, 2011.

Woo, C., and Yamamoto, N.: Falling bacterial communities from the atmosphere, Environmental Microbiome, 15, 22, 2020.

---

## Author Response (AR3)

| | |
|---|---|
| **From:** | Hiranuma, Naruki |
| **To:** | "editorial@copernicus.org"; susannah.burrows@pnnl.gov |
| **Subject:** | RE: [External] acp-2020-863 (author) - manuscript accepted with corrections |
| **Date:** | Wednesday, February 17, 2021 11:55:25 AM |

Dear Dr. Burrows and the ACP editorial team,

I am writing to acknowledge the receipt of your correction comment. The authors have agreed on revising the text as suggested (not shown --> not shown to protect location privacy), and I will upload the updated production files accordingly.

On behalf of the authors, I would like to express our appreciation to your continued support and all invaluable comments from referees. We hope to continue sharing success with you and ACP.

Best wishes,

Naruki Hiranuma
* * *
Naruki Hiranuma | Assistant Professor of Environmental Science
West Texas A&M University | Dept. of Life, Earth and Environmental Sciences
WTAMU Killgore Research Center 119, Canyon, TX 79016-0001
Phone: (806) 651-3872 | Fax: (806) 651-2928 | Email: nhiranuma@wtamu.edu

-----Original Message-----
From: editorial@copernicus.org <editorial@copernicus.org>
Sent: Tuesday, February 16, 2021 11:53 AM
To: Hiranuma, Naruki <nhiranuma@wtamu.edu>
Cc: Hiranuma, Naruki <nhiranuma@wtamu.edu>
Subject: [External] acp-2020-863 (author) - manuscript accepted with corrections

Dear Naruki Hiranuma,

We are pleased to inform you that the Editor report for the following ACP manuscript is now available:

Title: Ice-nucleating particles in precipitation samples from West Texas
Author(s): Hemanth S. K. Vepuri et al.
MS No.: acp-2020-863
MS type: Research article
Iteration: Correction

The Editor has decided that some corrections are necessary before the manuscript can be published. Please log in using your Copernicus Office user ID 203002 to find the Editor report at:
https://editor.copernicus.org/ACP/ms_records/acp-2020-863

We kindly ask you to upload the files required for the production process no later than 24 Feb 2021 at:
https://editor.copernicus.org/ACP/production_file_upload/acp-2020-863

Please find all information on manuscript submission at: https://www.atmospheric-chemistry-and-physics.net/for_authors/submit_your_manuscript.html

Please note that all Referee and editor reports, the author's response, as well as the different manuscript versions of the peer-review completion (post-discussion review of revised submission) will be published along with your final-revised paper in ACP.

You are invited to monitor the processing of your manuscript via your MS overview at:

[https://editor.copernicus.org/ACP/my_manuscript_overview](https://editor.copernicus.org/ACP/my_manuscript_overview)

In case any questions arise, please do not hesitate to contact me. Thank you very much for your cooperation.

Kind regards,

The editorial support team
Copernicus Publications
editorial@copernicus.org